# Psoriatic skin inflammation is promoted by c-Jun/AP-1-dependent CCL2 and IL-23 expression in dendritic cells

Philipp Novoszel[1] , Martin Holcmann[1], Gabriel Stulnig[1], Cristiano De Sa Fernandes[1], Victoria Zyulina[2], Izabela Borek[2], Markus Linder[1], Alexandra Bogusch[1], Barbara Drobits[1] , Thomas Bauer[1], Carmen Tam-Amersdorfer[2] , Patrick M Brunner[3], Georg Stary[3], Latifa Bakiri[3,4], Erwin F Wagner[3,4] , Herbert Strobl[2] & Maria Sibilia[1,*]

## Abstract

Toll-like receptor (TLR) stimulation induces innate immune responses involved in many inflammatory disorders including psoriasis. Although activation of the AP-1 transcription factor complex is common in TLR signaling, the specific involvement and induced targets remain poorly understood. Here, we investigated the role of c-Jun/AP-1 protein in skin inflammation following TLR7 activation using human psoriatic skin, dendritic cells (DC), and genetically engineered mouse models. We show that c-Jun regulates CCL2 production in DCs leading to impaired recruitment of plasmacytoid DCs to inflamed skin after treatment with the TLR7/8 agonist Imiquimod. Furthermore, deletion of c-Jun in DCs or chemical blockade of JNK/c-Jun signaling ameliorates psoriasis-like skin inflammation by reducing IL-23 production in DCs. Importantly, the control of IL-23 and CCL2 by c-Jun is most pronounced in murine type-2 DCs. CCL2 and IL-23 expression co-localize with c-Jun in type-2/inflammatory DCs in human psoriatic skin and JNK-AP-1 inhibition reduces the expression of these targets in TLR7/8-stimulated human DCs. Therefore, c-Jun/AP-1 is a central driver of TLR7-induced immune responses by DCs and JNK/c-Jun a potential therapeutic target in psoriasis.

**Keywords** c-Jun; dendritic cells; psoriasis; targeted therapy; TLR7
**Subject Categories** Immunology; Skin

## Introduction

Toll-like receptors (TLRs) are pattern recognition receptors of the innate immune system. TLRs recognize conserved motifs present on pathogens. Expression and ligand specificity is unique for each TLR and enables the recognition of bacterial, viral, and fungal pathogens (Iwasaki & Medzhitov, 2004). However, host-derived ligands such as self-RNA/DNA can cause inappropriate activation of TLR7/9, triggering auto-immune diseases such as systemic lupus erythematosus (SLE) or psoriasis (Lande *et al*, 2007; Ganguly *et al*, 2009).

Psoriasis is a multifactorial, chronic, inflammatory skin disease whose pathogenesis is dependent on predisposing genetic mutations and environmental triggers (Wagner *et al*, 2010). It has been demonstrated that initially, self-RNA/DNA released from stressed or dying keratinocytes stimulates intracellular TLRs in plasmacytoid DCs (pDCs) and cutaneous DCs to initiate a Th$_1$- and Th$_{17}$-dependent immune response culminating in the activation of additional immune and skin cells. This leads to immune cell-rich, thickened skin regions clinically characterized as lesional psoriasis (Boehncke & Schön, 2015). Some of these effects can be mimicked by small molecule immune response modifiers of the Imidazoquinoline family, including Imiquimod (IMQ) or Resiquimod (R848), which are potent immunostimulators of the antiviral TLR7/8 signaling pathway (Hemmi *et al*, 2002).

In mice, IMQ application induces a skin inflammation characterized by epidermal thickening, pDC and neutrophil influx, and inflammatory cytokine production (van der Fits *et al*, 2009). Using genetically engineered mouse models (GEMMs), DCs (Tortola *et al*, 2012) and $\gamma\delta$-T cells (Pantelyushin *et al*, 2012) were identified to mediate the production of IL-23 (Wohn *et al*, 2013) and IL-17 A/F (Pantelyushin *et al*, 2012; Riol-Blanco *et al*, 2014), respectively, to induce the inflammatory phenotype.

TLR7 and TLR8 are expressed in the endosomal compartment of pDCs, monocytes, and B cells. Signal transduction via the adaptor protein MyD88 results in the activation of transcription factors like NF-κB or IRF7 along with components of the AP-1 family. NF-κB induces transcription of various inflammatory cytokines, whereas

1   Department of Medicine I, Comprehensive Cancer Center, Institute of Cancer Research, Medical University of Vienna, Vienna, Austria
2   Division of Immunology and Pathophysiology, Otto Loewi Research Center, Medical University of Graz, Graz, Austria
3   Division of Immunology, Allergy and Infectious Diseases, Department of Dermatology, Medical University of Vienna, Vienna, Austria
4   Department of Laboratory Medicine, Medical University of Vienna, Vienna, Austria
    *Corresponding author. Tel: +43 1 40160 57502; E-mail: maria.sibilia@meduniwien.ac.at

IRF7 is critical for the production of Type-I interferons (Kaisho & Tanaka, 2008). However, the role of AP-1 proteins downstream of TLR7/8 has so far been barely investigated.

AP-1 proteins are basic leucine zipper transcription factors that form homo-or heterodimers. They include members of the Jun family, such as c-Jun and JunB, which play a prominent role in the skin. In mice, epidermal deletion of JunB affects skin, bone, and the hematopoietic system (Meixner et al, 2008; Uluckan et al, 2016) and combined epidermal deletion of c-Jun and JunB in adult mice results in a skin phenotype resembling psoriasis (Zenz et al, 2005). Previous studies have also shown a role for c-Jun (Riera-Sans & Behrens, 2007) and JunB (Yamazaki et al, 2017) in immune cell differentiation and production of a variety of cytokines in response to TLR stimulation (Vanden Bush & Bishop, 2008; Lin et al, 2011), which activates AP-1 proteins through the TRAF-6/TAK-1/JNK-p38 signaling axis (O'Neill et al, 2013).

Given the importance of Jun proteins in skin homeostasis and their activation downstream of TLRs, we set out to genetically dissect the role of c-Jun downstream of TLR7 signaling in different skin cells, immune cells, and DCs/pDCs, specifically, as these innate immune cells are crucial TLR7-responsive immune potentiators. We show that deletion of c-Jun in DCs results in an attenuated IMQ-induced skin inflammation. Mechanistically, c-Jun/AP-1 exerts these effects by directly controlling CCL2 and IL-23 expression in TLR7 activated DCs. Our findings have clinical implications for patients with cutaneous diseases such as psoriasis.

# Results

### IMQ-induced skin inflammation requires c-Jun in dendritic cells

To dissect the role of c-Jun/AP-1 in skin inflammation, we genetically inactivated c-Jun using *CD11c*-Cre and *K5*-Cre-ER$^{T2}$ mice, which target predominantly DCs or keratinocytes, respectively. Additionally, for a broader deletion of c-Jun, in particular in all hematopoietic cells, we employed the inducible *Mx1*-Cre line, where poly I:C treatment induces interferons to activate the Mx-1 promoter. The treatment scheme for c-Jun deletion in these mouse models and induction of skin inflammation by Imiquimod (IMQ) is illustrated in (Fig EV1A). Epidermal thickness was used as an initial read-out for skin inflammation.

*c-Jun*$^{\Delta/\Delta}$ *Mx1*-Cre mice had significantly less IMQ-induced skin inflammation compared to poly I:C-treated *c-Jun*$^{fl/fl}$ controls (Figs 1A and EV1B). This attenuated inflammatory response to IMQ was recapitulated by DC-specific deletion of c-Jun (*c-Jun*$^{\Delta/\Delta}$ *CD11c*-Cre) (Figs 1B and EV1C), while the keratinocyte-specific c-Jun deletion (*c-Jun*$^{\Delta/\Delta}$ *K5*-Cre-ER$^{T2}$ mice) responded to IMQ similar to Tamoxifen treated *c-Jun*$^{fl/fl}$ controls (Figs 1C and EV1D). Given that c-Jun seems to be required downstream of TLR7 signaling to mediate the inflammatory response in DCs, we focused our *in vivo* analyses on *c-Jun*$^{\Delta/\Delta}$ *CD11c*-Cre mice.

To better understand the role of c-Jun in the onset of skin inflammation, we analyzed the hallmarks of IMQ-induced skin inflammation, which are acanthosis (epidermal thickening), loss of barrier integrity, and infiltration of immune cells such as neutrophils and dermal γδ T cells, the major IL-17-producing immune cell in murine skin. *c-Jun*$^{\Delta/\Delta}$ *CD11c*-Cre mice showed a marked reduction

in acanthosis and were protected from IMQ-induced epidermal barrier breakdown as measured by trans-epithelial water loss (TEWL) (Fig 1D and E). In addition, we observed reduced keratinocyte proliferation (less Ki67$^+$ nuclei, less proliferative K5$^+$ to differentiated K10$^+$ cells) in *c-Jun*$^{\Delta/\Delta}$*CD11c*-Cre mice compared to control mice (Fig EV1E and F and Appendix Fig S1A and B). Infiltration of dermal γδ T cells (γδ TCR$^{int+}$), monocytes (CD11b$^+$Ly6C$^{hi}$), and neutrophils (CD11b$^+$Ly6G$^+$) following IMQ treatment was also significantly decreased in *c-Jun*$^{\Delta/\Delta}$*CD11c*-Cre mice (Fig 1F–H), while levels of the two major dermal DC subsets, CD11b$^+$ DCs and CD103$^+$ DCs, and of epidermal Langerhans cells remained unchanged (Fig EV1G–I). These results suggest that c-Jun does not directly control skin DC development, but rather their effector function downstream of TLR7 signaling, which is required for IMQ-induced skin inflammation.

### CCL2-mediated recruitment of pDCs to IMQ-inflamed skin depends on c-Jun in DCs

We next investigated which specific DC subset is affected by the absence of c-Jun. pDCs express TLR7 and have an essential role in psoriasis pathogenesis (Nestle et al, 2005). They are absent from healthy skin (Gilliet et al, 2004), but are recruited to inflamed skin by a mechanism involving the chemokine CCL2 (Drobits et al, 2012). Thus, we next analyzed specifically pDC recruitment to IMQ-inflamed skin of *c-Jun*$^{\Delta/\Delta}$*CD11c*-Cre mice. The number of pDCs was significantly reduced in *c-Jun*$^{\Delta/\Delta}$*CD11c*-Cre mice after IMQ treatment (Fig 2A–C), accompanied by a reduction in CCL2 skin protein expression (Fig 2D). DCs produce CCL2 upon TLR ligation (Mitchell & Olive, 2010), and c-Jun is a critical regulator of CCL2 expression in cultured fibroblasts (Wolter et al, 2008). We therefore examined whether c-Jun regulates CCL2 expression also in DCs downstream of TLR7 activation by analyzing TLR7 stimulated bone marrow-derived dendritic cells (BMDCs), and DCs sorted from IMQ-treated back skin. We used bone marrow (BM) from *c-Jun*$^{\Delta/\Delta}$ *Mx1*-Cre mice to generate c-Jun deficient BMDCs in all experiments, because of the superior deletion efficiency in the bone marrow precursors compared to *CD11c*-Cre. As shown in Fig 2 E and F, CCL2 expression in both of these DC populations requires c-Jun and is controlled by the TLR7/JNK signaling pathway (Fig 2G–I). To demonstrate the importance of CCL2 for pDC migration into the skin, we injected recombinant CCL2 (rCCL2) into mouse back skin. Injection of rCCL2 induced migration of pDCs into the skin and completely normalized skin pDC recruitment in IMQ-treated *c-Jun*$^{\Delta/\Delta}$*CD11c*-Cre mice (Fig 2J). Collectively, these data demonstrate the importance of c-Jun expression in DCs for the recruitment of pDCs to IMQ-inflamed skin through expression of the c-Jun/AP-1 target gene CCL2.

### c-Jun directly regulates IL-23 expression in DCs

While CCL2 was critical for pDC recruitment following IMQ treatment, it was, in contrast to deletion of c-Jun in DCs, dispensable for the IMQ-induced skin inflammation (Appendix Fig S2A–E), suggesting that other factors regulated by c-Jun are involved. Thus, we next examined the additional functions of c-Jun in DCs besides the control of CCL2 expression. In DCs sorted from IMQ-treated back

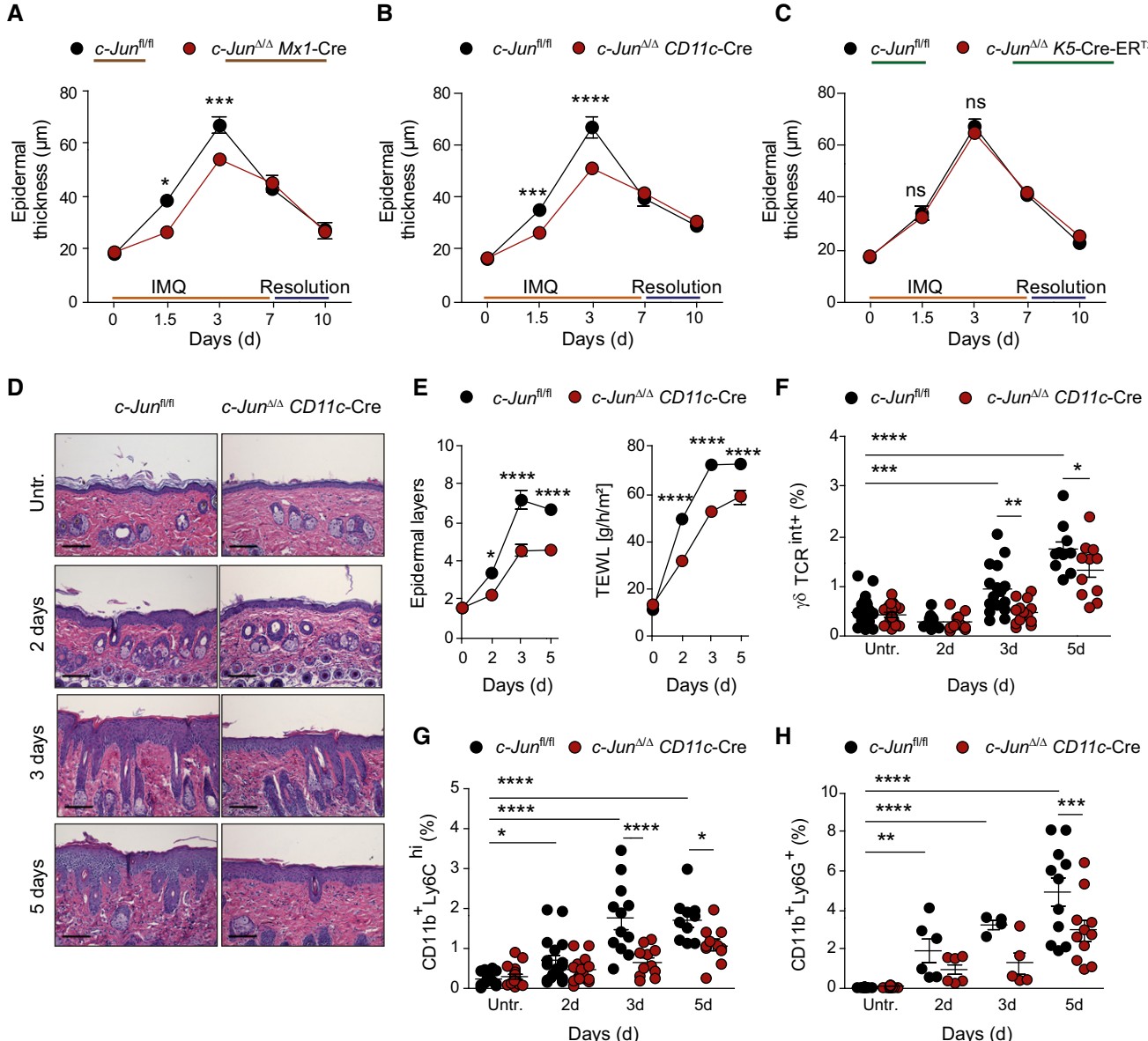

**Figure 1. IMQ-induced skin inflammation requires c-Jun in dendritic cells.**

A–C  Epidermal thickness of the back skin was analyzed at the indicated time points in $c$-$Jun^{\Delta/\Delta}$ $Mx1$-Cre (A), $c$-$Jun^{\Delta/\Delta}$ $CD11c$-Cre (B) and $c$-$Jun^{\Delta/\Delta}$ $K5$-Cre-ER$^{T2}$ mice (C) on hematoxylin and eosin (H&E) stained sections ($n$ = 5–21 (A), $n$ = 6–29 (B), $n$ = 5–18 (C); ≥ 2 independent experiments).

D  H&E stained sections of back skin from indicated mice treated with IMQ for 2, 3, or 5 days. Bright-field images, Magnification: 20×, Scale bar: 100 μm.

E  Layers of epidermis (left) and Trans-epidermal water loss (TEWL) (right) were analyzed in the back skin of indicated mice (Left: $n$ = 8–14, right: $n$ = 6–14; ≥ 2 independent experiments).

F–H  Flow cytometry of total back skin after 2, 3, and 5 days of IMQ treatment. Analyzed were dermal γδ T cells (γδ TCR$^{int+}$) (F), monocytes (CD11b$^+$Ly6C$^{hi}$) (G), and neutrophils (CD11b$^+$Ly6G$^+$) (H). Graphs show immune cells as % of live, single cells ($n$ = 10–28 (F), $n$ = 10–21 (G), $n$ = 4–25 (H); ≥ 2 independent experiments).

Data information: Data are shown as mean ± SEM. $P$-values were calculated by one-way ANOVA with Tukey (A-C, and E) or Bonferroni multiple comparison test (F- H). Statistical significance: ns > 0.05, *$P$ < 0.05, **$P$ < 0.01, ***$P$ < 0.001, ****$P$ < 0.0001. See Appendix Table S3 for exact $P$-values. Source data are available online for this figure.

skin, *c-Jun* mRNA was the most strongly up-regulated among the AP-1 proteins (Fig 3A). Moreover, c-Jun protein was highly expressed in skin dendritic cells of IMQ-treated $c$-$Jun^{fl/fl}$ mice, whereas it was absent in $c$-$Jun^{\Delta/\Delta}CD11c$-Cre mice (Figs 3B and EV2A). Consistently, c-Jun in DCs was necessary for the induction

of various pro-inflammatory cytokines and key mediators of IMQ-induced skin inflammation at specific time points. While only *Il23p19* and *Il17a* were significantly reduced 32h after IMQ treatment in $c$-$Jun^{\Delta/\Delta}CD11c$-Cre skin (Fig 3C), after 48h there was additionally a marked reduction in mRNA expression of a range of

cytokines and chemokines commonly up-regulated after IMQ application in the skin (van der Fits *et al,* 2009; Van Belle *et al,* 2012), including *Il22*, the chemokines for lymphocytes (*Ccl20*) and neutrophils (*Cxcl1*), antimicrobial proteins (*S100a8/9*), and pro-inflammatory cytokines, like *Tnfa* and *Il-6* (Fig EV2B). Accordingly, compared to controls, IL-23 and IL-17A levels were not increased by IMQ treatment in the skin of *c-Jun* $^{\Delta/\Delta}$ *CD11c*-Cre mice (Fig 3D). These results demonstrate that c-Jun in DCs is necessary for the induction of various pro-inflammatory cytokines that mediate the onset of IMQ-induced skin inflammation.

We next identified direct transcriptional targets of c-Jun in IMQ-stimulated BMDCs among the cutaneous, inflammatory mediators that were reduced in the absence of c-Jun (Figs 3C and EV2B). We found c-Jun to be a positive regulator for mRNA expression of the cytokines *Il23p19*, *Il22*, *Il6,* and the chemokines *Ccl20* and *Cxcl1* in BMDCs (Fig EV2C). Since IL-23 is a major driver of IMQ-induced skin inflammation and is predominantly produced by CD11c$^+$ DCs (Wohn *et al,* 2013; Riol-Blanco *et al,* 2014), we considered that IL-23 may represent the critical, direct transcriptional target of c-Jun/AP-1 essential for the onset of IMQ-induced skin inflammation.

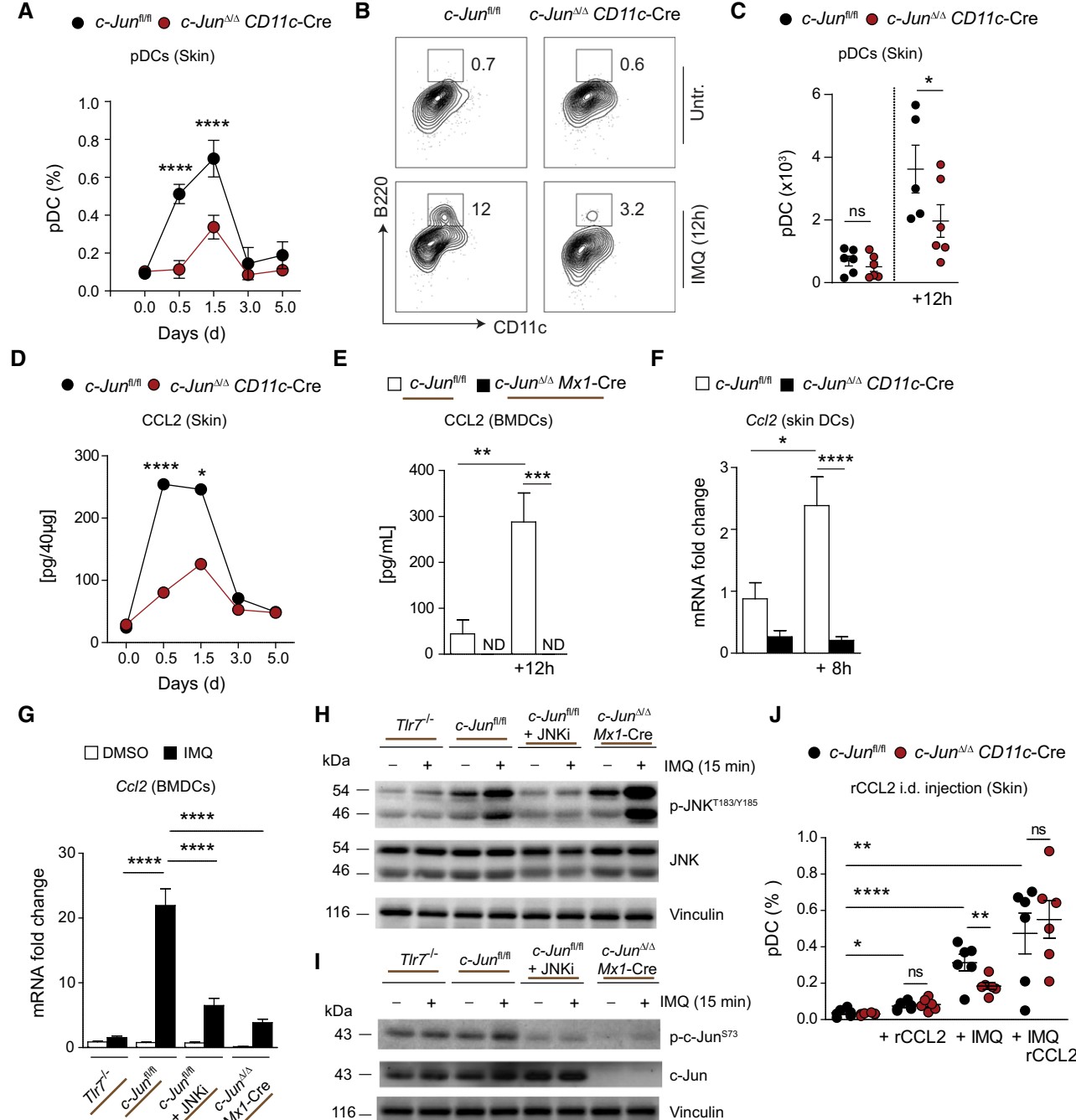

**Figure 2.**

**Figure 2. CCL2-mediated recruitment of pDCs to IMQ-inflamed skin depends on c-Jun in DCs.**

A   Skin infiltration of pDCs (BST-2$^+$ B220$^+$ CD11c$^{int}$ CD11b$^-$) was analyzed by flow cytometry. Graph shows immune cells as % of live, single cells ($n$ = 2–15; 1–3 independent experiments).

B   Representative flow cytometry plots of skin pDCs 12 h after IMQ treatment.

C   Total cell number of skin pDCs defined as in (B) is shown ($n$ = 5–6: 2 independent experiments)

D   ELISA for CCL2 performed on total back skin treated with IMQ ($n$ = 5–14; 2 independent experiments).

E   BMDCs generated from $c\text{-}Jun^{fl/fl}$ or $c\text{-}Jun^{\Delta/\Delta}$ Mx1-Cre BM were stimulated with IMQ for 12 h. CCL2 protein was quantified by ELISA ($n$ = 7–10; 3 independent experiments).

F   qRT–PCR detection of $Ccl2$ mRNA in CD11c$^+$MHCII$^+$ cells sorted from IMQ-treated back skin (8 h) ($n$ = 5–12; 3 experiments).

G   BMDCs were generated from $Tlr7^{-/-}$, $c\text{-}Jun^{fl/fl}$, or $c\text{-}Jun^{\Delta/\Delta}$ Mx1-Cre BM and stimulated with IMQ for 4 h and/or pretreated with SP600125 (JNKi, 25 μM) for 1 h. $Ccl2$ mRNA was quantified by qRT–PCR ($n$ = 4–18; 2–4 experiments).

H, I   Western blot analysis of Phospho- or total JNK and c-Jun in BMDCs of indicated genotype pretreated with DMSO (1: 1,000) or SP600125 (JNKi, 25 μM) for 1 h and stimulated with IMQ (5 μg/ml) for 15 min.

J   Flow cytometry of back skin 12 h after intradermal (i.d.) injection of 1 μg rCCL2 and/or IMQ treatment. Analyzed were pDCs (BST-2$^+$B220$^+$CD11c$^{int}$CD11b$^-$) shown as % of live, single cells ($n$ = 6; 2 independent experiments).

Data information: Data are shown as mean ± SEM. $P$-values were calculated by multiple $t$-test with the Holm–Šídák method (C) or one-way ANOVA with Bonferroni (A), Kruskal–Wallis (E) or Tukey (D, F, G, J) multiple comparison test. Statistical significance: ns > 0.05, *$P$ < 0.05, **$P$ < 0.01, ***$P$ < 0.001, ****$P$ < 0.0001. See Appendix Table S3 for exact $P$-values.

Source data are available online for this figure.

Closer investigations revealed that IL-23 expression and protein levels in BMDCs are regulated by TLR7/JNK/c-Jun signaling as measured by qRT–PCR (Fig 3E), by ELISA and intracellular flow cytometry (Fig 3F), and by immune-fluorescence (Fig EV2D). In addition, AP-1 inhibition with the small molecule, T-5224, completely abolished IL-23 and reduced CCL2 expression in IMQ-stimulated BMDCs (Appendix Fig S3A–D). Deletion of c-Jun also resulted in a significant reduction of $Il23p19$ mRNA expression in DCs sorted from IMQ-treated back skin (Fig 3G). Furthermore, we sorted non-immune (CD45$^-$), myeloid (Gr-1$^+$), and T (CD-3ε$^+$) cells and confirmed DCs as the most important source of $Il23p19$ mRNA after IMQ application (Fig EV2E and F). In contrast, IL-12p40, the second subunit of the hetero-dimeric cytokine IL-23, remained unchanged in the absence of c-Jun (Fig EV2G and H). Inhibition of NF-κB, which is known to regulate IL-23 expression in DCs (Carmody $et$ $al$, 2007; Zhu $et$ $al$, 2017), led to a reduction of $Il23p19$ mRNA similar to c-Jun deletion (~50%) (Fig EV2I), suggesting that both transcription factors contribute to IL-23 regulation.

Analysis of the IL-23p19 promoter sequence revealed several c-Jun binding sites and chromatin immune-precipitation assay demonstrated a significant increase in c-Jun binding to the proximal promoter of IL-23p19 in $c\text{-}Jun^{fl/fl}$ DCs after IMQ treatment compared to c-Jun-deficient DCs (Fig 3H and I). A luciferase reporter assay further confirmed the critical role of c-Jun/AP-1 for IL-23 expression. Blockade of JNK signaling or mutation of the putative AP-1 binding site significantly reduced IL-23p19 promoter activity in IMQ-stimulated, transfected RAW 264.7 cells (Fig 3J).

IL-23 induces IL-17A production in γδ-T cells, which contributes to IMQ-induced inflammation (Tortola $et$ $al$, 2012). Consistently, analysis of dermal γδ-T cells of $c\text{-}Jun^{\Delta/\Delta}CD11c$-Cre mice treated with IMQ revealed reduced IL-17A production compared to control mice (Fig 3K). Next, we injected recombinant IL-23 (rIL-23) in the back skin of mice treated with IMQ. Importantly, rIL-23 restored the response to IMQ in $c\text{-}Jun^{\Delta/\Delta}CD11c$-Cre mice (Fig 3L). However, rIL-23 injection was not sufficient to induce CCL2-dependent pDC migration to the skin (Appendix Fig S4A). Consistently, rIL-23 treatment only weakly induced $Ccl2$ mRNA expression, whereas $Il17a$ mRNA was induced strongly (Appendix Fig S4B). In addition, although BM-pDCs expressed

mRNA for the IL-12 receptor subunits $Il\text{-}12rb1$ and $Il\text{-}12rb2$ and could be stimulated with rIL-12, they lacked expression of the IL-23 receptor-specific subunit $Il\text{-}23r$, and rIL-23 failed to induce $Gzmb$ and $Trail$ mRNA expression (Appendix Fig S4C and D). To exclude that the observed effects were occurring only in response to IMQ, we investigated whether c-Jun is critical downstream of TLR7 signaling after stimulation with the non-Imidazoquinoline, adenine analog compounds, CL264 and CL307. IL-23, as well as CCL2 protein expression, were significantly reduced in c-Jun-deficient BMDCs stimulated with CL264 and CL307 (Appendix Fig S5A–D). Taken together, these results demonstrate that the TLR7/JNK/c-Jun signaling axis controls IL-23 production in DCs, providing a molecular explanation for the attenuated skin inflammation observed in $c\text{-}Jun^{\Delta/\Delta}CD11c$-Cre mice.

### c-Jun is essential in conventional type-2 DCs to control CCL2 and IL-23 expression

The skin harbors different DC subpopulations, also known as conventional type-1 and type-2 DCs with unique functional properties (Haniffa $et$ $al$, 2015). While cDC1 promote Th$_1$ immunity, cDC2 drive Th$_2$ and Th$_{17}$ responses (Collin & Bigley, 2018). We next asked whether c-Jun promotes CCL2 and IL-23 expression in a distinct cutaneous DC subset. To specify the relevant DC subpopulation and to investigate, whether macrophages contribute to the observed phenotypes, we sorted cDC1, cDC2, Langerhans cells, and macrophages from murine wild-type skin. IMQ-induced $c\text{-}Jun$, $Ccl2$, and $Il\text{-}23p19$ expression was most prominent in the cDC2 subset, but $Ccl2$ was also induced, to a lesser extent, in cDC1, LCs, and macrophages. All three genes were poorly induced in macrophages after IMQ treatment (Fig 4A). However, macrophages had the highest basal gene expression of $Ccl2$ (Appendix Fig S6A and B). When compared to $c\text{-}Jun^{fl/fl}$ mice, sorted cDC2 subsets from $c\text{-}Jun^{\Delta/\Delta}CD11c$-Cre mice expressed significantly reduced levels of $Ccl2$ and $Il23p19$, whereas the expression of these genes was not affected in sorted macrophages (Fig 4B and C). Moreover, $Ccl2$ and $Il23p19$ were also expressed at similar levels in CD11c expressing macrophages sorted from $c\text{-}Jun^{fl/fl}$ and $c\text{-}Jun^{\Delta/\Delta}CD11c$-Cre mice confirming that these macrophage subsets was not contributing to the observed

phenotype. In fact, c-Jun expression was also not affected in CD11c expressing macrophages from *c-Jun*^Δ/Δ*CD11c*-Cre mice as shown by qRT–PCR, demonstrating that c-Jun deletion by *CD11c*-Cre had not

occurred in these cells (Fig 4D and E). These results demonstrate that the phenotype observed is most likely due to c-Jun activity in the cDC2 subset.

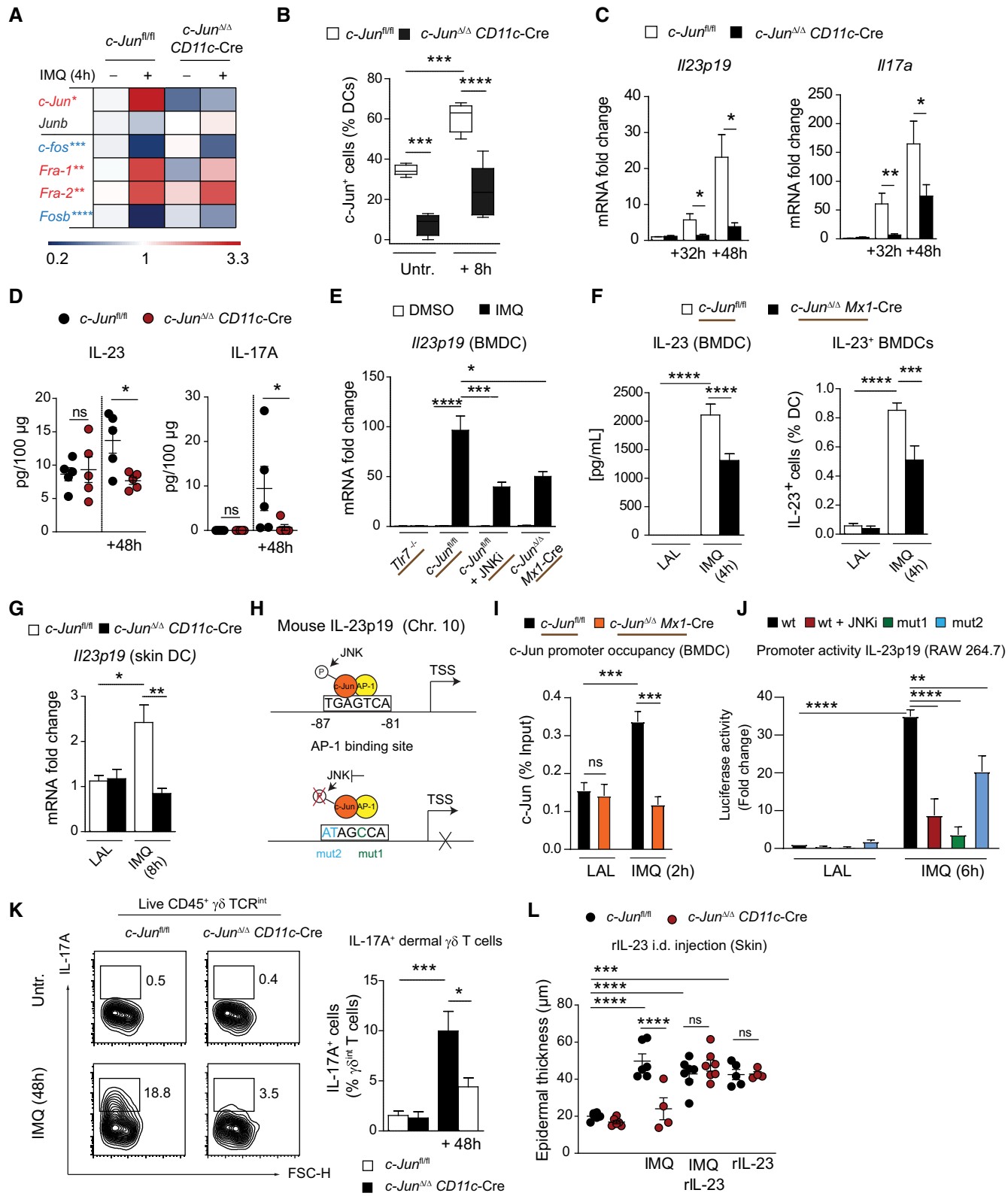

**Figure 3.**

**Figure 3.   c-Jun directly regulates IL-23 expression in DCs.**

A   Heat map visualization of AP-1 family member expression in DCs sorted from back skin of indicated mice after 4 h IMQ stimulation as quantified by qRT–PCR. Color code (red = up, blue = down-regulation) shows fold change of *Ap-1* mRNA to untreated *c-Jun*$^{fl/fl}$ DC control. Asterisk shows significant differences in IMQ activated to control *c-Jun*$^{fl/fl}$ DCs (n = 4–14; 2–4 independent experiments).

B   Graph shows the percentage of c-Jun positive skin DCs quantified by immunofluorescence performed as described in Fig EV2 (A) (n = 5, 3 experiments).

C   qRT–PCR detection of *Il23p19* and *Il17a* in total RNA isolated from the back skin of indicated mice treated with IMQ for 32 or 48 h (n = 10–21; 3–6 experiments).

D   IL-23 and IL-17A protein was quantified by Luminex multiplex assay in cutaneous lysates of indicated mice after IMQ treatment (48 h) (n = 5; 2 independent experiments).

E   qRT–PCR detection of *Il23p19* mRNA expression levels in BMDCs of indicated genotype stimulated with IMQ for 4 h and/or pretreated with JNKi (SP600125, 25 μM, 1 h) (n = 4–14; 2–5 independent experiments).

F   IL-23 protein was quantified by ELISA (1 × 10⁶ BMDCs/well) and by intracellular flow cytometry (IL-23p19$^+$p40$^+$ cells among live, single, CD45$^+$, CD11c$^+$ cells) in *c-Jun*$^{fl/fl}$ and *c-Jun*$^{Δ/Δ}$*Mx1*-Cre BMDCs stimulated with IMQ (Left: n = 10–13; Right: n = 9–10; ≥ 3 independent experiments).

G   qRT–PCR detection of *Il23p19* mRNA expression levels in CD11c$^+$MHCII$^+$ cells sorted from IMQ (8 h)-treated back skin of indicated mice (n = 5–12; 3 experiments).

H   Genomic location and sequence of a putative AP-1 binding site in the mouse promoter of IL-23p19. Blockade of c-Jun/AP-1 TF binding by inhibition of phosphorylation (JNKi, red), or mutation of the binding sequence (mut 1 = T -> C; green | mut 2 = TG -> AT; blue) reduces IL-23p19 promoter activity (see Fig 4J).

I   c-Jun chromatin immunoprecipitation of *c-Jun*$^{fl/fl}$ and *c-Jun*$^{Δ/Δ}$*Mx1*-Cre BMDCs stimulated with IMQ for 2 h (n = 4–6; 3 independent experiments).

J   Luciferase activity was quantified in lysates of RAW 264.7 cells transfected with pGL3 basic vector harboring wild-type (wt) or mutated (mut1 or 2) IL-23p19 promoter. Transfected cells were pretreated with JNKi (SP600125, 25 μM, 1 h) and stimulated with IMQ (6 h). Results are shown as fold change to pGL-3-wt transfected, LAL (Limulus amebocyte lysate)-treated RAW264.7 cells (n = 4; 2 independent experiments).

K   IL-17A production by dermal γδ T cells was analyzed in IMQ-treated skin (Day 2) of indicated mice by intracellular flow cytometry. Shown are representative plots (left) and a bar graph (right) of IL-17A$^+$ cells pregated on live, single, CD45$^+$, γδ TCR$^{int+}$ cells (n = 7–10; 3 independent experiments).

L   Epidermal thickness of back skin was quantified in H&E stained sections of indicated mice treated with rIL-23 (i.d., 1 μg) and/or IMQ for two consecutive days (n = 4–7; 3 independent experiments).

Data information: Data are shown as mean ± SEM. *P*-values were calculated by unpaired, two-tailed *t*-test (A, C) and one-way ANOVA with Tukey multiple comparison test (B, D-G, I-L). Statistical significance: ns > 0.05, *P < 0.05, **P < 0.01, ***P < 0.001, ****P < 0.0001. See Appendix Table S3 for exact *P*-values.
Source data are available online for this figure.

## Blocking JNK/c-Jun signaling ameliorates IMQ-induced skin inflammation

Next, we explored the therapeutic potential of blocking the TLR7/JNK/c-Jun signaling cascade in the IMQ-induced psoriasis-like skin inflammation model. *c-Jun*$^{fl/fl}$ mice were treated daily with IMQ and the JNK inhibitor (JNKi), SP600125. *Tlr7*$^{-/-}$, *c-Jun*$^{Δ/Δ}$ *CD11c*-Cre, and *Tlr7*$^{-/-}$ - *c-Jun*$^{Δ/Δ}$ *CD11c*-Cre mice were used as controls (Fig 5A). JNKi treatment ameliorated IMQ-induced skin thickening, trans-epidermal water loss (TEWL), keratinocyte proliferation, and differentiation comparable to deletion of c-Jun in DCs in the *c-Jun*$^{Δ/Δ}$*CD11c*-Cre mouse model. In addition, JNKi treatment of *Tlr7*$^{-/-}$ and *c-Jun*$^{Δ/Δ}$ *CD11c*-Cre mice showed that these effects were restricted to the TLR7/c-Jun signaling pathway (Figs 5B–E and EV3A–C). t-distributed stochastic neighbor embedding (t-SNE) —analyzes of the cutaneous immune cell populations revealed an IMQ-induced inflammatory skin phenotype in *c-Jun*$^{fl/fl}$ mice, which was absent in JNKi treated mice (Fig EV3D). Conventional FACS analysis confirmed the immune populations defined by t-SNE (Appendix Fig S7A and B) and showed a significantly reduced frequency of monocytes, neutrophils, and γδ-T cells in JNKi treated, as well as, *c-Jun*$^{Δ/Δ}$*CD11c*-Cre and *Tlr7*$^{-/-}$ mice (Figs 5F–H and EV3E and F). Lastly, a protein screen showed that JNKi treatment significantly reduced the pro-inflammatory cytokines IL-33 and IL-17A and the chemokines CCL2 and CXCL1 on therapy endpoint (5d IMQ) compared to the vehicle-treated control (Appendix Fig S7C).

We tested the therapeutic effects of JNKi in a second independent mouse model of psoriasis based on genetic, inducible and epidermal-specific deletion of *c-Jun* and *JunB* (*c-Jun/JunB*$^{Δ/Δ}$ *K5-Cre-ER*$^{T2}$ mice) (Zenz *et al*, 2005). Inhibition of JNK (Fig 5I) ameliorated psoriatic disease in 6 out of 8 mice, whereas disease progressed in 5 out of 7 control mice (Fig 5J–L) with reduced infiltration of monocytes and γδ-T cells (Fig 5M and N). As we demonstrated IL-23 signaling to be the crucial pathway activated by JNK/c-Jun, we next compared the efficiency of JNK inhibition to IL-23R blockade. Treatment with anti-IL-23R antibody alleviated IMQ-induced skin inflammation as effectively as JNKi treatment (Fig EV3G–I). These data show that pharmacological inhibition of JNK aimed to disrupt the pathogenic TLR7/JNK/c-Jun/IL-23 signaling axis ameliorates skin inflammation in psoriasis-like mouse models.

## c-Jun, CCL2, and IL-23 are co-expressed in type-2/inflammatory DCs of psoriatic lesions

Next, we investigated the human relevance of our findings by analyzing the expression of c-Jun in DCs of psoriatic lesions of patients. Immunofluorescence microscopy revealed co-localization of c-Jun protein with CD11c$^+$ DCs in lesional dermis. In contrast, in non-lesional skin, expression of c-Jun in DCs was weak or absent (Fig EV4A).

To strengthen this finding, we next assessed the expression of different DC markers in a published RNA-Seq Dataset generated from human psoriatic skin tissue (Data ref: Tsoi *et al*, 2019b). We found markers for cDC2 (*Cd1c*) and inflammatory DCs (*Cd14*) to be increased in lesional psoriasis (Fig 6A), as well as an increase in expression for *Tlr7/8*, *Jnk1*, *c-Jun*, *Ccl2*, and *Il23p19*. Consistently, CCL2 and IL-23p19 protein was increased in lesional compared to healthy skin. Lastly, *c-Jun* expression strongly correlated with *Ccl2* expression in lesional, but not in non-lesional or healthy skin (Fig EV4B–D). Therefore, we analyzed c-Jun expression in DCs with CD1a, CD1c, or CD14 markers in healthy, non-lesional, and psoriatic skin. c-Jun was expressed equally in CD1a, CD1c, and CD14 expressing cells (Fig 6B) and increased in lesional compared to healthy and non-lesional skin (Fig 6C). Moreover, co-expression of

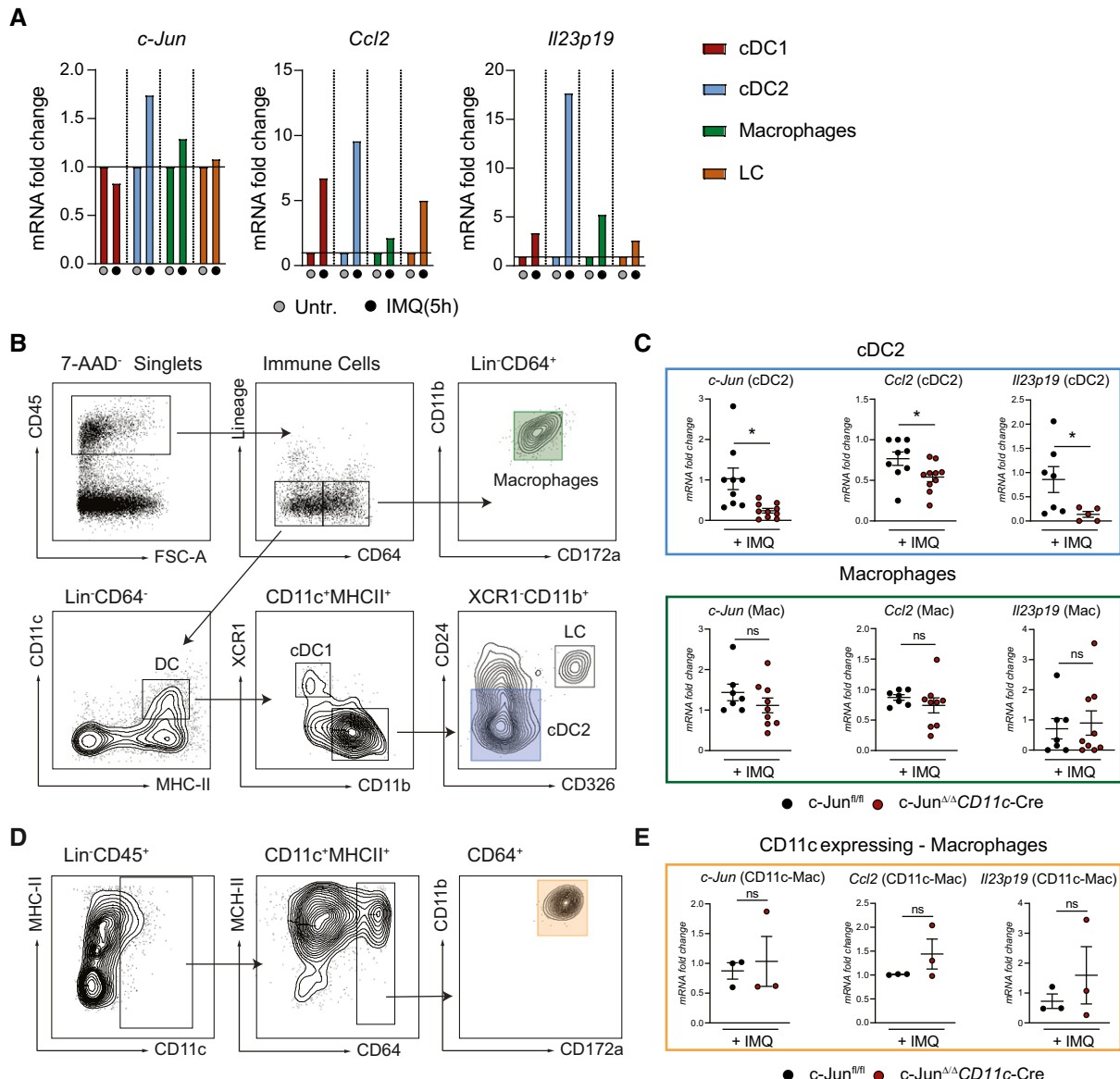

**Figure 4. c-Jun is essential in conventional type-2 DCs to control CCL2 and IL-23 expression.**

A  DC subsets (cDC1, cDC2), macrophages (MP), and Langerhans cells (LC) were sorted from IMQ-treated back skin (5 h). *c-Jun*, *Ccl2*, and *Il23p19* mRNA was analyzed by qRT–PCR. Sort strategy is shown in (B) (10 mice were pooled per condition, 1 experiment).

B  Sort strategy for the separation of MPs (CD45⁺Lin⁻CD64⁺CD172a⁺CD11b⁺), and the DC (CD45⁺Lin⁻CD64⁻CD11c⁺MHCII⁺) subsets cDC1 (XCR1⁺CD11b⁻ DC), cDC2 (XCR1⁻CD11b⁺CD24⁻/intCD326⁻) and LC (XCR1⁻CD11b⁺CD24⁺CD326⁺) in IMQ-treated (5 h) murine back skin.

C  qRT–PCR detection of *c-Jun*, *Ccl2*, and *Il23p19* in MPs and cDC2 sorted as described in (B) from indicated mice (MPs: n = 7–9; 3 independent experiments, cDC2: n = 5–10; 2–3 independent experiments and back skin from 2 mice was pooled).

D  Sort strategy for CD11c expressing MPs (CD45⁺Lin⁻CD11c^int/lowMHCII⁺CD64⁺CD11b⁺CD172a⁺) is shown. Murine skin was treated for 5 h with IMQ.

E  Expression of indicated targets was analyzed by qRT–PCR in cells sorted as described in (D) (n = 3; Back skin from 2 mice was pooled).

Data information: Data are shown as mean ± SEM. *P*-values were calculated by unpaired, two-tailed *t*-test (C, E). Statistical significance: ns > 0.05, *P < 0.05. See Appendix Table S3 for exact *P*-values.
Source data are available online for this figure.

c-Jun with CCL2 or IL-23 could be detected in CD1a⁺ cells (Fig 6D and E), as well as in CD1c⁺ cells and CD14⁺ cells (Fig EV4E and F). These results suggest that c-Jun expression in type-2/inflammatory DCs may also promote psoriasis in patients, by controlling the expression of CCL2 and IL-23.

**JNK/AP-1 Inhibitors repress CCL2 and IL-23 expression in human mo-DCs**

Next, we generated human monocyte-derived DCs (mo-DCs) to study the importance of Jun/JNK signaling for the expression of

CCL2 and IL-23, the two essential cytokines we identified to be regulated by c-Jun in murine DCs. mo-DCs stimulated with the TLR7/8 agonist Resiquimod (R848) showed phosphorylation of Jun/JNK, which was prevented by addition of the JNK inhibitor SP600125 (Fig 7A). Inhibition of Jun/JNK signaling with JNKi or T-5224 (AP-1 Inhibitor) reduced CCL2 and IL-23 expression in R848 stimulated

mo-DCs (Fig 7B and C), but DC maturation, as measured by CD80 and CD86 up-regulation, was unaffected (Fig EV5A–D). Next, we stimulated human mo-DCs with the psoriasis-promoting LL-37/RNA complex, a disease-relevant TLR7/8 agonist (Ganguly *et al*, 2009). LL-37/RNA complex induced maturation (Fig EV5E and F) and expression of CCL2 and IL-23 in human mo-DCs (Fig EV5G), which

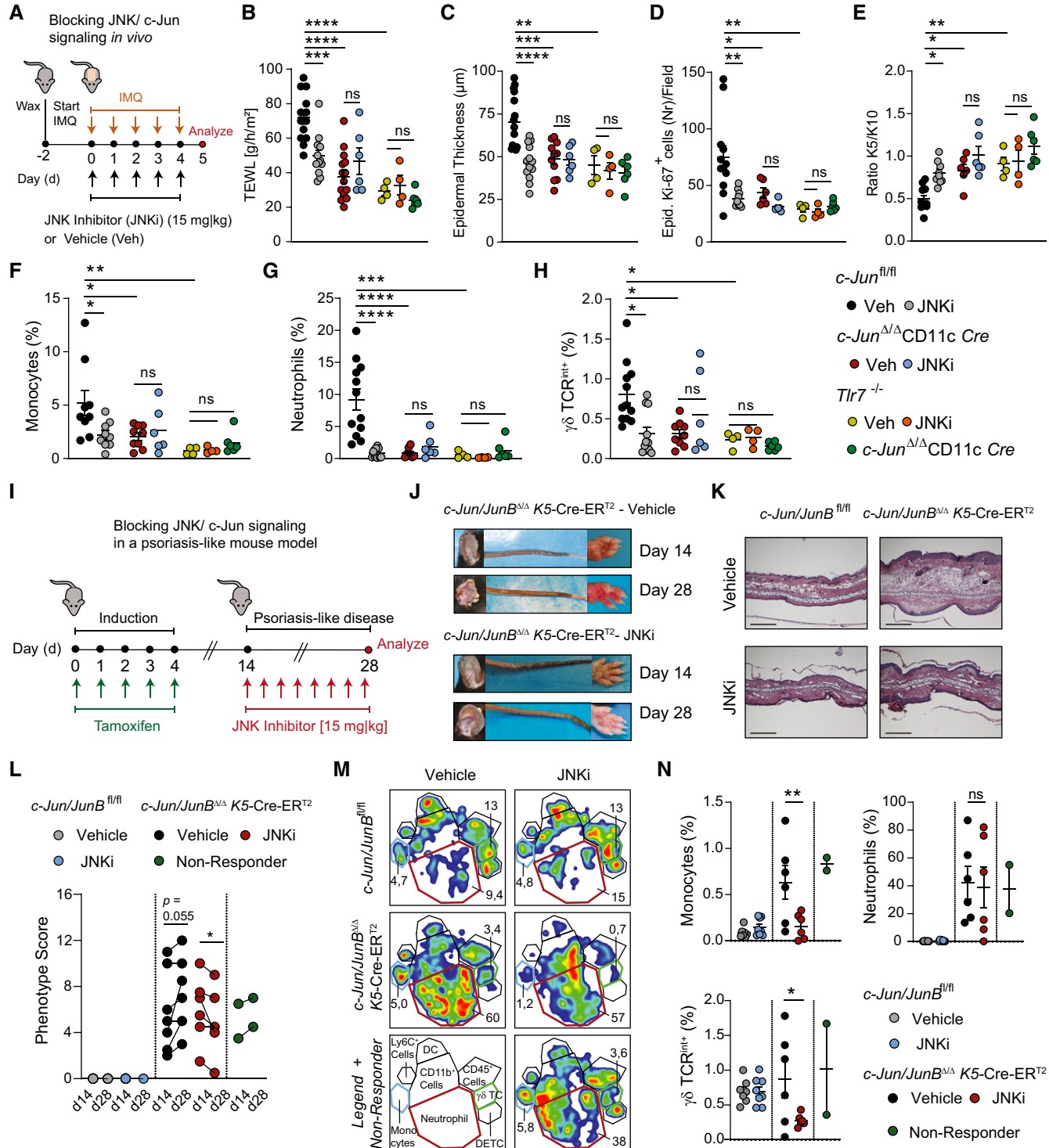

**Figure 5.**

**Figure 5.  Blocking JNK/c-Jun signaling ameliorates psoriasis-like skin inflammation.**

A       Experimental design for blocking JNK/c-Jun signaling in the IMQ-induced skin inflammation model. *c-Jun*^fl/fl^, *c-Jun*^Δ/Δ^*CD11c*-Cre, *Tlr7*^−/−^, and *Tlr7*^−/−^ *c-Jun*^Δ/Δ^*CD11c*-Cre mice were treated with IMQ and JNK inhibitor SP600125 (JNKi, 15 mg/kg, i.p.) or vehicle for 5 consecutive days.

B–E     After 5 days of treatment TEWL was measured on the back skin (B), epidermal thickness was analyzed on H&E stained sections (C), epidermal Ki-67⁺ cells (D) and the ratio of K5 to K10 positive areas (E) were analyzed by immunofluorescence as described in Fig EV3 (B, C) (*n* = 4–15; ≥ 2 independent experiments).

F–H     Quantification of immune cell populations as defined in Fig EV3 (E, F). Results are given as percentage of live, single cells (*n* = 4–12; 2–4 independent experiments).

I       Experimental design for blocking JNK/c-Jun signaling in *c-Jun/JunB* ^Δ/Δ^ *K5*-Cre-ER^T2^ mice. Psoriasis-like disease was induced by injection of Tamoxifen (i.p., 5 days). Treatment with JNK Inhibitor (15 mg/kg, every other day, i.p.) started on day 14 for 2 weeks.

J       Representative images of ear, tail and paw of *c-Jun/JunB* ^Δ/Δ^ *K5*-Cre-ER^T2^ at the start (Day 14) and end (Day 28) of treatment with JNK Inhibitor or vehicle.

K       H&E stained sections of mouse ear from indicated mice. Bright-field images, Scale bar: 250 µm, Magnification: 10×.

L       Psoriasis-like phenotype was scored from 0 to 4 for ear, tail, paw and snout. Shown is the cumulative score at therapy start (Day 14) and end (Day 28). Non-responders are *c-Jun/JunB* ^Δ/Δ^ *K5*-Cre-ER^T2^ mice treated with JNK Inhibitor, which showed disease-progression (*n* = 2–9, 2 independent experiments).

M       Representative t-SNE- plots (*n* = 2–3) show the cutaneous immune cell phenotype at therapy end (Day 28). Numbers adjacent to gates give cell populations frequency among live, single, CD45⁺ cells. t-SNE Legend-Plot labels the populations defined by t-SNE- algorithm.

N       Dermal γδ T cells (γδ TCR^int+^), monocytes (CD11b⁺Ly6C^hi^), and neutrophils (CD11b⁺Ly6G⁺) were analyzed by flow cytometry in mouse ear at the end of treatment (Day 28). Graphs show immune cells as % of live, single cells (*n* = 2–8, 2 independent experiments).

Data information: Data are shown as mean ± SEM. *P*-values were calculated by paired, two-tailed *t*-test (L) and one-way ANOVA with Tukey (B-H) or Bonferroni multiple comparison test (N). Statistical significance: ns > 0.05, **P* < 0.05, ***P* < 0.01, ****P* < 0.001, *****P* < 0.0001. See Appendix Table S3 for exact *P*-values.

Source data are available online for this figure.

was inhibited by JNK and AP-1 Inhibitors (Fig 7D–F). These results demonstrate that c-Jun in DCs contributes to psoriatic inflammation by regulating CCL2 and IL-23, two cytokines required for pDC recruitment and activation of IL-17A producing T cells.

# Discussion

Jun/AP-1 proteins are important for immune cell development and function, but their specific role in Toll-like receptor (TLR) signaling is poorly understood. In the present study, we discover a novel function of c-Jun in TLR7 signaling as a critical positive regulator of CCL2 and IL-23 expression in DCs. Deletion of c-Jun in CD11c⁺ DCs attenuates psoriasis-like skin inflammation induced by the TLR7 agonist IMQ. Importantly, in psoriatic skin we show prominent expression of c-Jun in different DCs, suggesting a critical, disease-relevant role in this chronic, auto-immune skin disease.

Both initiation (Nestle *et al*, 2005) and maintenance (Ganguly *et al*, 2009) of psoriatic lesions have been shown to depend on the engagement of TLR7/8 and TLR9 in DCs and pDCs by a complex that consists of self-RNA or DNA bound to an antimicrobial peptide, LL-37, which is overexpressed in psoriatic skin. Consistently, application of the TLR7 agonist IMQ has been shown to trigger a

psoriasis-like disease in humans (Fanti *et al*, 2006) and is used as a skin inflammation model of psoriasis in mice (van der Fits *et al*, 2009). Studies addressing how engagement of this disease-relevant signaling pathway is coupled to cytokine production are, however, missing. Our data link the expression of the psoriasis-driving cytokine IL-23 in TLR7/8 stimulated human monocyte-derived DCs to the JNK/MAPK pathway and activation of c-Jun. This not only provides a better understanding of how TLR7/8 can trigger psoriasis development, but also identifies potential new druggable targets.

Our data provide evidence that c-Jun/AP1 has a multi-faceted, cell-type specific function in the pathogenesis of psoriasis. The Jun/AP-1 family plays an important role in the epidermal compartment for the pathogenesis of psoriasis (Gago-Lopez *et al*, 2019). JunB, located on the PSOR6 locus, is down-regulated in psoriatic epidermis (Zenz *et al*, 2005), whereas c-Jun expression is up-regulated in the basal layer with a possible impact on keratinocyte proliferation (Mehic *et al*, 2005). We now provide evidence that c-Jun/AP-1 is critical, not only in epithelial cells, but also in DCs for the pathogenesis of psoriasis. Only epidermal c-Jun/JunB double mutant, but not the single mutant mice develop a psoriasis-like skin inflammation, indicating a compensatory function of c-Jun and JunB (Zenz *et al*, 2005). Conversely, deletion of c-Jun alone in CD11c⁺ cells reduced the IMQ-induced skin inflammation. Anti-inflammatory effects of

**Figure 6.  c-Jun, CCL2, and IL-23 are co-expressed in type-2/inflammatory DCs in psoriatic lesions.**

A       Gene expression values of *Cd207*, *Cd1a*, *Cd1c*, *Cd14*, *Xcr1* in healthy (*n* = 38), non-lesional (*n* = 27), and lesional (*n* = 28) skin. Expression values were obtained from the GEO Data Set [GSE121212]. Box and whiskers plot: Central band shows median, box extends from the 25^th^ to 75^th^ percentiles, and whiskers go down to the smallest (min) and up to the largest (max) value.

B       Immunofluorescence of c-Jun (green), CD1a or CD1c or CD14 (red) and DAPI in psoriatic lesions. Shown is a representative image (Magnification: 25×, Scale Bar: 100µm) with an Inset (Magnification: 63×, Scale Bar: 20 µm, Deconvoluted). Arrows indicate c-Jun⁺ DCs. Asterisk highlights a DC with prominent c-Jun expression as shown enlarged in 2 white-framed windows (DC Marker (red) + c-Jun (green, left) or DAPI (blue, right)).

C       Quantification of c-Jun⁺ DCs (CD1a, CD1c, or CD14) in psoriatic lesions as shown in (B). Four randomly chosen fields on one section were analyzed for each sample (*n* = 2 patient samples/condition). Box and whiskers plot: Central band shows median, box extends from the 25^th^ to 75^th^ percentiles, and whiskers go down to the smallest (min) and up to the largest (max) value.

D, E    Representative immunofluorescence of c-Jun (green), CD1a (red), CCL2 (D), or IL-23p19 (E) (white) and DAPI in psoriatic lesions. Arrows indicate triple-positive cells and an asterisk highlights a representative one that is shown enlarged in a white-framed inlet. Magnification: 40x. Scale bar: 50 µm (*n* = 2 patient samples).

Data information: Data are shown as mean ± SEM. *P*-values were calculated by one-way ANOVA with Tukey multiple comparison test (A, C). Statistical significance: ns > 0.05, **P* < 0.05, ***P* < 0.01, ****P* < 0.001, *****P* < 0.0001. See Appendix Table S3 for exact *P*-values.

Source data are available online for this figure.

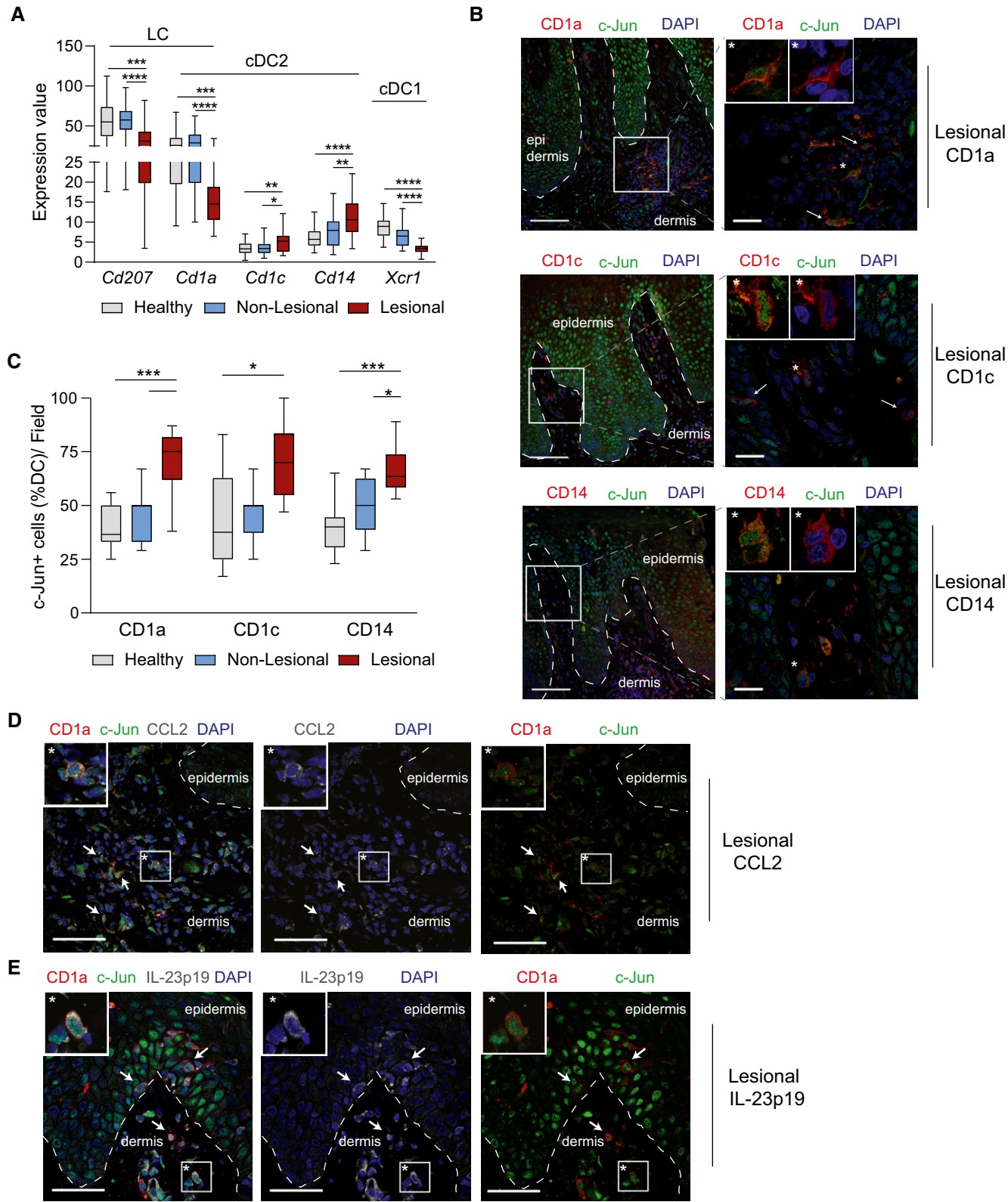

**Figure 6.**

JNK inhibition in immune-mediated diseases have been reported (Han *et al*, 2001; Mitsuyama *et al*, 2006). We now show that JNKi treatment ameliorates disease in the IMQ model and in a second psoriasis model using epidermal deletion of c-Jun/JunB. JNKi treatment did not cause toxic side effects, most likely because the inhibitor targets c-Jun, but not JunB (Kallunki *et al*, 1996). This further

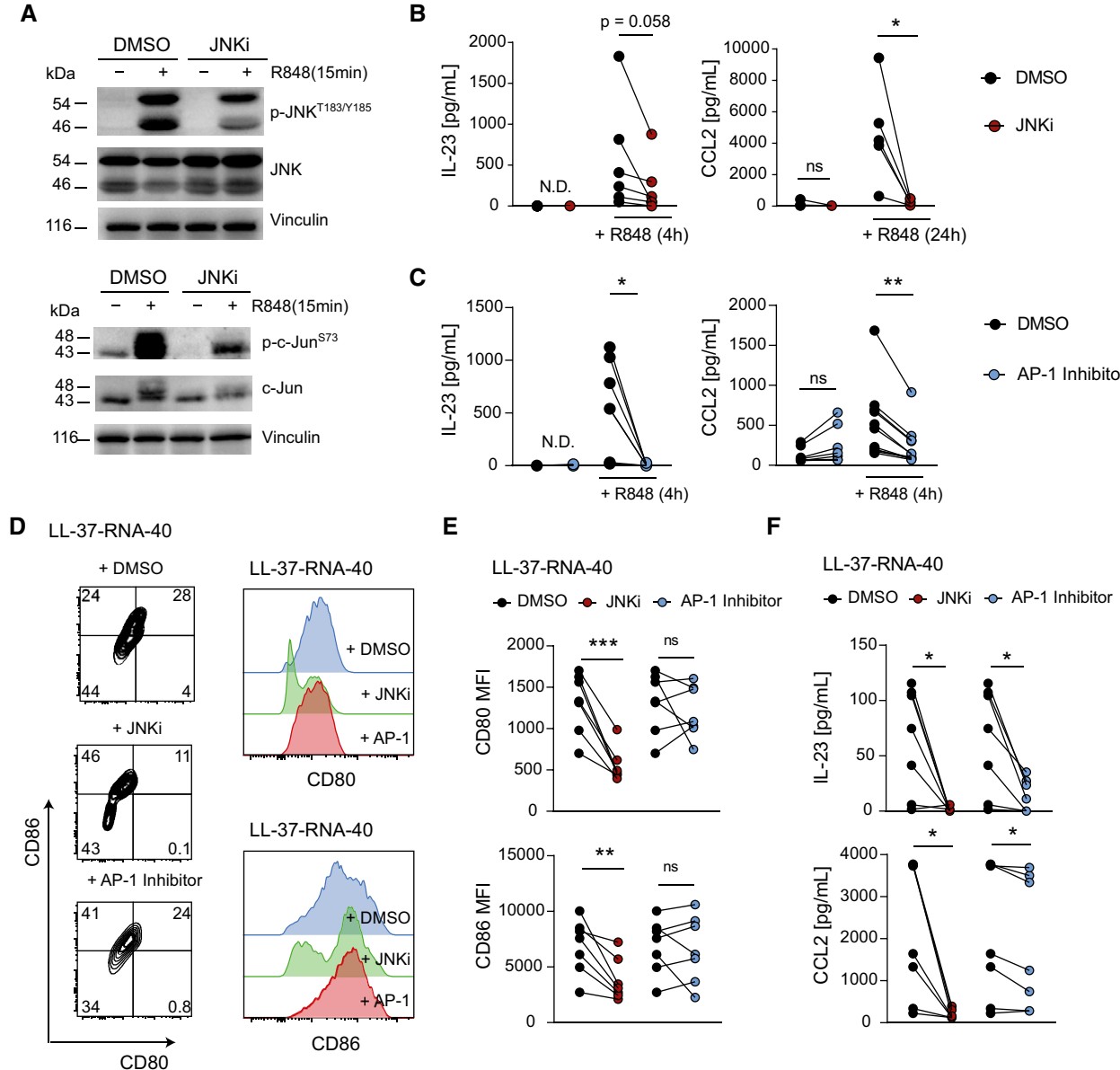

**Figure 7.  JNK/AP-1 Inhibitors repress CCL2 and IL-23 expression in human mo-DCs.**

A    Western blot analyses of Phospho- or total JNK and c-Jun in human mo-DCs pretreated with DMSO (1: 1,000) or SP600125 (JNKi, 25 μM) for 1 h and stimulated with R848 (10 μg/mL) for 15 min.

B, C  Human mo-DCs pretreated with JNKi (SP600125, 25 μM) (B) or AP-1 Inhibitor (T-5224, 20 μM) (C) for 1 h were stimulated with R848 (10 μg/ml). CCL2 and IL-23 were analyzed by ELISA ($n$ = 5–11; 2–3 independent experiments).

D    Representative flow cytometry plots (left) and histograms (right) of human mo-DCs stimulated with LL-37-RNA-40 for 24 h are shown. Inhibitors (JNKi; 25 μM or AP-1 Inhibitor; 20 μM) were given 1 h before stimulation. Plots shown are pregated on live, single, CD1a$^+$ cells.

E, F  Human mo-DCs described in (D) were analyzed for mean fluorescence intensity of CD80 or CD86 by flow cytometry (E) and CCL2 or IL-23 protein by ELISA (F) ($n$ = 7; 2 independent experiments).

Data information: Data are shown as mean $\pm$ SEM. $P$-values were calculated by paired, two-tailed $t$-test (B, C, E and F). Statistical significance: ns > 0.05, *$P$ < 0.05, **$P$ < 0.01, ***$P$ < 0.001. See Appendix Table S3 for exact $P$-values.
Source data are available online for this figure.

supports our hypothesis that c-Jun is essential for TLR7 signaling in DCs, but not in keratinocytes. Thus, disruption of JNK/c-Jun signaling by small molecules might be explored as a treatment option in psoriasis.

DCs are essential multipliers in inflammatory mouse models, including DSS-induced colitis (Berndt *et al*, 2007) and IMQ-induced skin inflammation (Tortola *et al*, 2012). In the latter, DCs are the predominant source of IL-23 (Wohn *et al*, 2013; Riol-Blanco *et al*,

2014), the cytokine critical for the induction of IMQ-induced skin inflammation (van der Fits *et al*, 2009), and in the pathogenesis of psoriasis (Lee *et al*, 2004; Di Cesare *et al*, 2009; Cai *et al*, 2011). Specific targeting of the IL-23p19 subunit has shown promising results for the treatment of psoriasis (Glitzner *et al*, 2014; Kopp *et al*, 2015). In the macrophage cell line RAW 264.7 IL-23 expression induced by LPS has been shown to depend on c-Jun (Liu *et al*, 2009) and in DCs activated by LPS/Wortmannin on JNK (Wang *et al*, 2011). In DCs, c-Jun is a critical regulator of IL-23 expression downstream of TLR7 and regulates IL-23 expression, via the JNK/MAPK pathway and direct binding of c-Jun to the IL-23 promoter, to an extent similar to NF-κB in macrophages and DCs (Carmody *et al*, 2007; Mise-Omata *et al*, 2007). c-Jun/AP-1 also regulates the expression of additional pro-inflammatory cytokines by DCs directly or indirectly via IL-23-induced inflammation. Consistently, injection of recombinant IL-23 completely normalized IMQ-induced skin inflammation in *c-Jun*$^{\Delta/\Delta}$*CD11c*-Cre mice. Further studies are needed to assess the role of JNK/c-Jun signaling in other IL-23 related autoimmune diseases, like IBDs, rheumatoid arthritis, or multiple sclerosis.

Besides DCs, also macrophages or monocytes have been shown to be critical producers of IL-23 in psoriasis (Lee *et al*, 2004; Hänsel *et al*, 2011; Kopp *et al*, 2015). However, a strict classification is complicated, because many of the cell surface markers used for identification, like CD14 or CD163, are shared between these immune cell populations (Zaba *et al*, 2009; Bourdely *et al*, 2020). Different inflammatory dermal DC populations in psoriatic skin, such as TIP-DCs and Slan-DCs (CD11c$^+$CD1c$^-$), which are capable of producing cytokines like TNF-α or IL-23, were characterized, but IL-23 expression has also been shown in CD1c or CD14 expressing dermal immune cells (Lowes *et al*, 2005; Zaba *et al*, 2009; Hänsel *et al*, 2011). We show prominent c-Jun expression in CD1a-, CD1c-, and CD14-positive cells of human psoriatic lesional skin, which co-localized with IL-23 expression, suggesting a role of c-Jun in type-2/inflammatory DCs subsets for the pathogenesis of psoriasis.

In murine skin, Langerin negative DCs are the main producers of IL-23 (Wohn *et al*, 2013). We used *CD11c*-Cre mice, which delete in DCs, but may also delete in a subpopulation of CD11c expressing macrophages. Since neither the expression of *c-Jun*, nor *Ccl2* and *Il23p19* was affected in CD11c expressing macrophages sorted from *c-Jun*$^{\Delta/\Delta}$*CD11c*-Cre mice, it is unlikely that macrophages significantly contribute to the observed psoriatic phenotype. In fact, murine skin macrophages are CD11c$^{low/int}$ (Tamoutounour *et al*, 2013), and therefore, c-Jun is likely not deleted by *CD11c*-Cre in cutaneous macrophages. Consistently, in CD11c-diphtheria toxin receptor (DTR) transgenic mice, cell depletion was induced in dermal DCs, but not in macrophages, resulting in reduced IMQ-induced *Il23p19* mRNA expression (Riol-Blanco *et al*, 2014). This suggests that cDC2 are the main contributor to the phenotype observed in IMQ-treated *c-Jun*$^{\Delta/\Delta}$*CD11c*-Cre mice.

Besides IL-23, we also identified CCL2 expression to depend on c-Jun in DCs. A role for CCL2 in the pathogenesis of psoriasis has been postulated. CCL2 is expressed in keratinocytes of lesional psoriatic skin and may, together with other cytokines, be involved in the recruitment of monocytes from the circulation (Harden *et al*, 2014; Behfar *et al*, 2018). We also show CCL2 to be increased in lesional compared to healthy human skin. Furthermore, we show a strong correlation between *c-Jun* and *Ccl2* in psoriasis and co-expression in DCs of lesional skin by immunofluorescence. However, in IMQ-

treated *Ccl2*$^{-/-}$ mice we did not observe an effect on skin thickening, although the immune cell infiltrate was altered with a prominent reduction in monocytes, suggesting a redundant role of CCL2.

In conclusion, we demonstrate that c-Jun is an essential downstream transcription factor in TLR7 stimulated DCs regulating the expression of CCL2 and IL-23, resulting in attenuated psoriatic skin inflammation. The critical role of c-Jun/AP-1 in TLR7 signaling that we unraveled in this study will likely open new possibilities for targeted therapies in psoriasis.

# Materials and Methods

### Mice

Mice were kept in the animal facility of the Medical University of Vienna in accordance with institutional policies and federal guidelines. *c-Jun*$^{fl/fl}$ (Behrens *et al*, 2002) mice were crossed to mice with Cre recombinase under control of the *CD11c* (Caton *et al*, 2007), or *Mx1* promoter (Kuhn *et al*, 1995) or to mice expressing an estrogen-receptor fusion Cre recombinase under control of the *K5* promoter (Brocard *et al*, 1997). *Tlr7*$^{-/-}$ (Hemmi *et al*, 2002) and *Ccl2*$^{-/-}$ (Lu *et al*, 1998) have been described previously. Mice were kept in a mixed background (129 Sv × C57BL/6). Mice had unlimited access to standard laboratory chow and water under a light dark cycle of 12 h in a room with a temperature of 22°C. *Mx1*-Cre mice were injected with poly I:C (200 μg, i.p., VWR) twice, at an interval of 5 days, to induce deletion and taken earliest 9 days after the last injection. *K5*-Cre-ER$^{T2}$ mice were injected with Tamoxifen (Tx, 1 mg, i.p., Sigma-Aldrich) for 5 consecutive days. *c-Jun*$^{fl/fl}$ mice were used as controls in experiments with the respective *c-Jun*$^{\Delta/\Delta}$ *Mx1*-Cre, *CD11c*-Cre, or *K5*-Cre-ER$^{T2}$ mice. Wild-type mice were used as controls for *Ccl2*$^{-/-}$. Control mice received the same treatment as experimental mice, e.g. poly I:C (*Mx1*-Cre) or Tx (*K5*-Cre-ER$^{T2}$).

### Imiquimod treatment

Female and male mice, 8–12 weeks of age were used for all experiments. Back skin of mice was shaved (Aesculap GT608) and waxed (Veet, Heidelberg, Germany) 48 h before the start of IMQ treatment. Mice were treated daily with a 5% cream formulation of IMQ (Aldara, Meda Pharma) on the back skin for up to 7 days and left untreated after day 7 for 3 additional days to study the resolution of skin inflammation.

### Pharmacological inhibition of JNK-IL23R signaling *in vivo*

The JNK inhibitor SP600125 (Abcam) was administered intraperitoneally at a dose of 15 mg/kg. As a control mice received the vehicle (5% DMSO, 5% Tween-80, and 30% PEG-300). Alternatively, mice received 15 mg/kg of a monoclonal antibody to mouse IL-23R (i.p., Merck Research Laboratories) or an isotype mouse IgG1 antibody (i.p., Hölzel).

### Psoriasis-like mouse model

The previously described *c-Jun/JunB*$^{\Delta/\Delta}$ *K5*-Cre-ER$^{T2}$ mice were kept in a mixed background (129 Sv × C57BL/6 background) for a

psoriasis-like phenotype to develop (Zenz *et al*, 2005) after injection of Tx (1 mg, i.p.) for 5 consecutive days. 14 days after the first Tx injection administration of JNK inhibitor SP600125 (15 mg/kg, i.p., every other day, Abcam) started for 2 weeks. Psoriasis severity of experimental mice was analyzed similar to a previously reported psoriasis severity scoring system (Glitzner *et al*, 2014). Shortly, ear, tail, paw, and snout were scored from 1 to 4 (healthy to severest).

### Intradermal cytokine injection

One μg of recombinant IL-23 (BioLegend) or CCL2 (BioLegend) was injected daily into a defined, marked area of the back skin. PBS was injected as a control.

### Flow cytometry

Single-cell suspensions were prepared (see Appendix Methods) and subsequently blocked with anti-CD16/32 antibody (BioLegend). Cell surface markers were stained with fluorescently labeled antibodies for 30 min at 4°C (see Appendix Table S1). After incubation, cells were washed, filtered, and stained with 7-AAD Viability Staining Solution (BioLegend) according to the manufactures recommendation to exclude dead cells. Cells were recorded on a LSR Fortessa cell analyzer (BD Biosciences) and analyzes was done with FlowJo software (Version 10.6.1, Treestar).

### Intracellular cytokine staining

Cells were incubated with Brefeldin A (BioLegend) for 4 h after stimulation. A fixable viability dye (Zombie Aqua, BioLegend) was used to distinguish dead cells. Cell surface markers were stained before cells were fixed and permeabilized with BD Cytofix/Cytoperm (BD Biosciences). Permeabilized cells were incubated with the appropriate intracellular antibody (see Appendix Table S1) or isotype control for 30 min at 4°C.

### Cell sorting

Immune cells were sorted from back skin. For sorting, a single-cell suspension was prepared and cells were stained with the appropriate fluorescently labeled antibodies. Cells were sorted into TRIzol LS Reagent (Thermo Fisher Scientific) with a FACS Aria III. RNA was extracted with the column-based miRNeasy Micro Kit (Qiagen) according to the manufacturer's protocol. cDNA synthesis was performed with Superscript IV Reverse Transcriptase (Thermo Fisher Scientific).

### TEWL measurement

TEWL of back skin was measured with a Tewameter® TM 300 probe attached to the MDD4-display device (Courage + Khazaka) according to manufacturer's recommendations.

### RNA extraction

Back skin was stored in RNALater (Applied Biosystems) at −20°C until use. To isolate RNA, the back skin was mechanically disrupted with a Precellys 24 homogenizer (Bertin, ~ 5,000 *g*, 2 × 30 s). Cell debris was removed by centrifugation. RNA was isolated with TRIzol Reagent by chloroform/ethanol extraction according to the manufacturer´s protocol (Invitrogen). Protoscript II (New England Biolabs) was used for cDNA synthesis. Real-time quantitative PCR (qRT–PCR) analysis was performed on an ABI7500-Fast Real Time PCR system (Applied Biosystems) with Power SYBR Green Master Mix (Applied Biosystems). Primer sequences are listed in Appendix Table S2.

### Protein quantification

A piece of back skin was snap frozen and stored at −80°C. Samples were homogenized with a Precellys homogenizer (Bertin) in Precellys tubes (Peqlab). Total protein quantification in skin lysates was done by Bradford protein assay according to the manufacturer´s protocol (Bio-Rad). 40 (ELISA) or 100 μg (Luminex) of total protein diluted in ELISA assay diluent (BioLegend) with a protease inhibitor cocktail (Roche) were used to perform an ELISA for CCL2 (BD Biosciences) or a Multiplex Luminex assay (Thermo Fisher Scientific) according to manufacturer's instructions. Analysis of the Multiplex Luminex assay was done on a Luminex MAGPIX System using the xPONENT Software.

### Cell culture

To generate BM-derived immune cells from *Mx1*-Cre mice *in vitro*, deletion was first induced by poly I:C injection *in vivo* (as described above). Controls were treated equally. BM was then isolated from tibia and femur, and red blood cells were lysed. Isolated cells were cultured in 4 ml RPMI-1640 (Gibco) with 10% FCS (PAA), penicillin, streptomycin, non-essential amino acids (Sigma-Aldrich), sodium pyruvate (Sigma-Aldrich), β-mercaptoethanol (Gibco), and GM-CSF (20ng/mL, Peprotech) for 3 days. On day 3, media was changed. On day 6, non-adherent cells (BMDCs) were harvested and used for experiments (CD11c$^+$ cells > 80%). BMDCs were stimulated with IMQ, CL264, or CL307 (InvivoGen). For inhibition BMDCs were pretreated with SP600125 (InvivoGen; 25 μM) or T-5224 (20 μM, ApexBio) for 1 h before stimulation. Protein levels of CCL2 (BD Biosciences) and IL-23 (BioLegend) were analyzed by ELISA in BMDC supernatants. BM-pDCs were generated as previously described (Drobits *et al*, 2012).

### Chromatin immunoprecipitation

The SimpleChIP Enzymatic Chromatin IP Kit (Cell Signaling Technology, #9003) was used to perform a chromatin immunoprecipitation. In short, BMDCs were stimulated with IMQ or LAL for 2 h, fixed with 1% (v/v) formaldehyde for 10 min at RT, cells were lysed, digested with micrococcal nuclease, and ultrasonicated using a Bioruptor Plus (diagenode) sonication device. Sonicated samples were incubated with 10 μg anti-c-Jun antibody (Cell signaling, 60A) o/n at 4°C, followed by an incubation with 30 μl CHIP grade magnetic beads (2 h, 4°C). DNA was isolated using spin columns after an elution and digestion of samples with Proteinase K (40 μg). Recovered DNA was analyzed by qRT–PCR with primers specific to the presumptive c-Jun binding site of the IL-23p19 promoter (see Appendix Table S2).

## Luciferase reporter assay

The IL-23p19 promoter was cloned into the pGL3-basic vector (Promega). Site-directed mutagenesis was performed using the Q5 Site-Directed Mutagenesis Kit according to the manufacture´s protocol (New England Biolabs). RAW 264.7 cells (ATCC) were cultured in Dulbecco's modified Eagle's medium (Gibco) supplemented with 10% FCS (PAA) and re-plated ($6 \times 10^5$ cells/well) one day before transfection. Transfection was done with the jetPRIME transfection reagent according to the manufacture´s protocol (Polyplus-transfection). After transfection (4 h), cells were pretreated (1 h) with JNK Inhibitor (SP600125, 25 μM) and stimulated with IMQ (6 h) before luciferase activity was measured using the Luciferase Assay System (Promega). Mutation primers are listed in Appendix Table S2.

## Human monocyte-derived dendritic cells

Buffy coats from healthy donors were purchased from the Medical University of Graz, Department of Transfusion Medicine, Austria. For the isolation of peripheral blood mononuclear cells, heparinized blood was separated by gradient centrifugation with Lymphoprep™ (Axis Shield). $CD14^+$ monocytes were positively selected according to the manufacturer's instructions (CD14 MicroBeads, Miltenyi Biotec). For moDC generation, $CD14^+$ monocytes ($1x10^6$ cells/ml) were cultured for 6 days in RPMI-1640 (Sigma-Aldrich) supplemented with 10% FBS and recombinant human cytokines: 35 ng/ml IL-4 and 100 ng/ml GM-CSF (PeproTech).

## LL-37/RNA-40 stimulation

Human mo-DCs were pretreated for 1 h with JNK inhibitor SP600125 (25 μM, InvivoGen) or the AP-1 inhibitor T-5224 (20 μM, ApexBio) before a stimulation with 10 μg/ml of Resiquimod (R848, InvivoGen) or RNA-40 (10 μg/ml; iba-lifesciences) in complex with LL-37 (50 μg/ml; InvivoGen) for 4 or 24 h. For complex formation, RNA-40 was incubated with LL-37 for 1 h at room temperature. Human CCL2 and IL-23 (Thermo Fisher Scientific) was analyzed by ELISA.

## Human patient material

Paraffin-embedded ($n = 2$) or frozen ($n = 2$) lesional and non-lesion ($n = 2$) materials from biopsy samples were available from psoriasis patients. Healthy skin ($n = 2$) was collected after plastic surgery. All samples were obtained through informed consent by an approved protocol (Ethics approval 27-071/ Medical University of Graz Institutional Review Board and 1360/2018/Medical University of Vienna). All experiments conformed to the principles set out in the WMA Declaration of Helsinki and the Department of Health and Human Services Belmont Report.

## Immunofluorescence of human skin

For double immunofluorescence (IF) of c-Jun with CD1a, CD1c or CD14 paraffin sections (5 μm) were deparaffinized in xylene and rehydrated with decreasing concentrations of ethanol according to standard method. Sections were subjected to HIER antigen retrieval in Target Retrieval Solution pH 6.0 (Agilent/Dako, USA) for 10 min.

### The paper explained

**Problem**

The TLR7/8 signaling pathway is an antiviral, innate immune defense mechanism implicated in the pathogenesis of psoriasis, a common inflammatory skin disease. TLR stimulation culminates in the activation of a set of downstream effectors, among them c-Jun/AP-1 proteins, but their role in regulating the cellular response to TLR7 signaling remains poorly understood.

**Results**

Here, we show that c-Jun/AP-1 is an essential mediator of the TLR7 signaling pathway in DCs to promote skin inflammation in two clinically relevant mouse models of psoriasis. Mechanistically, c-Jun/AP-1 regulates the pDC recruiting chemokine CCL2 and the T-cell activating cytokine IL-23 in murine and human DCs via the JNK/TLR7 signaling pathway. Inhibition of JNK/c-Jun activity by pharmacological blockade alleviated skin inflammation in a chemically (IMQ) and genetically induced murine psoriasis model. In psoriasis, we show co-expression of c-Jun with CCL2 and IL-23 in type-2/inflammatory DCs.

**Impact**

Our results have uncovered a novel role of c-Jun in immune cells for the pathogenesis of psoriasis, identified the JNK/c-Jun signaling axis as a druggable target, and thereby provided a new therapeutic treatment strategy for patients with psoriasis.

Slides were allowed to cool for 45 min before rinsing in Tris-Buffered Saline Tween-20 (0.5%) (TBS-T). Sections were blocked with 5% donkey serum, 5% BSA in TBS-T for 1 hour before incubation with primary antibodies (overnight, 4°C). Slides were washed (3 × 10 min TBS-T), and appropriate secondary AB was applied for 1 hour. DAPI was used to counterstain and slides were mounted with Dako Fluorescence Mounting Medium (Agilent, Inc., Santa Clara, CA).

For double IF of c-Jun with CD11c skin biopsies were embedded in optimal cutting temperature compound O. C. T.™ (Sakura), cut (5 μm), fixed in paraformaldehyde, blocked, and incubated with primary and secondary AB as described above.

For triple IF of c-Jun with DC markers (CD1a, CD1c, CD14) and cytokines (CCL2, IL-23), frozen skin was acetone fixed, washed (3 × PBS), blocked (5% donkey serum in PBS) for 1 h at room temperature, and incubated with the primary antibodies overnight (4°C). Subsequently, slides were washed (3 × PBS) and incubated with secondary antibodies (1 h, room temperature). Slides were counterstained with DAPI. Images were recorded on a LSM700 Confocal Microscope. ZEN 3.0 image acquisition software was used for image processing. On merged images adjustment of individual color channels was done with Adobe Photoshop. Deconvolution was done with Huygens software.

## Human RNA expression data

RNA-Seq data from (Tsoi *et al*, 2019a) were analyzed using the processed read-count data supplied by the submitter (Data ref: Tsoi *et al*, 2019b). Raw read-counts from healthy ($n = 38$), non-lesional ($n = 27$), and lesional ($n = 28$) patient-derived skin transcriptomes were TMM (Trimmed Mean of M-values) normalized using the edgeR package (Robinson *et al*, 2010; McCarthy *et al*, 2012) from the Bioconductor project to compare expression values.

## Statistical analysis

GraphPad Prism 8 software was used for statistical analysis. Unpaired and paired two-tailed Student's *t*-test, one-way, or two-way ANOVA with Tukey or Bonferroni multiple comparison test (selected pairs of columns) were used. Welch's correction was performed on Student's *t*-test, if variance between groups was significantly different. When possible normality of data was assessed with D'Agostino and Pearson test, outliers were identified by Grubbs´ test or ROUT Method. *P* values of < 0.05 were considered statistically significant (*$P < 0.05$, **$P < 0.01$, ***$P < 0.001$, ****$P < 0.0001$). Exact *P* values are listed in Appendix Table S3.

## Data availability

This study includes no data deposited in external repositories.

Expanded View for this article is available online.

## Acknowledgements

We thank Temenuschka Baykuscheva-Gentscheva for help with histology and Life Science Editors for editorial assistance. We are grateful to M. Hammer and the staff of the Department of Biomedical Research of the Medical University of Vienna for maintaining our mouse colonies. We thank J. Reisecker for help with confocal microscopy. This work was supported by grants from the Austrian Science Fund (FWF, PhD program W1212 "Inflammation and Immunity" and the European Research Council (ERC) Advanced grant (ERC-2015-AdG TNT-Tumors 694883) to M. Sibilia. M. Sibilia´s laboratory receives funding by the WWTF-project LS16-025 and the European Union's Horizon 2020 research and innovation program under the Marie Skłodowska-Curie grant agreement No. 766214 (Meta-Can). T. Bauer was supported by the FWF grants P27129-B20 and I 4300-B. H. Strobls lab was funded by FWF project P2572, FWF doctoral program W1241 and by Biotechmed Graz, Flagship project "Secretome". The Wagner laboratory is supported by the ERC (ERC-AdG 2016 CSI-Fun-741888), a H2020-MSCA-ITN (ITN-2019-859860 - CANCERPREV) and the MUW.

## Author contributions

PN performed and analyzed most of the experiments. GS and CF helped in some flow cytometry experiments. MH helped in some cell sorting experiments. IB and VZ performed experiments related to human monocyte-derived dendritic cells. HS, PMB, and GS provided psoriatic skin biopsies. CT-A performed some of the human IF staining´s. AB performed the promoter study. ML performed some Western Blot experiments. LB and EFW helped in chromatin immunoprecipitation experiments. BD, TB, and MH participated in experimental design and interpretation of data. MS conceived and supervised the project and provided the requested funding. All authors approved the final version of the manuscript.

## Conflict of interest

The authors declare that they have no conflict of interest.

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
