## [Review Process File · EMBO Molecular Medicine]

Psoriatic skin inflammation is promoted by c-Jun/AP-1 dependent CCL2 and IL-23 expression in Dendritic cells

Philipp Novoszel, Martin Holcman, Gabriel Stulnig, Cristiano De Sa Fernandes, Victoria Zyulina, Izabela Borek, Markus Linder, Alexandra Bogusch, Barbara Drobits, Thomas Bauer, Carmen Tam-Amersdorfer, Patrick Brunner, Georg Stary, Latifa Bakiri, Erwin Wagner, Herbert Strobl, and Maria Sibia

DOI: [10.15252/emmm.202012409](https://doi.org/10.15252/emmm.202012409)

Corresponding author: Maria Sibia (maria.sibia@meduniwien.ac.at)

Review Timeline:

Submission Date:	27th Mar 20
Editor Correspondence:	7th May 20
Author Correspondence:	14th May 20
Editorial Decision:	20th May 20
Appeal:	3rd Jun 20
Editorial Decision:	17th Jun 20
Revision Received:	23rd Sep 20
Editorial Decision:	14th Oct 20
Revision Received:	23rd Dec 20
Editorial Decision:	14th Jan 21
Revision Received:	28th Jan 21
Accepted:	1st Feb 21

Editor: Lise Roth

Transaction Report:

7th May 2020

Dear Maria,

Thank you for the submission of your manuscript to EMBO Molecular Medicine. We have now received feedback from two of the three reviewers who agreed to evaluate your manuscript. Referee #2 has unfortunately not returned his/her report so far despite several chasers, and in order to avoid further delay in the process we would like to make a decision now.

As you will see from the reports pasted below, while mentioning the interest of the findings, both referees also raise several major and partially overlapping concerns. Revising the manuscript according to the referees' recommendations appears to require a lot of additional work and experimentation, and I am unsure whether you will be able or willing to address those.

In some cases, we may consult with the authors before deciding on the outcome. In this case, we would like to know if you would be ready to perform the additional experiments required to address the referees' concerns, or alternatively to provide a detailed point-by-point response on how you plan to address these issues? We do understand the Covid-19 pandemic affects scientific work at every level, and we would be ready to extend the revision time if needed.

Looking forward to hearing from you,

With my best wishes,

Lise

Lise Roth, PhD
Editor
EMBO Molecular Medicine

Referee #1:
Comments on novelty/model system:

There are some discrepancies that need to be addressed - i.e. the designation of pDCs. Also, the background strains are not noted in the manuscript (Materials and Methods).

Remarks for author:

Comments on "Inflammation and anti-Immunity is promoted by c-Jun/AP-1-dependent CCL2 and IL-23 expression in Dendritic Cells" by Dr. Novoszel and colleagues. In this manuscript the authors demonstrate the role of c-Jun/AP-1 signaling in dendritic cells to IL-23 mediated inflammatory responses in the IMQ mouse model. The novel findings here are the role of C-jun in DCs and in driving pDC infiltration and IL-23 production by DCs. Other studies have shown dependence of IL-23 secretion by dendritic cells by the JNK pathway

(Wang Q, Immunology 2011; Liu W, JBD 2009). I have the following comment.

Major

One aspect of this paper that makes it hard to follow is that some of the data shown here doesn't fit well together, at least not in the context of psoriasis. For example, the tumor growth kinetics of melanoma cells as shown in Figure 3 is certainly interesting but doesn't really fit well with the psoriasis angle of the rest of the manuscript. Also, the IMQ model is overused as a model for studying the pathogenesis of psoriasis and it is a self-limited acute model that is used to study a complex chronic inflammatory disease. The authors should consider revising and addressing this.

Results: Why was Mx1-Cre chosen? The authors state that it is "broadly inducible", which is vague. Mx1-Cre was induced by poly I:C that activates TLR3 signaling. Because these mice were also treated with IMQ to activate TLR7 signaling, is it possible to distinguish which TLR is responsible for the observed inflammatory response? Is it possible that a prior activation of TLR3 by poly I:C affects the inflammatory response to IMQ? It would help if the negative and positive controls are clearly stated. Are positive controls the Mx1-Cre⁻, c-Jun^{flox/flox} mice that were treated with poly I:C at first and then with IMQ? Or were the controls just treated with IMQ? When the authors simply state "c-Jun Δ/Δ Mx1-Cre mice had significantly less IMQ induced skin inflammation", it is not exactly clear what they are being compared to. This also affects the BMDC experiments from these mice. Same applies to the tamoxifen-induced K5-Cre. Also, for all three c-Jun deletion models, was the deletion efficiency confirmed?

Why were Mx1-Cre⁺, and not CD11c-Cre⁺, mice used to isolate BMDC to look at the effect of IMQ on DC in vitro (Fig. 2C-2F)? What is the rationale for using either Mx1-Cre or CD11c-Cre for some experiments, but not the others (e.g. Fig. 3B-C vs. Fig G-L or Fig 5B vs. Fig. 5C)?

Figures 2A and 2H both look at pDCs by flow cytometry, but they designate pDCs differently (BST-2⁺ CD11c^{int}CD11b⁻ vs. B220⁺BST-2⁺CD11c^{int}CD11b⁻). How come? For some flow cytometry experiments (Fig. 1-2, Fig. 6K), an immune subpopulation that is being quantified is imputed as % of total live single cells. However, the authors use % of total CD45⁺ cell in other instances (Fig. 3, Fig. 6J). What is the rationale?

Figure 4B-C: In the mice that lack c-Jun in DCs, how can 15-40% of DCs be c-Jun⁺? Wouldn't flow cytometry be more suitable than IF for this experiment? Also, the figure caption for states "(C) Box and whiskers plot shows the percentage of c-Jun positive skin dendritic cells quantified by immunofluorescence performed as described in (C). (n= 5, 3 experiments)." Do the authors mean as described in B?

The CCL2 angle is interesting and the data with the pDCs, but how specific is this to pDCs? Did the authors look at the infiltration of other dendritic cell subsets in their model? Also, while pDCs have been implicated in psoriasis pathogenesis they seem to play the greatest role at the initiation or triggering of psoriatic plaques (Nestle FO, JEM 2005). Also, I'm not aware of pDCs having a cytotoxic role in psoriasis. Interestingly, the authors show that it is dispensable for IMQ-induced skin inflammation - this also makes these data not fit well in the context of psoriasis. Furthermore, the connection between CCL2 and psoriasis could be improved by showing increased CCL2 in psoriasis and co-localization with DCs in psoriatic

skin.

Minor.

Figure 5H - refers to "epidermal area" - why did the authors decide to look at the "area" instead of "epidermal thickness", which is much more traditionally used (and would be consistent with data elsewhere in this manuscript).

Page 11 "IL-23 induces IL-17A production in $\gamma\delta$ T cells, which contributes to IMQ-induced inflammation (Zaba et al, 2007)" - don't think this is what is stated in this paper. I don't think it mentioned gamma-delta cells, and certainly nothing about IMQ-induced inflammation. The authors may want to look at this more closely.

The authors should provide more information on the background strain used....were all the strains done on the same genetic background?

In Methods, the authors talk about JunBflox/flox mice, but no data on these mice is shown in the results.

Intro: If epidermal deletion of c-Jun and JunB causes psoriasis, then what is the possible explanation for the opposite effect here (c-Jun deletion in DCs ameliorating IMQ-induced skin inflammation)?.

Referee #3:

Comments on novelty/model system:

Technical quality (including statistical analysis).

Overall, the experiments are well performed. However, more precise Cre-models (e.g. Zbtb46-Cre, XCR1-Cre) are available to clearly show that it is DC-mediated.

Strength of the evidence for the conclusions drawn.

There is clear evidence that c-Jun plays a role in TLR7-induced skin inflammation. Novelty.

A new link of c-Jun with TLR7-induced skin inflammation.

Medical impact.

The study could lead to new treatment options for psoriasis by using small molecules against c-Jun.

Adequacy of model system.

As CD11c-Cre targets also monocytes and macrophages, other Cre-lines should be used. It would further strengthen the manuscript to identify, which DC subpopulation is responsible for the observed effect.

Remarks for author:

In the presented study by Novoszel et al., the role of c-Jun in the pathogenesis of a psoriasis model as well as on anti-tumor responses was analyzed. The authors clearly show that deletion of c-Jun in CD11c+ cells leads to ameliorated disease in Imiquimod (IMQ)-induced skin inflammation but also to weaker anti-tumor immune responses after treatment of B16F10 tumors with IMQ. In the latter case, they identify reduced CCL2 secretion by BMDCs and mRNA expression by skin DCs as cause for reduced pDC recruitment to the tumor. In the IMQ-induced skin inflammation model, the authors identified reduced secretion of IL-23 after deletion of c-Jun in CD11c+ cells as cause for the ameliorated disease. They

further showed that inhibition of c-Jun/AP-1 using small molecules can be used to treat the psoriasis-like disease. Overall, the study is well-performed and the results show clearly a role of c-Jun in TLR7-induced immune responses. However, in different parts of the manuscript the authors overestimate their results and should be more careful in the interpretation (see in the numbered comments below).

Major points:

1. In the manuscript, the authors state that c-Jun in DCs is necessary for IL-23 and CCL2 expression by using either c-Jun^{fl/fl} mice crossed with CD11c-Cre mice or BMDCs of Mx1-Cre mice crossed with c-Jun^{fl/fl} mice. However, CD11c is not exclusive for DCs but also expressed on macrophages, which is shown in Figure EV1. As the authors suggest that the secretion of CCL2 and IL-23 is mediated by cDCs, they should use the cDC-specific Zbtb46-Cre mice to avoid deletion of c-Jun in other cells than DCs. Further, the skin contains several subsets of cDCs with different functional properties. Therefore, it would strengthen the manuscript, if the authors identify whether all skin DC subsets respond to IMQ with secretion of CCL2 and IL-23 or only a certain subset.
2. In Figure 7, the authors want to translate their finding from the murine to the human system. Therefore, they perform immunofluorescence staining of skin sections of patients with psoriasis and stain for c-Jun as well as CD11c. However, CD11c is not a specific DC marker in humans as it is highly expressed on monocytes and macrophages. Furthermore, the images show that nearly all cells in the sections of the patients with psoriasis and of the healthy controls are c-Jun⁺ positive and the CD11c⁺ cells represent only a minor fraction of the c-Jun⁺ cells. Therefore, more evidence is needed that c-Jun in skin DCs is involved in IL-23 secretion. The authors should purify DCs from the skin analogously to the murine experiments.

Minor points:

3. In the results part and in the figure legends of the manuscript, the authors always use the name of the substance, Imiquimod, but in the Figure itself the tradename, Aldara. They either should change Aldara into Imiquimod in the figures or also mention Aldara in the body of the text (in the current version of the manuscript it is only mentioned at the end of the manuscript in the material and method section).
4. In figure 2A, the authors show that after deletion of c-Jun in CD11c⁺ cells less pDCs are recruited to the skin. However, even in wildtype mice pDCs represent less than 1% of the cells. As IMQ-treatment also leads to the recruitment of other immune cells, it would be helpful to show total numbers of recruited pDCs to exclude a proportional effect.
5. The authors claim that c-Jun in DCs is "necessary for the induction of various pro-inflammatory cytokines and key mediators of IMQ-induced skin inflammation (Fig. 4D-F)". However, except for IL-23p19 and IL-17 all shown cytokines and mediators are equally induced after 32h and only reduced after 48h. The authors should change the sentence accordingly and discuss, why c-Jun is only necessary at 48h but not 32h for some mediators (e.g. CXCL1).
6. The authors further claim that c-Jun is "essential for mRNA expression of the cytokines Il23p19, Il22, Il6 and the chemokines Ccl20 and Cxcl1 (Fig. EV3A)". However, Figure EV3A shows that all of those cyto- and chemokines are induced, especially in case of Il23p19, Il6 and Cxcl1. Therefore, the sentence should be changed as c-Jun is not "essential" but more a positive regulator.
7. The material and methods section is in some parts short and does not contain all details necessary to repeat the experiments.

14th May 2020

Dear Lise,

As promised please find enclosed our response to the reviewers with our planned experiments which we think should be doable within the next two months. Asking for breeding into an additional cre mouse or analysing sorted DCs from human skin is too much in my opinion and we should stay within a realistic frame. I hope you agree on that.

Looking forward receiving your opinion soon. We will meanwhile start performing the suggested experiments

best wishes and many thanks for your patience.

maria

Reviewer #1:

Comments on novelty/model system:

There are some discrepancies that need to be addressed - i.e. the designation of pDCs. Also, the background strains are not noted in the manuscript (Materials and Methods).

Remarks for author:

Comments on "Inflammation and anti-Immunity is promoted by c-Jun/AP-1-dependent CCL2 and IL-23 expression in Dendritic Cells" by Dr. Novoszel and colleagues. In this manuscript the authors demonstrate the role of c-Jun/AP-1 signaling in dendritic cells to IL-23 mediated inflammatory responses in the IMQ mouse model. The novel findings here are the role of C-jun in DCs and in driving pDC infiltration and IL-23 production by DCs. Other studies have shown dependence of IL-23 secretion by dendritic cells by the JNK pathway (Wang Q, Immunology 2011; Liu W, JBD 2009). I have the following comment.

Major:

One aspect of this paper that makes it hard to follow is that some of the data shown here doesn't fit well together, at least not in the context of psoriasis. For example, the tumor growth kinetics of melanoma cells as shown in Figure 3 is certainly interesting but doesn't really fit well with the psoriasis angle of the rest of the manuscript. Also, the IMQ model is overused as a model for studying the pathogenesis of psoriasis and it is a self-limited acute model that is used to study a complex chronic inflammatory disease. The authors should consider revising and addressing this.

We thank the reviewer for constructive criticisms of our manuscript. In principle we agree that the tumor results might not seem to fit well to the psoriasis data. However, with these results we wanted to strengthen our findings on the regulation of CCL2 by *c-Jun* in skin DCs. We know from previous studies with the B16-F10 melanoma model that CCL2 is required for efficient IMQ-induced recruitment of pDCs to the tumor site in the skin where they acquire a tumor killing capacity (Drobits et. al JCI 2012). With this experiment here we can show that in the absence of *c-Jun* in DCs, pDCs are not recruited to the tumor site upon IMQ treatment because of reduced CCL2 production. However, pDCs still maintain their killing capacity as shown by the *in vitro* killing assays demonstrating that this is independent of *c-Jun* expression. For this reason we would like to keep these data in the manuscript. Moreover, Reviewer 3 also appreciated these results. We will consider restructuring the manuscript according to the new results, which will be produced in the course of the revision.

We also agree with the reviewer that the commonly used IMQ model has limitations with regard to the pathogenesis of psoriasis. Therefore we have also employed an additional mouse model of psoriasis based on the inducible genetic deletion of *c-Jun* and *JunB* in the epidermis (*c-Jun/JunB*^{Δ/Δ} *K5-Cre* *ER*^{T2}). Also with this model (Figures EV 4 A-D) we can show that treatment with JNK inhibitor ameliorates the psoriasis phenotype emphasizing the translational potential of our therapeutic findings. We will move these results to the main part of the manuscript.

Why was Mx1-Cre chosen? The authors state that it is "broadly inducible", which is vague. Mx1-Cre was induced by poly I: C that activates TLR3 signaling. Because these mice were also treated with IMQ to activate TLR7 signaling, is it possible to distinguish which TLR is responsible for the observed inflammatory response? Is it possible that a prior activation of TLR3 by poly I: C affects the inflammatory response to IMQ? It would help if the negative and positive controls are clearly stated. Are positive controls the Mx1-Cre-, c-Jun flox/flox mice that were treated with poly I: C at first and then with IMQ? Or were the controls just treated with IMQ? When the authors simply state "c-Jun Δ/Δ Mx1-Cre mice had significantly less IMQ induced skin inflammation", it is not exactly clear what they are being compared to. This also affects the BMDC experiments from these mice. Same applies to the tamoxifen-induced K5-Cre. Also, for all three c-Jun deletion models, was the deletion efficiency confirmed?

We initially employed this Cre line for the inducible deletion of *c-Jun* in interferon responsive skin cells, such as keratinocytes and hematopoietic cells, and to thereafter identify, which cell type is specifically responsible for the observed phenotype, immune cells or parenchymal cells (Fig. 1B-D). We also kept it as a second, independent Cre line for deletion in hematopoietic cells to confirm the results shown with the *CD11c*-Cre line and exclude Cre-mediated artifacts. The time line in Fig. 1A shows that the last injection of poly I: C takes place at least 9 days before IMQ treatment starts. By this time there is no residual inflammatory signature detectable. As can be seen in Figure 1 B-D the IMQ-induced epidermal thickening is comparable between *c-Jun*^{fl/fl} mice (control in CD11c-Cre mouse model; Figure 1C) to *c-Jun*^{fl/fl} mice treated with poly I: C or Tamoxifen (control in *Mx-1* – Cre or *K5-Cre* *ERT2* mouse model, respectively; Figure 1B, D) arguing against an effect of prior TLR3 activation on the course of the inflammatory response. In all experiments *c-Jun*^{fl/fl} were used as controls and were treated equally to Cre+ mice. We will include all this information in the figures and methods section. Moreover, the deletion efficiency of *c-Jun* with *CD11c*-Cre can be found in Figure EV1 B, C; the deletion efficiency with all other Cre lines will be included as well in this figure and we are sorry of having omitted it in the current version.

Why were Mx1-Cre+, and not CD11c-Cre+, mice used to isolate BMDC to look at the effect of IMQ on DC in vitro (Fig. 2C-2F)? What is the rationale for using either Mx1-Cre or CD11c-Cre for some experiments, but not the others (e.g. Fig. 3B-C vs. Fig G-L or Fig 5B vs. Fig. 5C)?

We tried to strengthen our point by combining *in vitro* and *in vivo* findings and different Cre mouse models to exclude Cre-mediated artifacts. We used *Mx1*-Cre mice mainly for *in vitro* experiments because of the higher recombination efficiency of this Cre line in BMDCs compared to *CD11c*-Cre mice.

Figures 2A and 2H both look at pDCs by flow cytometry, but they designate pDCs differently (BST-2+ CD11cint CD11b- vs. B220+ BST-2+ CD11cint CD11b-). How come? For some flow cytometry experiments (Fig. 1-2, Fig. 6K), an immune subpopulation that is being quantified is imputed as % of total live single cells. However, the authors use % of total CD45+ cell in other instances (Fig. 3, Fig. 6J). What is the rationale?

pDCs were uniformly identified as being BST-2⁺B220⁺CD11b⁻CD11c^{int}. We apologize for the labeling error in Figure Legend 2A, this will be corrected. The numbers in Figure 6J represent the actual numbers of the plot, which is gated on CD45 and these numbers thus reflect the relative change within the immune cell infiltrate. For Figure 6K,

on the other hand, we show the percentage of live cells to point out that this is also reflected by the total number of the respective cell population in the skin.

Figure 4B-C: In the mice that lack c-Jun in DCs, how can 15-40% of DCs be c-Jun+? Wouldn't flow cytometry be more suitable than IF for this experiment? Also, the figure caption for states "(C) Box and whiskers plot shows the percentage of c-Jun positive skin dendritic cells quantified by immunofluorescence performed as described in (C). (n= 5, 3 experiments)." Do the authors mean as described in B?

Thank you for this comment. Although flow cytometry would be a very suitable experimental approach, we were not able to establish a convincing intracellular flow cytometry staining of *c-Jun* from ex vivo isolated skin DC. We thus performed IF. The residual c-Jun expression (15-40 %) reflects the incomplete deletion in CD11c + cells under inflammatory conditions, where CD11c levels are reduced and thus Cre expression is low. We are sorry for the labelling error, we will change this accordingly (B).

The CCL2 angle is interesting and the data with the pDCs, but how specific is this to pDCs? Did the authors look at the infiltration of other dendritic cell subsets in their model? Also, while pDCs have been implicated in psoriasis pathogenesis they seem to play the greatest role at the initiation or triggering of psoriatic plaques (Nestle FO, JEM 2005). Also, I'm not aware of pDCs having a cytotoxic role in psoriasis. Interestingly, the authors show that it is dispensable for IMQ-induced skin inflammation - this also makes these data not fit well in the context of psoriasis. Furthermore, the connection between CCL2 and psoriasis could be improved by showing increased CCL2 in psoriasis and co-localization with DCs in psoriatic skin.

We appreciate that the reviewer values our findings on the regulation of CCL2 in DCs by c-Jun and its role for the recruitment of pDC. As shown in Figure EV 1 G-I the infiltration of other dendritic cell subsets (CD103+ cDC1 and CD11b+ cDC2) is not affected. As the Reviewer points out pDC stimulation by viral TLR agonist's (TLR-7/9) has been shown to trigger psoriatic plaques. We therefore think that the data we present, showing a role of c-Jun in pDCs after stimulation of TLR7, has relevance for a better understanding of the function of pDCs in psoriasis. While we show that CCL2 contributes to the psoriatic inflammatory infiltrate by controlling pDC and monocyte-macrophage recruitment (see Appendix Figure 1), we demonstrate that epidermal thickening, an adverse event of IMQ therapy, is unaffected. Because these therapeutic implications can only be understood in context (inflammation & tumor data), we consider it important to present these tumor data in our manuscript. However, as stated above, we will try to communicate this part of the manuscript in a clearer way. We agree with the Reviewer that the connection between CCL2 and psoriasis should be further strengthened and we will therefore perform analysis of CCL2 expression in psoriasis and co-localization of CCL2 and c-Jun with DCs in psoriatic skin, as suggested.

Minor:

Figure 5H - refers to "epidermal area" - why did the authors decide to look at the "area" instead of "epidermal thickness", which is much more traditionally used (and would be consistent with data elsewhere in this manuscript).

We wanted to show that different methods of analyzing skin thickening give the same result, namely reduced thickening in *c-Jun*^{Δ/Δ} *CD11c*-Cre mice, but agree with the Reviewer that for reasons of uniformity it is better displayed as epidermal thickness.

Page 11 "IL-23 induces IL-17A production in $\gamma\delta$ T cells, which contributes to IMQ-induced inflammation (Zaba et al, 2007)" - don't think this is what is stated in this paper. I don't think it mentioned gamma-delta cells, and certainly nothing about IMQ-induced inflammation. The authors may want to look at this more closely.

Thank you for pointing out this mistake. We cited the wrong paper and will replace the reference by (Tortola et al, 2012).

The authors should provide more information on the background strain used....were all the strains done on the same genetic background?

All mice were in a mixed background as stated in the materials and methods section.

In Methods, the authors talk about JunB flox/flox mice, but no data on these mice is shown in the results.

Thank you for this comment. This respective section will be changed.

Intro: If epidermal deletion of c-Jun and JunB causes psoriasis, then what is the possible explanation for the opposite effect here (c-Jun deletion in DCs ameliorating IMQ-induced skin inflammation)?

Thank you for this comment. We believe that one of the major findings of this study is the apparent functional complexity and diverse cell-context dependent roles of Jun proteins in different cells of the skin (as mentioned briefly in the discussion). Keratinocytes, which are neither TLR7 responsive (Drobits et al, 2012a) (Supplemental Figure 1A) nor important producers of IL-23 in our model (see Figure EV 3D), contribute functionally different to skin homeostasis than DCs. Furthermore, while there seems to be some redundancy for Jun proteins in keratinocytes (psoriatic disease is only induced if both c-Jun and JunB are deleted in keratinocytes), our results here clearly demonstrate that this is not the case for DC, where c-Jun deletion is sufficient to ameliorate inflammation. This offers a therapeutic window for the treatment of psoriasis with topical JNK inhibitors, which should only inhibit c-Jun and not JunB as the latter does not seem to get phosphorylated by JNK. The observed opposite outcome also reflects the unique cell-specific TLR expression pattern and cytokine expression profile of epidermal vs. immune cells driving or ameliorating skin inflammation.

Reviewer #3:

Comments on novelty/model system:

Technical quality (including statistical analysis).

Overall, the experiments are well performed. However, more precise Cre-models (e.g. Zbtb46-Cre, XCR1-Cre) are available to clearly show that it is DC-mediated.

Strength of the evidence for the conclusions drawn.

There is clear evidence that c-Jun plays a role in TLR7-induced skin inflammation.

Novelty.

A new link of c-Jun with TLR7-induced skin inflammation.

Remarks for author:

In the presented study by Novoszel et al., the role of c-Jun in the pathogenesis of a psoriasis model as well as on anti-tumor responses was analyzed. The authors clearly show that deletion of c-Jun in CD11c+ cells leads to ameliorated disease in Imiquimod (IMQ)-induced skin inflammation but also to weaker anti-tumor immune responses after treatment of B16F10 tumors with IMQ. In the latter case, they identify reduced CCL2 secretion by BMDCs and mRNA expression by skin DCs as cause for reduced pDC recruitment to the tumor. In the IMQ-induced skin inflammation model, the authors identified reduced secretion of IL-23 after deletion of c-Jun in CD11c+ cells as cause for the ameliorated disease. They further showed that inhibition of c-Jun/AP-1 using small molecules can be used to treat the psoriasis-like disease. Overall, the study is well-performed and the results show clearly a role of c-Jun in TLR7-induced immune responses. However, in different parts of the manuscript the authors overestimate their results and should be more careful in the interpretation (see in the numbered comments below).

Major points:

1. In the manuscript, the author's state that c-Jun in DCs is necessary for IL-23 and CCL2 expression by using either c-Jun^{fl/fl} mice crossed with CD11c-Cre mice or BMDCs of Mx1-Cre mice crossed with c-Jun^{fl/fl} mice. However, CD11c is not exclusive for DCs but also expressed on macrophages, which is shown in Figure EV1. As the authors suggest that the secretion of CCL2 and IL-23 is mediated by cDCs, they should use the cDC-specific Zbtb46-Cre mice to avoid deletion of c-Jun in other cells than DCs. Further, the skin contains several subsets of cDCs with different functional properties. Therefore, it would strengthen the manuscript, if the authors identify whether all skin DC subsets respond to IMQ with secretion of CCL2 and IL-23 or only a certain subset.

We appreciate the positive feedback on the manuscript and the valuable comments on how to improve the manuscript. We have performed the majority of *in vivo* experiments with *CD11c-Cre* mice for two reasons:

- a) It deletes in all DC populations that we are interested in (cDCs and importantly pDCs).
- b) When we started our experiments this was the best one available.

We agree with the reviewer that it would be ideal to use *Zbtb46-Cre* mice to confirm that indeed cDCs of the skin are the critical producers of CCL2 and IL-23. However, at this stage this would be very time consuming and costly to breed a new Cre line into the *c-Jun* colony and repeat all critical experiments, which in our opinion would only produce confirmatory results. Moreover, we used the *Mx1-Cre* mice to have a second independent line to confirm our results and exclude Cre-mediated artifacts. The reviewer is right that we need to show more convincingly that the effect is mediated exclusively by cDCs and not by macrophages and we will include additional experiments and controls to clearly demonstrate that the effects are mediated by cDCs and/or by macrophages. We strongly believe that with the present data and proposed additional experiments we can satisfactorily explain the contribution of each of these cell types to the observed phenotype. A previous study that used CD11c-diphtheria toxin receptor (DTR) mice found dermal DCs, but not macrophages to be depleted, resulting in reduced IMQ-induced *Il23a* mRNA expression (Riol-Blanco et al, 2014) (Fig. 3I & Extended Data Fig. 8b,c). We also found cDCs to be the most important source of *Il23a* mRNA, while other immune cells are dispensable (see EV Figure 3D). To further narrow

down the respective cell type within the CD11c+ skin cells we propose to sort DC subsets and macrophages from the skin and analyze CCL2 and IL-23 expression. Since we detect only partial deletion of *c-Jun* expression in *CD11c-Cre+* macrophages (EV Figure 1) this will clarify whether they contribute to the observed effects and at the same time a more detailed description of the responsive cDC subpopulations will be possible. If necessary we can repeat this experiment after deletion of *c-Jun* with *Mx-Cre* to show whether *CD11c+* macrophages still express CCL2 and IL-23. Lastly, in the manuscript we will also better discuss the limitations of our mouse model and point to future studies in *Zbtb46* mice for a definitive assessment.

2. In Figure 7, the authors want to translate their finding from the murine to the human system. Therefore, they perform immunofluorescence staining of skin sections of patients with psoriasis and stain for *c-Jun* as well as CD11c. However, CD11c is not a specific DC marker in humans as it is highly expressed on monocytes and macrophages. Furthermore, the images show that nearly all cells in the sections of the patients with psoriasis and of the healthy controls are *c-Jun+* positive and the CD11c+ cells represent only a minor fraction of the *c-Jun+* cells. Therefore, more evidence is needed that *c-Jun* in skin DCs is involved in IL-23 secretion. The authors should purify DCs from the skin analogously to the murine experiments.

We agree with the reviewer that CD11c is a broad marker for DCs also in the human skin. However, given the heterogeneity of resident and migrating DC subpopulations in inflamed human psoriatic skin, IF staining was the only way to detect most of them. Although it is a great suggestion, unfortunately, we do not have access to fresh skin to purify DCs and can therefore not perform the proposed experiments in human DCs at this stage. However, to better characterize the DC subpopulation responsible for IL23 and CCL2 production we plan to perform more detailed IF staining's on human psoriasis samples for DC subpopulations (CD1a, CD141) along with *c-Jun* and IL23p19 co-staining's. These experiments should clarify the reviewer's concern without the need of purifying DCs from human skin.

Minor points:

3. In the results part and in the figure legends of the manuscript, the authors always use the name of the substance, Imiquimod, but in the Figure itself the tradename, Aldara. They either should change Aldara into Imiquimod in the figures or also mention Aldara in the body of the text (in the current version of the manuscript it is only mentioned at the end of the manuscript in the material and method section).

Thank you for this comment. IMQ will be used throughout the manuscript.

4. In figure 2A, the authors show that after deletion of *c-Jun* in CD11c+ cells less pDCs are recruited to the skin. However, even in wildtype mice pDCs represent less than 1% of the cells. As IMQ-treatment also leads to the recruitment of other immune cells, it would be helpful to show total numbers of recruited pDCs to exclude a proportional effect.

Thank you for this comment. We will improve this and provide total numbers for pDC infiltration at the critical 12h time-point.

5. The authors claim that c-Jun in DCs is "necessary for the induction of various pro-inflammatory cytokines and key mediators of IMQ-induced skin inflammation (Fig. 4D-F)". However, except for IL-23p19 and IL-17 all shown cytokines and mediators are equally induced after 32h and only reduced after 48h. The authors should change the sentence accordingly and discuss, why c-Jun is only necessary at 48h but not 32h for some mediators (e.g. CXCL1).

This overstatement will be rephrased. We think that previously described TLR7-independent effects of IMQ (Drobits et al, 2012b; Walter et al, 2013) cause an up-regulation of some, primarily keratinocyte-related cytokines, like CXCL1. This will be added to the discussion.

6. The authors further claim that c-Jun is "essential for mRNA expression of the cytokines Il23p19, Il22, Il6 and the chemokines Ccl20 and Cxcl1 (Fig. EV3A)". However, Figure EV3A shows that all of those cyto- and chemokines are induced, especially in case of Il23p19, Il6 and Cxcl1. Therefore, the sentence should be changed as c-Jun is not "essential" but more a positive regulator.

Thank you for this comment. We will rephrase the sentence accordingly.

7. The material and methods section is in some parts short and does not contain all details necessary to repeat the experiments.

We apologize for brevity in the material and methods section which is due to limitations in space. We will however try to improve this section accordingly.

References:

Drobits B, Holcman M, Amberg N, Swiecki M, Grundtner R, Hammer M, Colonna M, Sibilio M (2012a) Imiquimod clears tumors in mice independent of adaptive immunity by converting pDCs into tumor-killing effector cells. *J Clin Invest* 122: 575-585

Drobits B, Holcman M, Amberg N, Swiecki M, Grundtner R, Hammer M, Colonna M, Sibilio M (2012b) Imiquimod clears tumors in mice independent of adaptive immunity by converting pDCs into tumor-killing effector cells. *The Journal of Clinical Investigation* 122: 575-585

Riol-Blanco L, Ordoas-Montanes J, Perro M, Naval E, Thiriout A, Alvarez D, Paust S, Wood JN, von Andrian UH (2014) Nociceptive sensory neurons drive interleukin-23-mediated psoriasiform skin inflammation. *Nature* 510: 157-161

Tortola L, Rosenwald E, Abel B, Blumberg H, Schäfer M, Coyle AJ, Renaud J-C, Werner S, Kisielow J, Kopf M (2012) Psoriasiform dermatitis is driven by IL-36-mediated DC-keratinocyte crosstalk. *The Journal of Clinical Investigation* 122: 3965-3976

Walter A, Schäfer M, Cecconi V, Matter C, Urosevic-Maiwald M, Belloni B, Schönewolf N, Dummer R, Bloch W, Werner S et al (2013) Aldara activates TLR7-independent immune defence. *Nature Communications* 4: 1560

A		Surface markers	Function	Disease associated
	LC	CD1a+ CD1c+ CD11b+ CD11c+ CD207+ FceR1+ (AD)	Production of: CCL2 CCL17 CCL22 CCL24 CXCL8 Attract Mo, Eos, T _H 2 Induce T _H 2 proliferation and cytokine production	AD
	CD1c DC	CD1a+ CD1c+ CD11c+	Induction of T _H 1 and T _H 17 cell proliferation and cytokine production	Psoriasis
	CD141 DC	CD11c+ CD141+ CLEC9A+ XCR1+	Induction of T _H 1 and T _H 17 cell proliferation and cytokine production	Unknown
				
B				
InfDC		CD1a+ CD11b+ CD11c+ CD206+ CD209+ FceR1+ (AD) IgE+ (AD)	Production of: CCL3 IL-16 IL-1 IL-12p70 IL-18 Induce T _H 1 proliferation and cytokine production	AD
		CD11c+ CD206+ CD14+/- CD163+/- CD209+/-	Production of: TNF α iNOS IL-12 IL-23	Psoriasis
		CD123+ BDCA-2+ ChemR23+ (Ps) FceR1+ (AD) CD11c-	Secretion of IFN α inducing activation/maturation of DDC	Psoriasis

20th May 2020

Decision on your manuscript EMM-2020-12409

Dear Maria,

Thank you for providing a detailed point-by-point rebuttal letter to the referees' comments. As mentioned in my previous correspondence, both referees who evaluated your manuscript, while acknowledging the potential interest of the findings, also raised substantial concerns on your work. Upon cross-commenting, they stated:

Referee #1:

"At a minimum this paper would need major overhaul and have to repeat a large number of their experiments to address the multiple shortcomings that the current manuscript has."

Referee #3:

"In consideration of the other reviewer's comments and the overall look again on the manuscript it becomes more clear that most of the experiments would need to be repeated. The information on the different DC subsets would be essential to have it published in EMBO Molecular Medicine."

Given the extent of the revisions required by the referees, I contacted you to ask if and how you would be willing to address these concerns. You kindly provided a rebuttal letter, and also stated that you would not breed into an additional Cre mouse or analyze sorted DC from human skin. I forwarded your letter to both referees. Referee #3 has not gotten back to me, but referee #1 replied:

"I still struggle with this manuscript and overall do not feel that it rises to the level of being appropriate for EMBO Molecular Medicine.

The authors overall response is to change their manuscript minimally and with limited number of repeat or new experiments to address all the shortcomings raised by both reviewers, which in my view is problematic - also there is no evidence for "cytotoxic" plasmacytoid dendritic cells in psoriasis (the disease has no features of cell killing or apoptosis), and the melanoma data provided in this manuscript in the context of psoriasis (again in the context of cell killing) makes little to no sense - the authors response to the critiques are not adequate."

I am afraid that given the overall negative feedback from both referees, I have no other choice but to return the manuscript to you at this point with the decision that we cannot offer to publish it.

I am really sorry that I could not bring better news this time.

With my best wishes,

Lise

Lise Roth
Editor
EMBO Molecular Medicine

***** Reviewer's comments *****

Referee #1 (Comments on Novelty/Model System for Author):

There are some discrepancies that need to be addressed - i.e. the designation of pDCs. Also, the background strains are not noted in the manuscript (Materials and Methods).

Referee #1 (Remarks for Author):

Comments on "Inflammation and anti-Immunity is promoted by c-Jun/AP-1-dependent CCL2 and IL-23 expression in Dendritic Cells" by Dr. Novoszel and colleagues. In this manuscript the authors demonstrate the role of c-Jun/AP-1 signaling in dendritic cells to IL-23 mediated inflammatory responses in the IMQ mouse model. The novel findings here are the role of C-jun in DCs and in driving pDC infiltration and IL-23 production by DCs. Other studies have shown dependence of IL-23 secretion by dendritic cells by the JNK pathway (Wang Q, Immunology 2011; Liu W, JBD 2009). I have the following comment.

Major

One aspect of this paper that makes it hard to follow is that some of the data shown here doesn't fit well together, at least not in the context of psoriasis. For example, the tumor growth kinetics of melanoma cells as shown in Figure 3 is certainly interesting but doesn't really fit well with the psoriasis angle of the rest of the manuscript. Also, the IMQ model is overused as a model for studying the pathogenesis of psoriasis and it is a self-limited acute model that is used to study a complex chronic inflammatory disease. The authors should consider revising and addressing this.

Results: Why was Mx1-Cre chosen? The authors state that it is "broadly inducible", which is vague. Mx1-Cre was induced by poly I:C that activates TLR3 signaling. Because these mice were also treated with IMQ to activate TLR7 signaling, is it possible to distinguish which TLR is responsible for the observed inflammatory response? Is it possible that a prior activation of TLR3 by poly I:C affects the inflammatory response to IMQ? It would help if the negative and positive controls are clearly stated. Are positive controls the Mx1-Cre-, c-Jun^{flx/flx} mice that were treated with poly I:C at first and then with IMQ? Or were the controls just treated with IMQ? When the authors simply state "c-Jun^{Δ/Δ} Mx1-Cre mice had significantly less IMQ induced skin inflammation", it is not exactly clear what they are being compared to. This also affects the BMDC experiments from these mice. Same applies to the tamoxifen-induced K5-Cre. Also, for all three c-Jun deletion models, was the deletion efficiency confirmed?

Why were Mx1-Cre⁺, and not CD11c-Cre⁺, mice used to isolate BMDC to look at the effect of IMQ on DC in vitro (Fig. 2C-2F)? What is the rationale for using either Mx1-Cre or CD11c-Cre for some experiments, but not the others (e.g. Fig. 3B-C vs. Fig G-L or Fig 5B vs. Fig. 5C)?

Figures 2A and 2H both look at pDCs by flow cytometry, but they designate pDCs differently (BST-2⁺ CD11c^{int}CD11b⁻ vs. B220⁺BST-2⁺CD11c^{int}CD11b⁻). How come? For some flow cytometry experiments (Fig. 1-2, Fig. 6K), an immune subpopulation that is being quantified is imputed as % of total live single cells. However, the authors use % of total CD45⁺ cell in other instances (Fig. 3, Fig. 6J). What is the rationale?

Figure 4B-C: In the mice that lack c-Jun in DCs, how can 15-40% of DCs be c-Jun+? Wouldn't flow cytometry be more suitable than IF for this experiment? Also, the figure caption for states "(C) Box and whiskers plot shows the percentage of c-Jun positive skin dendritic cells quantified by immunofluorescence performed as described in (C). (n= 5, 3 experiments)." Do the authors mean as described in B?

The CCL2 angle is interesting and the data with the pDCs, but how specific is this to pDCs? Did the authors look at the infiltration of other dendritic cell subsets in their model? Also, while pDCs have been implicated in psoriasis pathogenesis they seem to play the greatest role at the initiation or triggering of psoriatic plaques (Nestle FO, JEM 2005). Also, I'm not aware of pDCs having a cytotoxic role in psoriasis. Interestingly, the authors show that it is dispensable for IMQ-induced skin inflammation - this also makes these data not fit well in the context of psoriasis. Furthermore, the connection between CCL2 and psoriasis could be improved by showing increased CCL2 in psoriasis and co-localization with DCs in psoriatic skin.

Minor.

Figure 5H - refers to "epidermal area" - why did the authors decide to look at the "area" instead of "epidermal thickness", which is much more traditionally used (and would be consistent with data elsewhere in this manuscript).

Page 11 "IL-23 induces IL-17A production in $\gamma\delta$ T cells, which contributes to IMQ-induced inflammation (Zaba et al, 2007)" - don't think this is what is stated in this paper. I don't think it mentioned gamma-delta cells, and certainly nothing about IMQ-induced inflammation. The authors may want to look at this more closely.

The authors should provide more information on the background strain used....were all the strains done on the same genetic background?

In Methods, the authors talk about JunBflox/flox mice, but no data on these mice is shown in the results.

Intro: If epidermal deletion of c-Jun and JunB causes psoriasis, then what is the possible explanation for the opposite effect here (c-Jun deletion in DCs ameliorating IMQ-induced skin inflammation)?

Referee #3 (Comments on Novelty/Model System for Author):

1. Technical quality (including statistical analysis).

Overall, the experiments are well performed. However, more precise Cre-models (e.g. Zbtb46-Cre, XCR1-Cre) are available to clearly show that it is DC-mediated.

2. Strength of the evidence for the conclusions drawn.

There is clear evidence that c-Jun plays a role in TLR7-induced skin inflammation.

3. Novelty.

A new link of c-Jun with TLR7-induced skin inflammation.

4. Medical impact.

The study could lead to new treatment options for psoriasis by using small molecules against c-Jun.

5. Adequacy of model system.

As CD11c-Cre targets also monocytes and macrophages, other Cre-lines should be used. It would further strengthen the manuscript to identify, which DC subpopulation is responsible for the

observed effect.

Referee #3 (Remarks for Author):

In the presented study by Novoszel et al., the role of c-Jun in the pathogenesis of a psoriasis model as well as on anti-tumor responses was analyzed. The authors clearly show that deletion of c-Jun in CD11c⁺ cells leads to ameliorated disease in Imiquimod (IMQ)-induced skin inflammation but also to weaker anti-tumor immune responses after treatment of B16F10 tumors with IMQ. In the latter case, they identify reduced CCL2 secretion by BMDCs and mRNA expression by skin DCs as cause for reduced pDC recruitment to the tumor. In the IMQ-induced skin inflammation model, the authors identified reduced secretion of IL-23 after deletion of c-Jun in CD11c⁺ cells as cause for the ameliorated disease. They further showed that inhibition of c-Jun/AP-1 using small molecules can be used to treat the psoriasis-like disease. Overall, the study is well-performed and the results show clearly a role of c-Jun in TLR7-induced immune responses. However, in different parts of the manuscript the authors overestimate their results and should be more careful in the interpretation (see in the numbered comments below).

Major points:

1. In the manuscript, the authors state that c-Jun in DCs is necessary for IL-23 and CCL2 expression by using either c-Jun^{fl/fl} mice crossed with CD11c-Cre mice or BMDCs of Mx1-Cre mice crossed with c-Jun^{fl/fl} mice. However, CD11c is not exclusive for DCs but also expressed on macrophages, which is shown in Figure EV1. As the authors suggest that the secretion of CCL2 and IL-23 is mediated by cDCs, they should use the cDC-specific Zbtb46-Cre mice to avoid deletion of c-Jun in other cells than DCs. Further, the skin contains several subsets of cDCs with different functional properties. Therefore, it would strengthen the manuscript, if the authors identify whether all skin DC subsets respond to IMQ with secretion of CCL2 and IL-23 or only a certain subset.
2. In Figure 7, the authors want to translate their finding from the murine to the human system. Therefore, they perform immunofluorescence staining of skin sections of patients with psoriasis and stain for c-Jun as well as CD11c. However, CD11c is not a specific DC marker in humans as it is highly expressed on monocytes and macrophages. Furthermore, the images show that nearly all cells in the sections of the patients with psoriasis and of the healthy controls are c-Jun⁺ positive and the CD11c⁺ cells represent only a minor fraction of the c-Jun⁺ cells. Therefore, more evidence is needed that c-Jun in skin DCs is involved in IL-23 secretion. The authors should purify DCs from the skin analogously to the murine experiments.

Minor points:

3. In the results part and in the figure legends of the manuscript, the authors always use the name of the substance, Imiquimod, but in the Figure itself the tradename, Aldara. They either should change Aldara into Imiquimod in the figures or also mention Aldara in the body of the text (in the current version of the manuscript it is only mentioned at the end of the manuscript in the material and method section).
4. In figure 2A, the authors show that after deletion of c-Jun in CD11c⁺ cells less pDCs are recruited to the skin. However, even in wildtype mice pDCs represent less than 1% of the cells. As IMQ-treatment also leads to the recruitment of other immune cells, it would be helpful to show total numbers of recruited pDCs to exclude a proportional effect.
5. The authors claim that c-Jun in DCs is "necessary for the induction of various pro-inflammatory cytokines and key mediators of IMQ-induced skin inflammation (Fig. 4D-F)". However, except for IL-23p19 and IL-17 all shown cytokines and mediators are equally induced after 32h and only reduced after 48h. The authors should change the sentence accordingly and discuss, why c-Jun is only necessary at 48h but not 32h for some mediators (e.g. CXCL1).
6. The authors further claim that c-Jun is "essential for mRNA expression of the cytokines Il23p19,

Il22, Il6 and the chemokines Ccl20 and Cxcl1 (Fig. EV3A)". However, Figure EV3A shows that all of those cyto- and chemokines are induced, especially in case of Il23p19, Il6 and Cxcl1. Therefore, the sentence should be changed as c-Jun is not "essential" but more a positive regulator.

7. The material and methods section is in some parts short and does not contain all details necessary to repeat the experiments.

As a service to authors, EMBO provides authors with the possibility to transfer a manuscript that one journal cannot offer to publish to another EMBO publication. The full manuscript and if applicable, reviewers reports are automatically sent to the receiving journal to allow for fast handling and a prompt decision on your manuscript. For more details of this service, and to transfer your manuscript to another EMBO title please click on Link Not Available

Vienna, 03-Jun-2020

Dear Lise,

Thank you for having considered our manuscript for publication in EMBO Molecular Medicine. We regret that after the review process and our point-by-point response you came to the conclusion that our manuscript does not fulfill the criteria for publication due to the negative comments of the reviewers. Herewith we would like to appeal and ask you to reconsider your decision since we believe that it is based on incomplete information and misunderstandings, maybe also due to the special situation caused by the Corona pandemic and the limited feedback of some of the reviewers.

One major drawback for our revision was the lack of return by reviewer #2 and therefore we would kindly ask you to seek additional independent opinions from experts to evaluate our manuscript.

Concerning the two reviews that were already obtained, we would like to add the following in addition to what was already mentioned in the detailed point-by-point response that we submitted earlier and are including in the current appeal submission.

Reviewer #1 raised a number of points which indicated that she/he had not read the entire manuscript thoroughly. Most importantly, the reviewer insisted on a second mouse model in which all experiments should be repeated although such a second model is provided in our manuscript. This mouse model (*c-Jun/JunB*^{Δ/Δ} K5-Cre ER^{T2} mice) is a well-accepted mouse model of psoriasis where the hallmarks of human psoriasis are recapitulated, the important role of pDC has been confirmed, and the efficiency of several classical human treatments (e.g. IL23-IL23R blockade) have been demonstrated^{1,2,3}. We even provide a third model by stimulating with LL-37-RNA, which is a key factor in driving human psoriasis, to confirm the mechanism of action *in vitro*. We doubt that this reviewer has carefully evaluated these data as he/she didn't even mention them in his/her revision or reply to our point by point response nor did he/she present arguments that would explain the necessity of an additional, third *in vivo* model. Although there are a number of mouse models for psoriasis available (e.g. K5.Stat3C, K5.Tie2, K5.TGF-β1, K14-Areg) none of them completely recapitulates human psoriasis. We have compared two models, one where the psoriatic phenotype is caused by alterations in the epidermis (*c-Jun/JunB*^{Δ/Δ} K5-Cre ER^{T2} mice) and the other where the phenotype is caused by activation of immune cells (IMQ), both of which recapitulate hallmarks of human psoriasis. With both models *in vivo* and an additional one *in vitro* (LL-37-RNA) our results led to the same conclusion.

Other major points this reviewer has raised were already clearly explained in the manuscript or in our point by point response: i) The use of *Mx-Cre* mice was necessary to reach complete recombination of the *c-Jun*^{fl/fl} allele *in vitro*. Those data on recombination efficiency that were lacking will be provided. ii) There was no effect of TLR3 stimulation by poly I: C or Tamoxifen treatment on IMQ-induced skin inflammation (Figure 1B-D, epidermal thickness in control mice). iii) The reviewer suggested flow cytometry instead of IF to confirm deletion of *c-Jun* in the inflamed skin. While we were not able to show *c-Jun* staining's by intracellular flow cytometry we did provide RNA levels of cDCs sorted from IMQ-treated back skin in Figure 4A which clearly shows reduction of mRNA and thus efficient deletion of *c-Jun* in cDCs in our model. iv) Finally, the reviewer discusses our

CCL2 data which he finds interesting but not put in the right context. We explained that CCL2 indeed mainly affects recruitment of pDC but no other DC subpopulations (Figure EV1 G-I). To clarify whether our findings on *c-Jun* dependent CCL2 production also holds true in human psoriasis we are going to perform multicolor immunofluorescence staining's including DC markers, *c-Jun* and CCL2 on human skin sections from psoriasis patients.

Thus, we can adequately address most of the "major shortcomings" reviewer #1 has mentioned. One point remaining to be discussed is whether our results on the role of *c-Jun* in the anti-tumor effect of IMQ should be part of this manuscript. Reviewer #3 did not mention these data as being not fitting into the manuscript. However, given the fact that in the reply to our point-by-point response reviewer #1 again insisted on removing the tumor data, we can modify the manuscript accordingly focusing on the role of CCL2 and IL-23 in psoriasis. We have no objections in removing these results should a third independent reviewer reach the same conclusion.

Another misunderstanding of this reviewer relates to "cytotoxic pDCs" in psoriasis. We never ever postulated something like this in the whole paper and do not understand why the reviewer even mentions this. For us this is another clear indication that the reviewer did not carefully read our paper.

Reviewer #3 mainly criticizes the *Cre* lines used in our experiments which do not allow for the correct characterization of the different DC subsets in the skin that are responsible for the observed effects of *c-Jun* deletion on CCL2 and IL-23. The reviewer is right, but at the time we started this project, these were the best *Cre* lines available for our study. We offered to perform additional experiments to clarify the issues raised by this reviewer, but unfortunately this reviewer did not reply to our point by point response. As mentioned in our detailed response, we are going to perform cell sorting from IMQ-treated skin for cDC1, cDC2 and macrophages to identify in which of these cells *Ccl2* and *Il-23* are expressed and how this is affected by deletion of *c-Jun*. Using this technology we are able to give a specific answer to the main question of this reviewer.

The reviewer should therefore give a plausible scientific explanation on the added value and the conceptual advance of crossing into another *Cre*-line, which will take another 2 years at least. Repeating all the experiments with a third *Cre* line would in our opinion likely provide just confirmatory results

at the best. Moreover, we currently do not have these mice and importing them will turn out to be difficult given the current Corona crisis.

The second major criticism of this reviewer was the use of CD11c as a marker for DCs in human skin IF stainings. We agree with the reviewer that CD11c is a rather unspecific marker, however, it is the only one present on most DC subsets in inflamed human skin. Compared to healthy human skin psoriatic skin contains a number of infiltrating DC subpopulations with different marker expression. But how can we sort fresh human DCs from psoriatic skin and repeat the experiments we did in mice? It is impossible to get access to fresh human skin for the isolation of different DC subtypes in sufficient amounts to be able to perform FACS analysis for intracellular molecules etc., as big pieces of human skin would be required. Although certainly interesting, this is asking for something impossible to do and requiring single cell RNA analysis which in our opinion goes beyond the scope of this manuscript.

To address the question which human DC subtype expresses c-Jun, CCL2 and/or IL-23 we are therefore going to perform multicolor staining's on patient samples from psoriatic skin with established markers for human cDCs such as CD1a, CD1c, CD14 and CD141 and Xcr1 for resident but also FcER1 and iNOS for infiltrating DC subpopulations ^{4,5}.

We are confident that with these experiments we can address this reviewer's concern and we can convincingly specify the DC subpopulation responsible for our observed phenotype in human psoriasis and the corresponding mouse model.

We hope that our arguments can persuade you to reconsider your decision.

Many thanks for your consideration and best wishes

Maria

References:

1. Zenz, R., *et al.* Psoriasis-like skin disease and arthritis caused by inducible epidermal deletion of Jun proteins. *Nature* **437**, 369 (2005).
2. Glitzner, E., *et al.* Specific roles for dendritic cell subsets during initiation and progression of psoriasis. *EMBO Mol Med* **6**, 1312-1327 (2014).
3. Gago-Lopez, N., *et al.* Role of bulge epidermal stem cells and TSLP signaling in psoriasis. *EMBO Molecular Medicine* **11**, e10697 (2019).
4. Collin M, Bigley V. Human dendritic cell subsets: an update. *Immunology*. 2018 May;154(1):3-20. doi: 10.1111/imm.12888. Epub 2018 Feb 27. PMID: 29313948
5. Haniffa M, Gunawan M, Jardine L. Human skin dendritic cells in health and disease. *J Dermatol Sci*. 2015 Feb;77(2):85-92. doi: 10.1016/j.jdermsci.2014.08.012. Epub 2014 Sep 10. PMID: 25301671

17th Jun 2020

Dear Maria,

Thank you for your appeal asking us to reconsider our decision on your manuscript.

I have carefully read your letter and as discussed with you, communicated your points with two external advisers.

The first adviser stated:

"In general, it is a good manuscript with some interesting information. It is a bit "oversold" and previous data on AP-1 regulation of IL-23 should be cited (see reviewer comments). Most importantly, some of the effects are not robust (see for example data in Fig. 1G-I, where only a few mice with higher numbers of immune cells are responsible for the statistical significance. This high variability may result from the mixed genetic background (which is indeed not defined...). The authors should at least discuss these limitations. The number of replicates for some in vivo experiments is also problematic. In particular, all experiments shown in Fig. EV4 were performed with only three mice. This is clearly insufficient, in particular since one mutant mouse does not even show a phenotype. The n number would have to be increased to at least 6-8.

In addition, some important controls are missing. For example, JNKi control in CD11-Cre-Junfl/fl mice (Fig. 6D). This is an important control and would show the specificity of the inhibitor.

Overall, I agree with various criticisms of the reviewers [...]. However, I do not think that the request to use another Cre mouse is justified - this would be a completely new story and the identification of the responsible DC subset would not add much to the overall message. In addition, it is indeed not possible to get large amounts of fresh psoriatic skin for ethical reasons.

Overall, it is a borderline case, but I think that a revision is justified. Clearly, such a revision would be rather extensive, but it should be possible."

Detailed comments from this adviser on the rebuttal letter:

1/ Reviewer #1:

Comments on novelty/model system:

- The mixed background is mentioned, but not which mixture. In general mixed backgrounds are problematic for immunological experiments and this may explain the high variability of some of the data (.e.g. 1G.I). This problem should at least be discussed.
- Previous publications reporting the dependence of IL-23 secretion by dendritic cells by the JNK pathway should be cited.

Major point #1:

- The tumor data can be kept in the manuscript. The issue can be solved by explaining in details why tumor experiments were performed.
- Only three mice per genotype were analysed in the additional mouse model of psoriasis. The n number should be increased.

Major point #2:

- The cells in which the deletion occurred were not shown and the efficacy of knockout at the protein level is not shown for any of the mouse models. This should be included in a revision.

Regarding the different controls, the rebuttal letter gives a good explanation, but all controls should be explicitly mentioned in the manuscript

Major point #3:

- Clarify in each experiment why a particular mouse model was used. The Mx-Cre model is inducible and therefore, is not suitable to study development/maturation of pDCs (Fig. EV2A-C).

Minor comments, last point:

- The reviewer did not understand this point and therefore, the authors should more explicitly point to the different roles of AP1 family members in keratinocytes vs DCs in the context of psoriasis. However, this finding indeed complicates the treatment and I am not convinced that JNK inhibition will solve the problem. This should be more carefully discussed.

Reviewer #3:

Major point 1:

- The use of another Cre mouse is not essential. This would require repetition of a large number of mouse experiments and the additional info would be limited. However, the limitations of the approach should be clearly mentioned and the proposed in vitro experiments should be performed.

Minor point:

The material and methods section should be expanded.

The second advisor agreed that additional Cre mouse was not necessary.

Therefore, after internal discussions about your manuscript with my colleagues, I would be happy to reconsider my decision and invite revision of the manuscript.

Addressing the reviewers' concerns as indicated in your rebuttal letter, as well as the comments indicated by the first adviser will be necessary for further considering the manuscript in our journal. Additional Cre mouse and analysis of sorted DC from human skin will not be required.

EMBO Molecular Medicine encourages a single round of revision only and therefore, acceptance or rejection of the manuscript will depend on the completeness of your responses included in the next, final version of the manuscript.

Please also contact us as soon as possible if similar work is published elsewhere. If other work is published, we may not be able to extend the revision period beyond three months.

I look forward to receiving your revised manuscript.

With my best wishes,

Lise

Lise Roth, PhD

Editor

EMBO Molecular Medicine

*** Instructions to submit your revised manuscript ***

** PLEASE NOTE ** As part of the EMBO Publications transparent editorial process initiative (see our Editorial at <https://www.embopress.org/doi/pdf/10.1002/emmm.201000094>), EMBO Molecular Medicine will publish online a Review Process File to accompany accepted manuscripts.

To submit your manuscript, please follow this link:

- 1) a .docx formatted version of the manuscript text (including Figure legends and tables). Please make sure that the changes are highlighted to be clearly visible to referees and editors alike.
- 2) separate figure files*
- 3) supplemental information as Expanded View and/or Appendix. Please carefully check the authors guidelines for formatting Expanded view and Appendix figures and tables at <https://www.embopress.org/page/journal/17574684/authorguide#expandedview>
- 4) a letter INCLUDING the reviewers' reports and your detailed responses to their comments (as Word file)

Also, and to save some time should your paper be accepted, please read below for additional information regarding some features of our research articles:

- 5) The paper explained: EMBO Molecular Medicine articles are accompanied by a summary of the articles to emphasize the major findings in the paper and their medical implications for the non-specialist reader. Please provide a draft summary of your article highlighting
 - the medical issue you are addressing,
 - the results obtained and
 - their clinical impact.

6) For more information: There is space at the end of each article to list relevant web links for further consultation by our readers. Could you identify some relevant ones and provide such information as well? Some examples are patient associations, relevant databases, OMIM/proteins/genes links, author's websites, etc...

7) Author contributions: the contribution of every author must be detailed in a separate section (before the acknowledgments).

8) EMBO Molecular Medicine now requires a complete author checklist (<https://www.embopress.org/page/journal/17574684/authorguide>) to be submitted with all revised manuscripts. Please use the checklist as a guideline for the sort of information we need WITHIN the manuscript as well as in the checklist. This is particularly important for animal reporting, antibody dilutions (missing) and exact p-values and n that should be indicated instead of a range.

9) Every published paper now includes a 'Synopsis' to further enhance discoverability. Synopses are displayed on the journal webpage and are freely accessible to all readers. They include a short stand first (maximum of 300 characters, including space) as well as 2-5 one sentence bullet points that summarise the paper. Please write the bullet points to summarise the key NEW findings. They should be designed to be complementary to the abstract - i.e. not repeat the same text. We encourage inclusion of key acronyms and quantitative information (maximum of 30 words / bullet point). Please use the passive voice. Please attach these in a separate file or send them by email, we will incorporate them accordingly.

You are also welcome to suggest a striking image or visual abstract to illustrate your article. If you do please provide a jpeg file 550 px-wide x 400-px high.

10) A Conflict of Interest statement should be provided in the main text

11) Please note that we now mandate that all corresponding authors list an ORCID digital identifier. This takes <90 seconds to complete. We encourage all authors to supply an ORCID identifier, which will be linked to their name for unambiguous name identification.

Currently, our records indicate that the ORCID for your account is 0000-0001-6129-5613.

Link Not Available

12) The system will prompt you to fill in your funding and payment information. This will allow Wiley to send you a quote for the article processing charge (APC) in case of acceptance. This quote takes into account any reduction or fee waivers that you may be eligible for. Authors do not need to pay any fees before their manuscript is accepted and transferred to our publisher.

Photos 400-800 DPI

*Additional important information regarding figures and illustrations can be found at <http://bit.ly/EMBOPressFigurePreparationGuideline>

Reviewer #1:

Comments on novelty/model system:

There are some discrepancies that need to be addressed - i.e. the designation of pDCs. Also, the background strains are not noted in the manuscript (Materials and Methods).

Remarks for author:

Comments on "Inflammation and anti-Immunity is promoted by c-Jun/AP-1-dependent CCL2 and IL-23 expression in Dendritic Cells" by Dr. Novoszel and colleagues. In this manuscript the authors demonstrate the role of c-Jun/AP-1 signaling in dendritic cells to IL-23 mediated inflammatory responses in the IMQ mouse model. The novel findings here are the role of C-jun in DCs and in driving pDC infiltration and IL-23 production by DCs. Other studies have shown dependence of IL-23 secretion by dendritic cells by the JNK pathway (Wang Q, Immunology 2011; Liu W, JBD 2009). I have the following comment.

Major:

One aspect of this paper that makes it hard to follow is that some of the data shown here doesn't fit well together, at least not in the context of psoriasis. For example, the tumor growth kinetics of melanoma cells as shown in Figure 3 is certainly interesting but doesn't really fit well with the psoriasis angle of the rest of the manuscript. Also, the IMQ model is overused as a model for studying the pathogenesis of psoriasis and it is a self-limited acute model that is used to study a complex chronic inflammatory disease. The authors should consider revising and addressing this.

We thank the reviewer for constructive criticisms of our manuscript. In principle, we agree that the tumor results might not seem to fit well to the psoriasis data. However, with these results we wanted to strengthen our findings on the regulation of CCL2 by c-Jun in skin DCs. We find it intriguing that lack of CCL2 does not affect the IMQ induced psoriatic phenotype, whereas the anti-tumor function of IMQ in the skin is impaired due to impaired recruitment of pDCs to the skin. We know from previous studies with the B16-F10 melanoma model that CCL2 is required for efficient IMQ-induced recruitment of pDCs to the tumor site in the skin, where they acquire a tumor killing capacity (Drobits et. al JCI 2012). With this experiment we can show that in the absence of c-Jun in DCs, pDCs are not recruited to the tumor site upon IMQ treatment because of reduced CCL2 production. However, pDCs still maintain their killing capacity, as shown by the *in vitro* killing assays, demonstrating that this is independent of c-Jun expression. We now discuss these aspects in more detail in the new version of the manuscript and have removed the "anti-tumor" aspect from the title of the paper.

We agree with the reviewer that the commonly used IMQ model has limitations with regard to the pathogenesis of psoriasis. Therefore, we have also employed an additional mouse model of psoriasis based on the inducible genetic deletion of c-Jun and JunB in the epidermis (*c-Jun/JunB*^{ΔΔ} *K5-Cre ER^{T2}*). In this model lack of c-Jun and JunB in keratinocytes induces skin inflammation resembling psoriasis that is also dependent on IL23R signaling (Glitzner et al. Kopp et al). We have increased the number of animals in this experiment and we can show also in

this model that treatment with JNK inhibitor ameliorates the psoriasis phenotype emphasizing the translational potential of our therapeutic findings (Figures 6 I-N).

Why was Mx1-Cre chosen? The authors state that it is "broadly inducible", which is vague. Mx1-Cre was induced by poly I: C that activates TLR3 signaling. Because these mice were also treated with IMQ to activate TLR7 signaling, is it possible to distinguish which TLR is responsible for the observed inflammatory response? Is it possible that a prior activation of TLR3 by poly I: C affects the inflammatory response to IMQ? It would help if the negative and positive controls are clearly stated. Are positive controls the Mx1-Cre-, c-Jun flox/flox mice that were treated with poly I: C at first and then with IMQ? Or were the controls just treated with IMQ? When the authors simply state "c-Jun Δ/Δ Mx1-Cre mice had significantly less IMQ induced skin inflammation", it is not exactly clear what they are being compared to. This also affects the BMDC experiments from these mice. Same applies to the tamoxifen-induced K5-Cre. Also, for all three c-Jun deletion models, was the deletion efficiency confirmed?

We initially employed the *Mx1-Cre* line for the broad inducible deletion of c-Jun in interferon-responsive skin cells, such as keratinocytes and hematopoietic cells, and to identify thereafter, which cell-type is specifically responsible for the observed phenotype, immune cells or parenchymal cells (Fig. 1A-C). We also kept it as a second, independent Cre line for deletion in hematopoietic cells to confirm the results shown with the *CD11c-Cre* line and exclude Cre-mediated artifacts. In Fig EV1A we now improved the schematic representation of the treatment schemes for the different mouse models. *c-Jun*^{fl/fl} mice were used as controls and were treated equally to Cre+ mice in all experiments. We also indicated poly I: C (bronze) or Tamoxifen (green) treated control or knock-out mice with a unique color-code throughout the manuscript to clearly indicate any treatment in all experiments performed. Additionally, we expanded the methods section to describe in detail our protocol to induce deletion, which includes a section on the treatment of control groups.

Our data in Fig. 1 A-C argue against an effect of prior TLR3 activation (poly I: C treatment) during the course of the inflammatory response. First, the last injection of poly I: C takes place at least 9 days before IMQ treatment starts (Fig. EV1A). By this time, there is no residual inflammatory signature detectable. Second, IMQ-induced epidermal thickening is comparable between untreated, poly I: C or Tamoxifen treated *c-Jun*^{fl/fl} mice (Figure 1A-C). Following the same strategy, BMDC and BM-pDC were generated from mice treated with poly I: C *in vivo* and bone marrow was harvested earliest 9 days later.

We are sorry for having omitted to mention the deletion efficiency in the initial version of our manuscript, as this was previously published several times. Figure EV1 was revised to include protein analyses on sections showing c-Jun deletion in the various conditional knock-out mouse models employed (Figure EV1B-D).

Why were Mx1-Cre+, and not CD11c-Cre+, mice used to isolate BMDC to look at the effect of IMQ on DC *in vitro* (Fig. 2C-2F)? What is the rationale for using either Mx1-Cre or CD11c-Cre for some experiments, but not the others (e.g. Fig. 3B-C vs. Fig G-L or Fig 5B vs. Fig. 5C)?

We tried to strengthen our point by combining *in vitro* and *in vivo* findings and different Cre mouse models to exclude Cre-mediated artifacts. We used *Mx1-Cre* mice mainly for *in vitro* experiments because of the higher

recombination efficiency of this *Cre* line in BMDCs compared to *CD11c-Cre* mice, whereas *ex vivo* isolated cells were taken from *CD11c Cre* mice. We mention this now in a sentence in the main text for Fig 2. *In vivo*, *Mx1-Cre* mice were used only for the experiments shown in Fig. 1 A.

Figures 2A and 2H both look at pDCs by flow cytometry, but they designate pDCs differently (BST-2+ *CD11c*^{int} *CD11b*⁻ vs. B220+ BST-2+ *CD11c*^{int} *CD11b*⁻). How come? For some flow cytometry experiments (Fig. 1-2, Fig. 6K), an immune subpopulation that is being quantified is imputed as % of total live single cells. However, the authors use % of total *CD45*⁺ cell in other instances (Fig. 3, Fig. 6J). What is the rationale?

pDCs were uniformly identified as being BST-2⁺B220⁺*CD11b*⁻*CD11c*^{int}. We apologize for the labeling error in Figure Legend 2A, this was corrected and a representative flow cytometry plot was added (Fig. 2B).

We wanted to emphasize that changes in the immune infiltrate are reflected by changes among total live, single cells. For example the former Fig. 6J (now Fig. EV3E, F) did show the actual numbers of the plot, which was gated on *CD45*, while the old Fig. 6K (now new Fig. 6 F-H) showed % of live cells to point out that this is also reflected by the total number of the respective cell population in the skin. Both revealed significant differences between control and knock-out.

We agree, with the Reviewer that it is more concise to stick with one quantification method, which is now represented as % of total live, single cells throughout the manuscript. Furthermore, Fig. 6J has been replaced by Fig EV3E-F and Fig. 6K was replaced by Fig. 6F-H.

Figure 4B-C: In the mice that lack *c-Jun* in DCs, how can 15-40% of DCs be *c-Jun*⁺? Wouldn't flow cytometry be more suitable than IF for this experiment? Also, the figure caption for states "(C) Box and whiskers plot shows the percentage of *c-Jun* positive skin dendritic cells quantified by immunofluorescence performed as described in (C). (n= 5, 3 experiments)." Do the authors mean as described in B?

Thank you for this comment. Although flow cytometry would be a very suitable experimental approach, we were not able to establish a convincing intracellular flow cytometry staining of *c-Jun* from *ex vivo* isolated skin DC. We thus performed IF. The residual *c-Jun* expression (15-40 %) reflects the incomplete deletion in *CD11c*⁺ cells under inflammatory conditions, where *CD11c* levels are reduced and thus *Cre* expression is low. We have now included *c-Jun* expression levels from sorted skin DC and macrophage populations, which shows that in *cDC2* subsets isolated from *c-Jun*^{fl/fl} *CD11c-Cre* mice, *c-Jun* expression is significantly reduced but still detectable in some samples compared to *c-Jun*^{fl/fl} cells. It also shows that macrophages still express *c-Jun* at similar levels as controls, including *CD11c* expressing macrophages from *c-Jun*^{fl/fl} *CD11c-Cre* where *Jun* deletion did not seem to occur at all (Fig 5 C, E). We are sorry for the labelling error and have corrected it accordingly.

The *CCL2* angle is interesting and the data with the pDCs, but how specific is this to pDCs? Did the authors look at the infiltration of other dendritic cell subsets in their model? Also, while pDCs have been implicated in psoriasis pathogenesis they seem to play the greatest role at the initiation or triggering of psoriatic plaques (Nestle FO, JEM 2005). Also, I'm not aware of pDCs having a cytotoxic role in psoriasis. Interestingly, the authors show that it is dispensable for IMQ-induced skin inflammation - this also makes these data not fit well in the

context of psoriasis. Furthermore, the connection between CCL2 and psoriasis could be improved by showing increased CCL2 in psoriasis and co-localization with DCs in psoriatic skin.

We appreciate that the reviewer values our findings on the regulation of CCL2 in DCs by c-Jun and its role for the recruitment of pDC. As shown in Figure EV 1 G-I the infiltration of other dendritic cell subsets (CD103⁺ cDC1 and CD11b⁺ cDC2) is not affected. As the Reviewer points out pDC stimulation by viral TLR agonists (TLR-7/9) has been shown to trigger psoriatic plaques. We therefore think that the data we present, showing a role of c-Jun in pDCs downstream of TLR7, has relevance for a better understanding of the function of pDCs in psoriasis. We agree, a cytotoxic function of pDCs has not been shown in psoriasis and it was not our initial intention to propose this. It is however an interesting idea and cytotoxic molecules, like Trail produced by CD11c⁺ cells have been shown to promote psoriasis (Zaba et al, 2010). Thus, cytotoxic effects of pDCs could indeed influence the pathogenesis of psoriasis. While we show that CCL2 contributes to the psoriatic inflammatory infiltrate by controlling pDC and monocyte-macrophage recruitment (see Appendix Fig 3D), we demonstrate that epidermal thickening, an adverse event of IMQ treatment, is not affected in CCL2 knockout mice (Fig 3F). Because these therapeutic implications can only be understood in context (inflammation & tumor data), we consider it important to present these tumor data in our manuscript. We discuss this now in more detail in the new version of the manuscript.

To further strengthen the connection between CCL2 and psoriasis we analyzed a published RNA-Seq Dataset (GSE121212) to show that *Ccl2* expression is up-regulated in psoriatic lesions and that its expression strongly correlates with that of *c-Jun*. This new data are shown in (Fig EV4B-D). Finally, we also show CCL2 protein expression in c-Jun⁺ DCs by immunofluorescence (Fig 7D, Fig EV4E).

Minor:

Figure 5H - refers to "epidermal area" - why did the authors decide to look at the "area" instead of "epidermal thickness", which is much more traditionally used (and would be consistent with data elsewhere in this manuscript).

We wanted to show that different methods of analyzing skin thickening give the same result, namely reduced thickening in *c-Jun*^{Δ/Δ} *CD11c*-Cre mice, but agree with the Reviewer that for reasons of uniformity it is better displayed as epidermal thickness as shown in the new Fig 4L.

Page 11 "IL-23 induces IL-17A production in γδ T cells, which contributes to IMQ-induced inflammation (Zaba et al, 2007)" - don't think this is what is stated in this paper. I don't think it mentioned gamma-delta cells, and certainly nothing about IMQ-induced inflammation. The authors may want to look at this more closely.

Thank you for pointing out this mistake. We cited the wrong paper and replaced the reference by the correct one (Tortola et al, 2012).

The authors should provide more information on the background strain used....were all the strains done on the same genetic background?

All mice were in a mixed background as stated in the materials and methods section.

In Methods, the authors talk about JunB flox/flox mice, but no data on these mice is shown in the results.

Sorry this was a typing mistake and has been corrected.

Intro: If epidermal deletion of c-Jun and JunB causes psoriasis, then what is the possible explanation for the opposite effect here (c-Jun deletion in DCs ameliorating IMQ-induced skin inflammation)?

Thank you for this comment. We believe that one of the major findings of this study is the apparent functional complexity and diverse cell-context dependent roles of Jun proteins in different cells of the skin (as mentioned in the discussion). Keratinocytes, which are neither TLR7 responsive (Drobits et al, 2012) (Supplemental Figure 1A) nor important producers of IL-23 in our model (see Figure EV 2F), contribute functionally differently to skin homeostasis than DCs. Furthermore, while there seems to be redundancy for Jun proteins in keratinocytes (psoriatic disease is only induced if both *c-Jun* and *JunB* are deleted in keratinocytes), our results here clearly demonstrate that this is not the case for DCs, where c-Jun deletion is sufficient to ameliorate inflammation. This offers a therapeutic window for the treatment of psoriasis with topical JNK inhibitors, which should only inhibit c-Jun and not *JunB* as the latter does not seem to get phosphorylated by JNK. The observed opposite outcome also reflects the unique cell-specific TLR expression pattern and cytokine expression profile of epidermal vs. immune cells driving or ameliorating skin inflammation. We mention this now more detailed in the discussion.

Reviewer #3:

Comments on novelty/model system:

Technical quality (including statistical analysis).

Overall, the experiments are well performed. However, more precise Cre-models (e.g. Zbtb46-Cre, XCR1-Cre) are available to clearly show that it is DC-mediated.

Strength of the evidence for the conclusions drawn.

There is clear evidence that c-Jun plays a role in TLR7-induced skin inflammation.

Novelty.

A new link of c-Jun with TLR7-induced skin inflammation.

Remarks for author:

In the presented study by Novoszel et al., the role of c-Jun in the pathogenesis of a psoriasis model as well as on anti-tumor responses was analyzed. The authors clearly show that deletion of c-Jun in CD11c+ cells leads to ameliorated disease in Imiquimod (IMQ)-induced skin inflammation but also to weaker anti-tumor immune responses after treatment of B16F10 tumors with IMQ. In the latter case, they identify reduced CCL2 secretion by BMDCs and mRNA expression by skin DCs as cause for reduced pDC recruitment to the tumor. In the IMQ-induced skin inflammation model, the authors identified reduced secretion of IL-23 after deletion of c-Jun in CD11c+ cells as cause for the ameliorated disease. They further showed that inhibition of c-Jun/AP-1 using small molecules can be used to treat the psoriasis-like disease. Overall, the study is well-performed and the results

show clearly a role of c-Jun in TLR7-induced immune responses. However, in different parts of the manuscript the authors overestimate their results and should be more careful in the interpretation (see in the numbered comments below).

Major points:

1. In the manuscript, the author's state that c-Jun in DCs is necessary for IL-23 and CCL2 expression by using either c-Junfl/fl mice crossed with CD11c-Cre mice or BMDCs of Mx1-Cre mice crossed with c-Junfl/fl mice. However, CD11c is not exclusive for DCs but also expressed on macrophages, which is shown in Figure EV1. As the authors suggest that the secretion of CCL2 and IL-23 is mediated by cDCs, they should use the cDC-specific Zbtb46-Cre mice to avoid deletion of c-Jun in other cells than DCs. Further, the skin contains several subsets of cDCs with different functional properties. Therefore, it would strengthen the manuscript, if the authors identify whether all skin DC subsets respond to IMQ with secretion of CCL2 and IL-23 or only a certain subset.

We appreciate the positive feedback on the manuscript and the valuable comments on how to improve the manuscript. We have performed the majority of *in vivo* experiments with *CD11c-Cre* mice for two reasons:

- a) It deletes in all DC populations that we are interested in (cDCs and importantly pDCs).
- b) When we started our experiments this was the best Cre line available for DC-specific deletion.

We agree with the reviewer that it would be ideal to use *Zbtb46-Cre* mice to confirm that indeed cDCs of the skin are the critical producers of CCL2 and IL-23. However, at this stage it would have been very time consuming and costly to breed a new Cre line into the c-Jun colony and repeat all critical experiments, which in our opinion would only produce confirmatory results.

A previous study that used *CD11c-Diphtheria Toxin Receptor (DTR)* mice found dermal DCs, but not macrophages to be depleted, resulting in reduced IMQ-induced *Il23a* mRNA expression (Riol-Blanco et al, 2014) (Fig. 3I & Extended Data Fig. 8b,c). We also found cDCs to be the most important source of *Il23a* mRNA, while other immune cells are dispensable (see EV Figure 2F). To further narrow down the respective cell type within the CD11c⁺ skin cells, as suggested by this Reviewer, we sorted DC subsets (cDC1, cDC2, Langerhans cells) and macrophages from the skin and analyzed *c-Jun*, *Ccl2* and *Il23p19* expression by quantitative RT-PCR. Among the sorted immune cell populations cDC2 showed the strongest induction of *Ccl2* and *Il23p19* after IMQ application, and expression was significantly reduced when *c-Jun* was deleted (new Fig 5A). In contrast, total macrophages or CD11c expressing macrophages sorted from *c-Jun*^{Δ/Δ} *CD11c-Cre* mice retained *c-Jun*, *Ccl2* and *Il23p19* expression (new Fig 5B-E). These results indicate that the observed effects are not mediated by macrophages, but by loss of *c-Jun* in the cDC2 subset.

2. In Figure 7, the authors want to translate their finding from the murine to the human system. Therefore, they perform immunofluorescence staining of skin sections of patients with psoriasis and stain for c-Jun as well as CD11c. However, CD11c is not a specific DC marker in humans as it is highly expressed on monocytes and macrophages. Furthermore, the images show that nearly all cells in the sections of the patients with psoriasis and of the healthy controls are c-Jun⁺ positive and the CD11c⁺ cells represent only a minor fraction of the c-

Jun⁺ cells. Therefore, more evidence is needed that c-Jun in skin DCs is involved in IL-23 secretion. The authors should purify DCs from the skin analogously to the murine experiments.

We agree with the reviewer that CD11c is a broad marker for DCs also in the human skin. However, given the heterogeneity of resident and migrating DC subpopulations in inflamed human psoriatic skin, IF staining for CD11c was the only way to detect most of them (Boltjes & van Wijk, 2014).

It is a great suggestion to purify human DCs from the skin analogously to the mouse. Unfortunately, we do not have access to fresh skin from psoriasis patients to purify DCs and were therefore not able to perform expression analysis of DC isolated from human skin.

However, to confirm our finding that c-Jun is expressed in DCs, we have analyzed c-Jun expression in human skin employing additional DC markers associated mainly with cDC2 (CD1a, CD1c and CD14). We did this for two reasons. First, our analysis of DC subsets in murine skin showed that cDC2 were the most prominent producers of IL-23 and CCL2 after deletion of c-Jun (Fig. 5A). Secondly, analysis of a published RNA-Seq Dataset (GSE121212) showed cDC2 transcripts (*Cd1c*, *Cd14*) to be enriched in lesional psoriasis, whereas transcripts for cDC1 (*Xcr1*) and Langerhans cells (*Cd207*) were reduced (Fig. 7A). Consistent to our initial results, which showed c-Jun expression in CD11c⁺ cells, we found c-Jun to be expressed in CD1a, CD1c and CD14⁺ cells but also in other cells (Fig. 7B). Quantification further revealed increased c-Jun expression in DCs of lesional psoriasis (Fig. 7C). Lastly, although DCs cells represent only a minor fraction of c-Jun⁺ cells in lesional psoriasis, we show by immunofluorescence that IL-23 expression is most prominent in these cells (Fig 7E).

Minor points:

3. In the results part and in the figure legends of the manuscript, the authors always use the name of the substance, Imiquimod, but in the Figure itself the tradename, Aldara. They either should change Aldara into Imiquimod in the figures or also mention Aldara in the body of the text (in the current version of the manuscript it is only mentioned at the end of the manuscript in the material and method section).

Thank you for this comment. Aldara was replaced by IMQ throughout the manuscript.

4. In figure 2A, the authors show that after deletion of c-Jun in CD11c⁺ cells less pDCs are recruited to the skin. However, even in wildtype mice pDCs represent less than 1% of the cells. As IMQ-treatment also leads to the recruitment of other immune cells, it would be helpful to show total numbers of recruited pDCs to exclude a proportional effect.

We have addressed this point by showing the total numbers of pDC infiltrating the skin at the critical 12h time-point in our new Fig. 2C.

5. The authors claim that c-Jun in DCs is "necessary for the induction of various pro-inflammatory cytokines and key mediators of IMQ-induced skin inflammation (Fig. 4D-F)". However, except for IL-23p19 and IL-17 all shown cytokines and mediators are equally induced after 32h and only reduced after 48h. The authors should change the sentence accordingly and discuss, why c-Jun is only necessary at 48h but not 32h for some mediators (e.g. CXCL1).

This overstatement was rephrased accordingly in the manuscript.

6. The authors further claim that c-Jun is "essential for mRNA expression of the cytokines Il23p19, Il22, Il6 and the chemokines Ccl20 and Cxcl1 (Fig. EV3A)". However, Figure EV3A shows that all of those cyto- and chemokines are induced, especially in case of Il23p19, Il6 and Cxcl1. Therefore, the sentence should be changed as c-Jun is not "essential" but more a positive regulator.

These data are now shown in Fig 4C and Fig EV2B and we rephrased the sentence accordingly.

7. The material and methods section is in some parts short and does not contain all details necessary to repeat the experiments.

We apologize for brevity in the material and methods section, which was due to limitations in space. We have now expanded it to describe all procedures accordingly.

First adviser's comments:

"In general, it is a good manuscript with some interesting information. It is a bit "oversold" and previous data on AP-1 regulation of IL-23 should be cited (see reviewer comments). Most importantly, some of the effects are not robust (see for example data in Fig. 1G-I, where only a few mice with higher numbers of immune cells are responsible for the statistical significance. This high variability may result from the mixed genetic background (which is indeed not defined...). The authors should at least discuss these limitations. The number of replicates for some in vivo experiments is also problematic. In particular, all experiments shown in Fig. EV4 were performed with only three mice. This is clearly insufficient, in particular since one mutant mouse does not even show a phenotype. The n number would have to be increased to at least 6-8. In addition, some important controls are missing. For example, JNKi control in CD11-Cre-Junfl/fl mice (Fig. 6D). This is an important control and would show the specificity of the inhibitor. Overall, I agree with various criticisms of the reviewers [...]. However, I do not think that the request to use another Cre mouse is justified - this would be a completely new story and the identification of the responsible DC subset would not add much to the overall message. In addition, it is indeed not possible to get large amounts of fresh psoriatic skin for ethical reasons. Overall, it is a borderline case, but I think that a revision is justified. Clearly, such a revision would be rather extensive, but it should be possible."

Many thanks to the advisor for a positive overall evaluation of our manuscript, which gave us the chance to submit a revised and greatly improved version of our manuscript.

Detailed comments from this adviser on the rebuttal letter:1/

Reviewer #1: Comments on novelty/model system: -

The mixed background is mentioned, but not which mixture. In general mixed backgrounds are problematic for immunological experiments and this may explain the high variability of some of the data (.e.g. 1G.I). This problem should at least be discussed.

We apologize for not having specified the mouse background more precisely. We also agree that this can result in higher variability. However, *c-Jun/JunB*^{Δ/Δ} K5-Cre-ER^{T2} must be kept in a mixed background for a phenotype to develop. Thus, our rationale was to use all our mouse models in a mixed background, to be more concise. We discuss this in the new version of the manuscript in the methods section and discussion.

Previous publications reporting the dependence of IL-23 secretion by dendritic cells by the JNK pathway should be cited.

We added this publication to the Discussion section.

Major point #1:

- The tumor data can be kept in the manuscript. The issue can be solved by explaining in details why tumor experiments were performed.

We rephrased the results section and discussion, to explain more clearly our rationale for performing the tumor experiments. See also response to Rev. 1

- Only three mice per genotype were analysed in the additional mouse model of psoriasis. The n number should be increased.

We increased the number of experimental mice in our analysis. We confirmed our initial results and show statistically significant improvement of the overall psoriatic phenotype and a reduced frequency in infiltrating monocytes and $\gamma\delta$ T cells. However, similar to our first experiment one mouse did not improve ("Non-responder"). It remains to be investigated, whether, for example, a higher dosage of the inhibitor or longer treatment could have helped for these "Non-Responders". These data are now shown in the new Fig 6 I-N.

Moreover, we also repeated our experiment with the JNK inhibitor in the IMQ-induced skin inflammation model to add additional controls. These results confirm that the JNK inhibitor is only effective when c-Jun or TLR7 are present in DCs and thus support our hypothesis that the TLR7/JNK/c-Jun axis represents the critical signaling pathway in IMQ-induced psoriasis-like skin inflammation. These results are shown in our new Fig. 6A-H and Fig. EV3A-F.

Major point #2:

The cells in which the deletion occurred were not shown and the efficacy of knockout at the protein level is not shown for any of the mouse models. This should be included in a revision.

We revised Fig. EV1 B-D and now show the loss of c-Jun in the relevant skin cell populations (keratinocytes, immune cells, DCs).

Regarding the different controls, the rebuttal letter gives a good explanation, but all controls should be explicitly mentioned in the manuscript

All controls/treatments are now mentioned in every Figure by a color code and explained in detail in the Figure legends and Materials and Methods section.

Major point #3: - Clarify in each experiment why a particular mouse model was used. The Mx-Cre model is inducible and therefore, is not suitable to study development/maturation of pDCs (Fig. EV2A-C).

For *in vitro* experiments the Mx-Cre model was used due to its superior (near complete) deletion efficiency in DCs. Deletion was induced *in vivo* at least 9 days before isolating bone marrow cells to make sure that no residual effects of poly I: C treatment could interfere with IMQ stimulation. Second, we employed the Mx1-Cre model as an alternative to the CD11c-Cre model to exclude Cre-mediated artifacts. A detailed description was added to the methods section. See also response to Rev. 1.

In principle, we do not see any problem employing an inducible mouse model to study cellular development, however, we agree with the Reviewer that the Mx-Cre model is not ideal for pDC development. Thus, Fig. EV2 was revised and pDC development/maturation was analyzed in the spleen and skin-draining lymph nodes of *c-Jun^{Δ/Δ} CD11c-Cre* mice *in vivo* (new Appendix Fig S2A – F).

Minor comments, last point: - The reviewer did not understand this point and therefore, the authors should more explicitly point to the different roles of AP1 family members in keratinocytes vs DCs in the context of psoriasis. However, this finding indeed complicates the treatment and I am not convinced that JNK inhibition will solve the problem. This should be more carefully discussed.

We have provided a better explanation in the Discussion (see also response to Rev. 1).

Reviewer #3: Major point 1:

The use of another Cre mouse is not essential. This would require repetition of a large number of mouse experiments and the additional info would be limited. However, the limitations of the approach should be clearly mentioned and the proposed *in vitro* experiments should be performed.

We now better explained the limitations of our mouse model, which are stated in the Results (Fig 5) and in the Discussion section, and the *ex vivo* experiments were performed to address the role of macrophages (see also response to Rev. 3).

Minor point: The material and methods section should be expanded.

The Materials and Methods section was expanded.

The second advisor agreed that additional Cre mouse was not necessary.

This is greatly appreciated.

References:

Boltjes A, van Wijk F (2014) Human dendritic cell functional specialization in steady-state and inflammation. *Frontiers in immunology* 5: 131-131

Drobits B, Holcman M, Amberg N, Swiecki M, Grundtner R, Hammer M, Colonna M, Sibilio M (2012) Imiquimod clears tumors in mice independent of adaptive immunity by converting pDCs into tumor-killing effector cells. *J Clin Invest* 122: 575-585

Riol-Blanco L, Ordovas-Montanes J, Perro M, Naval E, Thiriot A, Alvarez D, Paust S, Wood JN, von Andrian UH (2014) Nociceptive sensory neurons drive interleukin-23-mediated psoriasiform skin inflammation. *Nature* 510: 157-161

Tortola L, Rosenwald E, Abel B, Blumberg H, Schäfer M, Coyle AJ, Renaud J-C, Werner S, Kisielow J, Kopf M (2012) Psoriasiform dermatitis is driven by IL-36-mediated DC-keratinocyte crosstalk. *The Journal of Clinical Investigation* 122: 3965-3976

Zaba LC, Fuentes-Duculan J, Eungdamrong NJ, Johnson-Huang LM, Nograles KE, White TR, Pierson KC, Lentini T, Suárez-Fariñas M, Lowes MA et al (2010) Identification of TNF-related apoptosis-inducing ligand and other molecules that distinguish inflammatory from resident dendritic cells in patients with psoriasis. *The Journal of allergy and clinical immunology* 125: 1261-1268.e1269

14th Oct 2020

Dear Maria,

Thank you for the submission of your revised manuscript to EMBO Molecular Medicine. We have now received the enclosed reports from the two referees who reviewed the new version of your manuscript. As you will see, they are supportive of publication, pending minor revisions, and I am thus pleased to inform you that we will be able to accept your manuscript pending the following final minor amendments:

1) Referees' comments:

Please address referees #4 and #5's comments. Regarding referee #5, point A, please rearrange the manuscript as you see fit to improve focus, clarity and interest. After discussion with my colleagues, we leave at your discretion the choice of figures you would like to have in the main manuscript, or in the Appendix. See also detailed instructions here:

2) Please include a .docx formatted version of the manuscript text (including legends for main figures, EV figures and tables). Please make sure that the changes are highlighted to be clearly visible.

3) Please include individual production quality figure files as .eps, .tif, .jpg (one file per figure).

4) Please include a .docx formatted letter INCLUDING the reviewers' reports and your detailed point-by-point responses to their comments. As part of the EMBO Press transparent editorial process, the point-by-point response is part of the Review Process File (RPF), which will be published alongside your paper.

5) Please include in your main manuscript a Data Availability section listing the accession numbers for the primary datasets produced in this study and deposited in public repositories. If no primary datasets has been produced, please indicate "This study includes no data deposited in external repositories".

6) We would also encourage you to include the source data for figure panels that show essential data. Numerical data should be provided as individual .xls or .csv files (including a tab describing the data). For blots or microscopy, uncropped images should be submitted (using a zip archive if multiple images need to be supplied for one panel). Additional information on source data and instruction on how to label the files are available at

7) Our journal encourages inclusion of *data citations in the reference list* to directly cite datasets that were re-used and obtained from public databases. Data citations in the article text are distinct from normal bibliographical citations and should directly link to the database records from which the data can be accessed. In the main text, data citations are formatted as follows: "Data ref: Smith et al, 2001" or "Data ref: NCBI Sequence Read Archive PRJNA342805, 2017". In the Reference list, data citations must be labeled with "[DATASET]". A data reference must provide the database

name, accession number/identifiers and a resolvable link to the landing page from which the data can be accessed at the end of the reference. Further instructions are available at .

8) The paper explained: EMBO Molecular Medicine articles are accompanied by a summary of the articles to emphasize the major findings in the paper and their medical implications for the non-specialist reader. Please provide a draft summary of your article highlighting

9) For more information: There is space at the end of each article to list relevant web links for further consultation by our readers. Could you identify some relevant ones and provide such information as well? Some examples are patient associations, relevant databases, OMIM/proteins/genes links, author's websites, etc...

10) Author contributions: Erwin F. Wagner is listed under EWF, please update.

11) Every published paper now includes a 'Synopsis' to further enhance discoverability. Synopses are displayed on the journal webpage and are freely accessible to all readers. They include a short stand first (maximum of 300 characters, including space) as well as 2-5 one-sentences bullet points that summarizes the paper. Please write the bullet points to summarize the key NEW findings. They should be designed to be complementary to the abstract - i.e. not repeat the same text. We encourage inclusion of key acronyms and quantitative information (maximum of 30 words / bullet point). Please use the passive voice. Please attach these in a separate file or send them by email, we will incorporate them accordingly.

Please also suggest a striking image or visual abstract to illustrate your article. If you do please provide a png file 550 px-wide x 400-px high.

12) As part of the EMBO Publications transparent editorial process initiative (see our Editorial at <http://embomolmed.embopress.org/content/2/9/329>), EMBO Molecular Medicine will publish online a Review Process File (RPF) to accompany accepted manuscripts. In the event of acceptance, this file will be published in conjunction with your paper and will include the anonymous referee reports, your point-by-point response and all pertinent correspondence relating to the manuscript. Let us know whether you agree with the publication of the RPF and as here, if you want to remove or not any figures from it prior to publication.

I look forward to receiving your revised manuscript.

Yours sincerely,

Lise Roth

Lise Roth, PhD
Editor
EMBO Molecular Medicine

To submit your manuscript, please follow this link:

Link Not Available

***** Reviewer's comments *****

Referee #4 (Comments on Novelty/Model System for Author):

The authors performed additional experiments and included additional controls as requested by the reviewers. The manuscript was significantly revised. These changes/additions have clearly improved the quality of the manuscript. The following minor points should be addressed:

- 1.) Results, first paragraph: The authors should be more precise regarding the targeting. CD11-Cre is not specific for DC and K5 is not specific for keratinocytes. I suggest to write, "xxx mice, which target predominantly dendritic cells or keratinocytes, respectively". In this paragraph I also suggest to clarify that the Mx-1 promoter is activated by interferon, which is induced by poly(I:C).
- 2.) Fig. 3F should be 3E and vice versa to be consistent with the citation in the text.
- 3.) Fig. 3F does not show tumor regression as mentioned in the text (page 8, line 17).
- 4.) Last sentence of page 9 that is highlighted in yellow: I suggest the following change: ...IL-23 and IL-17 levels were not increased by imiquimod treatment in the skin of c-jun d/d CD11c-Cre mice.
- 5.) Page 14, first paragraph: I suggest to express this more carefully: These results suggest that c-JUN expression in cDC2 may also promote psoriasis in patients by controlling..
- 6.) Legend to Fig. 4H. Please add "(see Fig. 4J)" at the end of the sentence.
- 7.) Legend to Fig. 4J. Please explain the abbreviation "LAL"
- 8.) Legend to Fig. 5A: The word "independent" should be deleted, since this is the result of one experiment (using pooled RNA)

Referee #5 (Remarks for Author):

The authors have achieved a thorough investigation of the role of c-Jun expression in CD11c⁺ cells in the pathogenesis of two preclinical mouse models of psoriasis as well as in a mouse model of immunotherapy against melanoma.

The data clearly show that genetic inactivation of c-Jun specifically in CD11c⁺ cells ameliorates disease in an acute and self-resolving mouse model of psoriasis induced by topical application of Imiquimod (IMQ).

The underlying mechanisms were dissected by performing a series of elegant and complementary experiments, including several that have been performed to address the reviewers' comments from the first round of reviews, which has strongly enforced the robustness of the manuscript.

- 1) Loss of c-Jun expression in total skin CD11c⁺ cells leads to a strong and significant reduction of the expression of the Il23p19 and Il17a genes in the back skin of mice treated with IMQ, and to a significant decrease of IL-23 and IL-17A protein levels in cutaneous lysate.
- 2) The frequency of IL-17A⁺ gamma-delta T cells is significantly decreased in the skin of IMQ-treated mice conditionally inactivated for c-Jun expression in CD11c⁺ cells.
- 3) Injection of recombinant IL-23 in the skin of IMQ-treated mice conditionally inactivated for c-Jun expression in CD11c⁺ cells (c-Jun_Delta-DC mice) restores the increase in epidermal thickness.
- 4) Bone marrow-derived dendritic cells (BMDC) genetically inactivated for c-Jun show a significant decrease in their expression of the Il23p19 gene and of the IL-23 protein upon their in vitro stimulation with IMQ, similar to the decrease induced by pharmacological treatment of WT BMDC cultures with a JNK inhibitor.
- 5) The promoter of the Il23p19 gene includes a putative binding site for c-Jun/AP1. This promoter is proven to bind c-Jun based on a reporter luciferase assay. This binding is abrogated by mutations of the putative binding site as well as by pharmacological inhibition of JNK-dependent c-Jun phosphorylation. Binding of c-Jun to the promoter in BMDC is enhanced upon IMQ treatment.
- 6) The induction of the c-Jun and Il23p19 genes is the strongest in type 2 conventional dendritic cells (cDC2s) as compared to type 1 conventional dendritic cells (cDC1s), Langerhans cells and macrophages isolated from the back skin of IMQ-treated mice, and this increase is selectively lost in cDC2s from cJun_Delta-DC mice whereas this is not the case for macrophages.
- 7) Treatment of the mice with a JNK inhibitor at the same time as topical IMQ application recapitulates the decrease in the skin inflammation observed in c-Jun_Delta-DC mice for all parameters measured (epidermal thickness, water loss, keratinocyte proliferation and differentiation, monocyte, neutrophil and gamma-delta T cell infiltration).
- 8) In another preclinical model of mouse psoriasis, due to conditional genetic inactivation of c-Jun and JunB in keratinocytes, treatment of the mice with a JNK inhibitor also significantly decrease skin inflammation for all parameters measured (ear and paw swelling, epidermal thickening, monocyte, neutrophil and gamma-delta T cell infiltration).

All of these data strongly support the conclusion that, in two different mouse preclinical models of psoriasis, JNK-dependent phosphorylation of c-Jun in DCs promote their production of IL-23 which in turn drives the activation of skin gamma-delta T cell for IL-17A expression and the downstream recruitment of neutrophils, leading to skin inflammation.

To the best of my appreciation, this study is very convincing. It is based on a thorough series of rigorous, elegant and complementary experiments of high technical quality. The conclusions drawn by the authors thus seem very robust. To the best of my knowledge, the discovery that cell-intrinsic c-Jun signaling in DCs is critical to promote their expression of pro-inflammatory cytokines

downstream of TLR7 triggering is original.

Moreover, the author provide data supporting the conclusion that the observations made in preclinical mouse models are relevant to psoriatic patients.

1) Genes known to be selectively expressed in human cDC2s or cDC2-like cells are highly expressed in human lesional psoriatic skin as compared to non-lesional skin and to skin from healthy controls, whereas genes selectively expressed in Langerhans cells or in cDC1s are decreased.

2) c-Jun is highly expressed in cells bearing markers of cDC2 or cDC2-like cells in human lesional psoriatic skin, with a significantly frequency of such c-Jun+ cDC2/cDC2-like cells in human lesional psoriatic skin as compared to non-lesional skin and to skin from healthy controls.

3) IL-23p19 is highly expressed in cells bearing markers of cDC2 or cDC2-like cells in human lesional psoriatic skin.

4) TLR7/8 stimulation induces JNK activation in human MoDCs, and pharmacological inhibition of JNK leads to a strong and significant decrease of their expression of IL-23. This is not only the case with the synthetic nucleic acid base analog R848 but also with complexes between the LL37 anti-microbial peptide and RNA oligonucleotides which mirrors the natural LL37/RNA complexes found in psoriatic skin.

Hence, the novelty of the study is high. The demonstration that pharmacological inhibition of JNK strongly decreases skin inflammation in two preclinical mouse models of psoriasis has a high medical impact.

The authors have made major efforts to address the concerns or criticisms that the reviewers raised during the first review round, which has led to a significant improvement of the manuscript based both on the inclusion of many novel results and on completion of the information on the designs of experiments and on the technical protocols.

A few minor points, detailed below, could be addressed to improve the paper, depending whether the other reviewers and the editors concur with that view.

Minor points.

A) Improving paper focus, clarity and interest, in particular for the nonspecialist, by shuttling some data between main, EV and supplementary information, or even removing the data on pDCs and on melanoma.

Since Ccl2 responses, and by inference pDCs, are shown to be dispensable in the promotion of skin inflammation/psoriasis, I suggest shuttling these data to the EV or Supplemental information, or even removing it. This seems all the more logical as both pDCs and the IFN-I pathway were shown to be dispensable for IMQ-induced skin inflammation by using two different genetic model of specific pDC depletion, DC-E2-2-/- mice for constitutive lack of pDCs and hBDCA-2DTR mice for conditional ablation of pDCs (Wohn et al, 2013, cited in the manuscript).

As was pointed out by at least two of the other reviewers/advisers, the data on melanoma interrupts the logical flow of the story on skin inflammation/psoriasis, distracting the reader from the main focus of the story and making the reading of the paper hard to follow. This data on melanoma should thus also be moved to the supplementary data, or even removed.

Maybe the data on pDCs and Ccl2 in IMQ-induced activation/skin inflammation or in melanoma

immunotherapy (Fig. 2; Fig. 3) could actually be kept for another paper focused on the mechanisms controlling TLR7-triggered pDC recruitment to the skin and pDC antitumor functions in melanoma. This would then allow to move back some of the EV figure to main figures, such as Figure EV1 and Figure EV3.

B) To compare c-Jun and Il23p19 gene expression across skin cell subsets, that data must be shown without normalization for each cell type to expression levels in untreated mice. From figures 5A and EV2F, it is not possible to conclude that cDC2 are the main expressers of c-Jun and Il23p19, since for each cell type expression levels are normalized to 1 for the untreated conditions. Data are thus shown as fold change in cells from IMQ-treated mice of the ground levels in cells from untreated mice. However, whether differences exist in ground level expression between cell types is not documented.

C) For immunohistofluorescence on human skin sections, figure legends must precise of how many sections from how many different individuals the micrograph shown are representative. This is stated for Fig7. B-C (two samples [different individuals?] for each conditions, with four randomly chosen fields per skin section) and Fig. EV4A. However, this is not specified for Fig. 7D-E, and Fig. EV4D-F.

D) A quantitation is required to conclude that prominent co-expression of c-Jun with IL-23 could be detected in human skin CD1a+, CD1c+ or CD14+ cells (Fig 7E and EV4F). This would require performing immunofluorescence stainings with anti-CD1a, CD1c and CD14 in the same color, to quantitate the percent of IL-23+ cells expressing at least one of these markers and/or c-Jun.

E) Could the authors explain how come that IL-23 titers are 10 times lower in Fig. S4D as compared to Fig. 4F?

F) Could the authors comment on the fact that very high levels of Ifng are induced in the cells purified as BMpDCs upon stimulation with IMQ or IL-12 (Fig. S5)? One may wonder whether this could not reflect contamination of the cell population by B220+ NK cells. Even in GM-CSF BM-DC cultures, contaminating NK cells have been shown to be present and to account for all of the Ifng expression induced upon LPS stimulation (PMID: 32665300). Thus, this is also likely to occur in FLT3-L BM-DC cultures.

Point-by-point response (Reviewer):**Referee #4** (Comments on Novelty/Model System for Author):

The authors performed additional experiments and included additional controls as requested by the reviewers. The manuscript was significantly revised. These changes/additions have clearly improved the quality of the manuscript. The following minor points should be addressed:

1.) Results, first paragraph: The authors should be more precise regarding the targeting. CD11-Cre is not specific for DC and K5 is not specific for keratinocytes. I suggest to write, "xxx mice, which target predominantly dendritic cells or keratinocytes, respectively". In this paragraph I also suggest to clarify that the Mx-1 promoter is activated by interferon, which is induced by poly (I: C).

We have improved the wording and changed the sentence in the results section accordingly.

2.) Fig. 3F should be 3E and vice versa to be consistent with the citation in the text.

The tumor data were removed from the manuscript due to the suggestions of Reviewer #1 and Reviewer # 5. The figure nomenclature has changed accordingly.

3.) Fig. 3F does not show tumor regression as mentioned in the text (page 8, line 17).

We have completely removed this figure from the manuscript as specified above.

4.) Last sentence of page 9 that is highlighted in yellow: I suggest the following change: ...IL-23 and IL-17 levels were not increased by Imiquimod treatment in the skin of c-Jun d/d CD11c-Cre mice.

We have changed this sentence as suggested.

5.) Page 14, first paragraph: I suggest to express this more carefully: These results suggest that c-JUN expression in cDC2 may also promote psoriasis in patients by controlling.

We weakened our statement in the mentioned paragraph as suggested.

6.) Legend to Fig. 4H. Please add "(see Fig. 4J)" at the end of the sentence.

We added "(see Fig. 4J)" to the Legend of Fig.4H.

7.) Legend to Fig. 4J. Please explain the abbreviation "LAL"

Thank you for this hint. We explain the abbreviation now in the Figure legend of Fig. 4J.

8.) Legend to Fig. 5A: The word "independent" should be deleted, since this is the result of one experiment (using pooled RNA)

The Reviewer is correct. We removed the word "independent".

Referee #5 (Remarks for Author):The authors have achieved a thorough investigation of the role of c-Jun expression in CD11c+ cells in the pathogenesis of two preclinical mouse models of psoriasis as well as in a mouse model of immunotherapy against melanoma. The data clearly show that genetic inactivation of c-Jun specifically in CD11c+ cells ameliorates disease in an acute and self-resolving mouse model of psoriasis induced by topical application of Imiquimod (IMQ).The underlying mechanisms were dissected by performing a series of elegant and complementary experiments, including several that have been performed to address the reviewers' comments from the first round of reviews, which has strongly enforced the robustness of the manuscript.

1) Loss of c-Jun expression in total skin CD11c+ cells leads to a strong and significant reduction of the expression of the Il23p19 and Il17a genes in the back skin of mice treated with IMQ, and to a significant decrease of IL-23 and IL-17A protein levels in cutaneous lysate.

2) The frequency of IL-17A+ gamma-delta T cells is significantly decreased in the skin of IMQ-treated mice conditionally inactivated for c-Jun expression in CD11c+ cells.

3) Injection of recombinant IL-23 in the skin of IMQ-treated mice conditionally inactivated for c-Jun expression in CD11c+ cells (c-Jun_Delta-DC mice) restores the increase in epidermal thickness.

4) Bone marrow-derived dendritic cells (BMDC) genetically inactivated for c-Jun show a significant decrease in their expression of the Il23p19 gene and of the IL-23 protein upon their in vitro stimulation with IMQ, similar to the decrease induced by pharmacological treatment of WT BMDC cultures with a JNK inhibitor.

5) The promoter of the Il23p19 gene includes a putative binding site for c-Jun/AP1. This promoter is proven to bind c-Jun based on a reporter luciferase assay. This binding is abrogated by mutations of the putative binding site as well as by pharmacological inhibition of JNK-dependent c-Jun phosphorylation. Binding of c-Jun to the promoter in BMDC is enhanced upon IMQ treatment.

6) The induction of the c-Jun and Il23p19 genes is the strongest in type 2 conventional dendritic cells (cDC2s) as compared to type 1 conventional dendritic cells (cDC1s), Langerhans cells and macrophages isolated from the back skin of IMQ-treated mice, and this increase is selectively lost in cDC2s from cJun_Delta-DC mice whereas this is not the case for macrophages.

7) Treatment of the mice with a JNK inhibitor at the same time as topical IMQ application recapitulates the decrease in the skin inflammation observed in c-Jun_Delta-DC mice for all parameters measured (epidermal thickness, water loss, keratinocyte proliferation and differentiation, monocyte, neutrophil and gamma-delta T cell infiltration).

8) In another preclinical model of mouse psoriasis, due to conditional genetic inactivation of c-Jun and JunB in keratinocytes, treatment of the mice with a JNK inhibitor also significantly decrease skin inflammation for all

parameters measured (ear and paw swelling, epidermal thickening, monocyte, neutrophil and gamma-delta T cell infiltration).

All of these data strongly support the conclusion that, in two different mouse preclinical models of psoriasis, JNK-dependent phosphorylation of c-Jun in DCs promote their production of IL-23 which in turn drives the activation of skin gamma-delta T cell for IL-17A expression and the downstream recruitment of neutrophils, leading to skin inflammation.

To the best of my appreciation, this study is very convincing. It is based on a thorough series of rigorous, elegant and complementary experiments of high technical quality. The conclusions drawn by the authors thus seem very robust. To the best of my knowledge, the discovery that cell-intrinsic c-Jun signaling in DCs is critical to promote their expression of pro-inflammatory cytokines downstream of TLR7 triggering is original.

Moreover, the author provide data supporting the conclusion that the observations made in preclinical mouse models are relevant to psoriatic patients.

1) Genes known to be selectively expressed in human cDC2s or cDC2-like cells are highly expressed in human lesional psoriatic skin as compared to non-lesional skin and to skin from healthy controls, whereas genes selectively expressed in Langerhans cells or in cDC1s are decreased.

2) c-Jun is highly expressed in cells bearing markers of cDC2 or cDC2-like cells in human lesional psoriatic skin, with a significantly frequency of such c-Jun+ cDC2/cDC2-like cells in human lesional psoriatic skin as compared to non-lesional skin and to skin from healthy controls.

3) IL-23p19 is highly expressed in cells bearing markers of cDC2 or cDC2-like cells in human lesional psoriatic skin.

4) TLR7/8 stimulation induces JNK activation in human MoDCs, and pharmacological inhibition of JNK leads to a strong and significant decrease of their expression of IL-23. This is not only the case with the synthetic nucleic acid base analog R848 but also with complexes between the LL37 anti-microbial peptide and RNA oligonucleotides which mirrors the natural LL37/RNA complexes found in psoriatic skin.

Hence, the novelty of the study is high. The demonstration that pharmacological inhibition of JNK strongly decreases skin inflammation in two preclinical mouse models of psoriasis has a high medical impact.

The authors have made major efforts to address the concerns or criticisms that the reviewers raised during the first review round, which has led to a significant improvement of the manuscript based both on the inclusion of many novel results and on completion of the information on the designs of experiments and on the technical protocols.

A few minor points, detailed below, could be addressed to improve the paper, depending whether the other reviewers and the editors concur with that view.

Minor points.

A) Improving paper focus, clarity and interest, in particular for the non-specialist, by shuttling some data between main, EV and supplementary information, or even removing the data on pDCs and on melanoma.

Since Ccl2 responses, and by inference pDCs, are shown to be dispensable in the promotion of skin inflammation/psoriasis, I suggest shuttling these data to the EV or Supplemental information, or even removing it. This seems all the more logical as both pDCs and the IFN-I pathway were shown to be dispensable for IMQ-induced skin inflammation by using two different genetic model of specific pDC depletion, DC-E2-2^{-/-} mice for constitutive lack of pDCs and hBDCA-2DTR mice for conditional ablation of pDCs (Wohn et al, 2013, cited in the manuscript).

As was pointed out by at least two of the other reviewers/advisers, the data on melanoma interrupts the logical flow of the story on skin inflammation/psoriasis, distracting the reader from the main focus of the story and making the reading of the paper hard to follow. This data on melanoma should thus also be moved to the supplementary data, or even removed.

Maybe the data on pDCs and Ccl2 in IMQ-induced activation/skin inflammation or in melanoma immunotherapy (Fig. 2; Fig. 3) could actually be kept for another paper focused on the mechanisms controlling TLR7-triggered pDC recruitment to the skin and pDC antitumor functions in melanoma. This would then allow to move back some of the EV figure to main figures, such as Figure EV1 and Figure EV3.

We thank the Reviewer for the interest, positive assessment of our revised manuscript and the knowledgeable, accurate summary of our work. We agree that the initial focus of our manuscript, role of c-Jun in TLR7 signaling, has shifted, due to the points raised in the first round of Revisions, to a paper dealing mostly with c-Jun in psoriasis. Thus, we agreed in correspondence with the Editor to remove the tumor melanoma part, which was also suggested initially by Reviewer #1. We also agree with the conclusions drawn by the Reviewer that our data showing that the CCL2-pDC axis is dispensable for IMQ-induced skin inflammation fits to current literature. However, we still think it is relevant to show these data for the following reasons:

1. pDCs initiate psoriasis (Nestle et al, 2005), but whether the recruitment to inflamed skin was essential has remained unclear.
2. Reviewer #1 found the CCL2-pDC angle interesting and asked for more data on the connection of CCL2 and psoriasis.

We thus restructured the manuscript by removing tumor related data.

B) To compare c-Jun and Il23p19 gene expression across skin cell subsets, that data must be shown without normalization for each cell type to expression levels in untreated mice. From figures 5A and EV2F, it is not possible to conclude that cDC2 are the main expressers of c-Jun and Il23p19, since for each cell type expression levels are normalized to 1 for the untreated conditions. Data are thus shown as fold change in cells from IMQ-treated mice of the ground levels in cells from untreated mice. However, whether differences exist in ground level expression between cell types is not documented.

The Reviewer is correct. We wanted to emphasize TLR7-induced changes by normalizing to 1, the untreated controls. However, we agree that it is relevant and interesting to see the differences in the ground level expression and have therefore added these data in a new separate Appendix Fig S6, for clarity we decided to keep Figure 5 and EV2F in its current version as it highlights targets that are induced in a given cell subpopulation and thus contribute significantly to the inflammation triggered by IMQ.

C) For immunohistofluorescence on human skin sections, figure legends must precise of how many sections from how many different individuals the micrograph shown are representative. This is stated for Fig7. B-C (two samples [different individuals?] for each conditions, with four randomly chosen fields per skin section) and Fig. EV4A. However, this is not specified for Fig. 7D-E, and Fig. EV4D-F.

We added the information as suggested.

D) A quantitation is required to conclude that prominent co-expression of c-Jun with IL-23 could be detected in human skin CD1a+, CD1c+ or CD14+ cells (Fig 7E and EV4F).

This would require performing immunofluorescence stainings with anti-CD1a, CD1c and CD14 in the same color, to quantitate the percent of IL-23+ cells expressing at least one of these markers and/or c-Jun.

The Reviewer is correct. The proposed experiment would also address this point in a clever way. However, to establish a staining with so many different antibodies would also be difficult and require additional human fresh psoriatic material. After correspondence with the Editor, we decided to express this statement more carefully in the manuscript and weaken the conclusions drawn from this finding.

E) Could the authors explain how come that IL-23 titers are 10 times lower in Fig. S4D as compared to Fig. 4F?

The reviewer is correct. The titers are different because different numbers of cells were plated in the indicated experiments. We have no added this information in the corresponding figure legend.

F) Could the authors comment on the fact that very high levels of Ifng are induced in the cells purified as BMpDCs upon stimulation with IMQ or IL-12 (Fig. S5)? One may wonder whether this could not reflect contamination of the cell population by B220+ NK cells. Even in GM-CSF BM-DC cultures, contaminating NK cells have been shown to

be present and to account for all of the Ifng expression induced upon LPS stimulation (PMID: 32665300). Thus, this is also likely to occur in FLT3-L BM-DC cultures.

This is an interesting question raised by the Reviewer. To control for a possible NK cell contamination in FLT3L cultures we performed flow cytometry using different NK cell markers (NK1.1, NK46p and NKG2D), however, we could not find any population expressing any of these markers neither among B220 cells nor in the culture itself (see below). Moreover, pDCs have been shown in previous studies to be capable of Ifng production (Suto et al, 2005). Thus, it will be interesting in the future to explore to which extent pDCs could contribute to Ifng expression in tissue inflammation.

References

Nestle FO, Conrad C, Tun-Kyi A, Homey B, Gombert M, Boyman O, Burg G, Liu Y-J, Gilliet M (2005) Plasmacytoid dendritic cells initiate psoriasis through interferon- α production. *The Journal of experimental medicine* 202: 135-143

Suto A, Nakajima H, Tokumasa N, Takatori H, Kagami S-i, Suzuki K, Iwamoto I (2005) Murine Plasmacytoid Dendritic Cells Produce IFN- γ upon IL-4 Stimulation. *The Journal of Immunology* 175: 5681-5689

14th Jan 2021

Dear Maria,

Thank you for the submission of your revised manuscript to EMBO Molecular Medicine, and please accept my apologies for the delay in getting back to you, which is due to the limited staff and high number of manuscripts submitted to our office during the holiday season, and to the lockdown situation. I have been through your revisions and point-by-point rebuttal letter, and I'm happy to say that we will be ready to accept your manuscript once the following minor editorial concerns will be addressed:

1) Main manuscript text:

- Please answer/correct the changes suggested by our data editors in the main manuscript file (in track changes mode). This file will be sent to you in the next couple of days. Please use this file for any further modification.
- Please remove the highlighted text.
- Material and methods: We note that you have included Material and Methods in the Appendix file. As we are not limited in space, you might want to include (some of) these methods in the main manuscript file. Please provide antibody dilutions. Please indicate whether or not the cells were tested for mycoplasma contamination. Regarding human subjects, please include the full statement that informed consent was obtained from all subjects and that the experiments conformed to the principles set out in the WMA Declaration of Helsinki and the Department of Health and Human Services Belmont Report.
- Please replace "Declaration of interests" by "Conflicts of interests"

2) Figures:

Figure 2E and Figure 2F are not referenced in the main text, please carefully check the callouts for all figures.

3) Checklist:

- section C/7: please indicate whether the cells were tested for mycoplasma contamination
- section E/12: please include the full statement that informed consent was obtained from all subjects and that the experiments conformed to the principles set out in the WMA Declaration of Helsinki and the Department of Health and Human Services Belmont Report.

4) Thank you for providing a synopsis. I slightly edited it to match our style and format, please let me know if you agree with the following:

Based on genetically engineered mouse models (GEMMs) and human psoriasis biopsies, this study suggests that c-Jun in Dendritic Cells (DC) contributes to psoriasis by controlling CCL2 and IL-23 production, and further identifies the JNK/c-Jun axis as a druggable target.

- TLR7 (IMQ)-induced skin inflammation was attenuated in mice lacking c-Jun in DCs.
- TLR7/JNK/c-Jun signalling was required for CCL2 and IL-23 transcription in human and murine DCs.
- c-Jun was co-expressed with CCL2 and IL-23 in type-2/inflammatory DCs of human psoriatic skin.
- Treatment with JNK inhibitor alleviated skin inflammation in mouse models of psoriasis.

I look forward to receiving your revised manuscript.

With my best wishes,

Lise

Lise Roth, PhD
Editor
EMBO Molecular Medicine

To submit your manuscript, please follow this link:

Link Not Available

Photos 400-800 DPI

*Additional important information regarding figures and illustrations can be found at <https://bit.ly/EMBOPressFigurePreparationGuideline>

The system will prompt you to fill in your funding and payment information. This will allow Wiley to send you a quote for the article processing charge (APC) in case of acceptance. This quote takes into account any reduction or fee waivers that you may be eligible for. Authors do not need to pay any fees before their manuscript is accepted and transferred to our publisher.

The authors performed the requested editorial changes.

YOU MUST COMPLETE ALL CELLS WITH A PINK BACKGROUND ↓
PLEASE NOTE THAT THIS CHECKLIST WILL BE PUBLISHED ALONGSIDE YOUR PAPER

Corresponding Author Name: Maria Sibilia
Journal Submitted to: EMBO Molecular Medicine
Manuscript Number: EMM-2020-12409